# GL-LowPopArt: A Nearly Instance-Wise Minimax-Optimal Estimator for Generalized Low-Rank Trace Regression

Junghyun Lee [1]   Kyoungseok Jang [2]   Kwang-Sung Jun [3]   Milan Vojnović [4]   Se-Young Yun [1]

## Abstract

We present GL-LowPopArt, a novel Catoni-style estimator for generalized low-rank trace regression. Building on LowPopArt (Jang et al., 2024), it employs a two-stage approach: nuclear norm regularization followed by matrix Catoni estimation. We establish state-of-the-art estimation error bounds, surpassing existing guarantees (Fan et al., 2019; Kang et al., 2022), and reveal a novel experimental design objective, $\mathrm{GL}(\pi)$. The key technical challenge is controlling bias from the nonlinear inverse link function, which we address by our two-stage approach. We prove a *local* minimax lower bound, showing that our GL-LowPopArt enjoys instance-wise optimality up to the condition number of the ground-truth Hessian. Applications include generalized linear matrix completion, where GL-LowPopArt achieves a state-of-the-art Frobenius error guarantee, and **bilinear dueling bandits**, a novel setting inspired by general preference learning (Zhang et al., 2024b). Our analysis of a GL-LowPopArt-based explore-then-commit algorithm reveals a new, potentially interesting problem-dependent quantity, along with improved Borda regret bound than vectorization (Wu et al., 2024).

## 1. Introduction

Low-rank structures are ubiquitous across diverse domains, where the estimation of high-dimensional, low-rank matrices frequently pops up (Chen & Chi, 2018). Beyond simply possessing a low-rank structure, real-world observations are often subject to nonlinearities. One ubiquitous example is modeling discrete event occurrences by the Poisson point processes (Mutný & Krause, 2021; Kingman, 1992), such as crime rate (Shirota & Gelfand, 2017) and environmental modeling (Heikkinen & Arjas, 1999). In news recommendation and online ad placement, outputs are often quantized, representing categories such as "click" or "no click" (Bennett & Lanning, 2007; Richardson et al., 2007; Stern et al., 2009; Li et al., 2010; 2012; McMahan et al., 2013). Other applications involve predicting interactions between multiple features, including hotel-flight bundles (Lu et al., 2021), online dating/shopping (Jun et al., 2019), protein-drug pair searching (Luo et al., 2017), graph link prediction (Berthet & Baldin, 2020), stock return prediction (Fan et al., 2019), and recently, even preference learning (Zhang et al., 2024b) among others. In these settings, it is natural to model the problem as matrix-valued covariates passed through a nonlinear regression model. In particular, when the observations are (assumed to be) sampled from the generalized linear model (McCullagh & Nelder, 1989), these diverse problems fall under the umbrella of *generalized low-rank trace regression* (Fan et al., 2019), which we now describe.

**Problem Setting.** $\boldsymbol{\Theta}_\star \in \mathbb{R}^{d_1 \times d_2}$ is an unknown matrix of rank at most $r \ll d_1 \wedge d_2$, and $\mathcal{A} \subseteq \mathbb{R}^{d_1 \times d_2}$ is an arm-set (e.g., sensing matrices). The learner's goal is to output $\widehat{\boldsymbol{\Theta}}$ of rank at most $r$ that well-estimates $\boldsymbol{\Theta}_\star$ from some observations $\{(\boldsymbol{X}_t, y_t)\}_{t \in [N]}$, collected as follows.

For a given budget $N \in \mathbb{N}$, a *sampling policy (design)* is a sequence $\pi = (\pi_t)_{t \in [N]} \subset \mathcal{P}(\mathcal{A})^{\otimes [N]}$. When the learner uses $\pi$, at each time $t \in [N]$, she samples a $\boldsymbol{X}_t \sim \pi_t$ and observes $y_t$ sampled from *generalized linear model (GLM)* whose (conditional) density is given as follows:

$$p(y_t | \boldsymbol{X}_t; \boldsymbol{\Theta}_\star) \propto \exp\left( \frac{y_t \langle \boldsymbol{X}_t, \boldsymbol{\Theta}_\star \rangle - m(\langle \boldsymbol{X}_t, \boldsymbol{\Theta}_\star \rangle)}{g(\tau)} \right).$$

Here, $m : \mathbb{R} \to \mathbb{R}$ is the log-partition function, $\tau$ is the dispersion parameter, $g : \mathbb{R} \to \mathbb{R}_{>0}$ is a fixed function, and the density is with respect to some known base measure (e.g., Lebesgue, counting). We refer to $\mu := \dot{m}$ as the *inverse link function*. We assume that all components of the GLM, other than $\boldsymbol{\Theta}_\star$, are known to the learner.

[1]Kim Jaechul Graduate School of AI, KAIST, Seoul, Republic of Korea [2]Department of AI, Chung-Ang University, Seoul, Republic of Korea [3]Department of Computer Science, University of Arizona, Tucson, USA [4]Department of Statistics, London School of Economics, London, UK. Correspondence to: Se-Young Yun <yunseyoung@kaist.ac.kr>.

*Proceedings of the 42$^{nd}$ International Conference on Machine Learning*, Vancouver, Canada. PMLR 267, 2025. Copyright 2025 by the author(s).

For clarity, we distinguish between two learning setups. In the *adaptive scenario*, each $\pi_t \in \mathcal{P}(\mathcal{A})$ may depend on past observations. This setting is standard in interactive learning problems such as bandits (Lattimore & Szepesvári, 2020) and active learning (Settles, 2012). In the *nonadaptive (passive) scenario*, $\pi_t = \pi$ for a known $\pi \in \mathcal{P}(\mathcal{A})$ fixed before the interaction begins. Despite the difference, we omit the $t$-dependence from here on, as our algorithm in the adaptive scenario only switches policy once: $\pi_1$ in Stage I and a Stage I-dependent $\pi_2$ in Stage II.

**Related Works.** Owing to its ubiquity, much work have been done in providing statistically and computationally efficient estimators for this problem, both generally (Fan et al., 2019; Kang et al., 2022) and in specific scenarios such as *generalized linear matrix completion* (Cai & Zhou, 2013; 2016; Davenport et al., 2014; Lafond, 2015; Lafond et al., 2014; Klopp, 2014; Klopp et al., 2015) and learning low-rank preference matrix (Rajkumar & Agarwal, 2016). Corresponding minimax lower bounds have also been proven that are tight with respect to rank $r$, dimension $d_1, d_2$, and sample size $N$; see Appendix A for further related works.

**Main Contributions.** While prior work has made significant progress, a crucial aspect has been overlooked: the instance-specific nature of curvature. To our knowledge, all the existing analyses rely on worst-case bounds for curvature, neglecting its variation and obscuring the problem's true difficulty. For example, known performance guarantees for generalized linear matrix completion depend inversely w.r.t. $\min_{|z| \leq \gamma} \dot{\mu}(z)$, where $\gamma > 0$ is such that $\max_{i,j} |(\boldsymbol{\Theta}_\star)_{ij}| \leq \gamma$ and $\dot{\mu}$ is the derivative of the inverse link function. For instance, when $\mu(z) = (1 + e^{-z})^{-1}$, this leads to a dependence of $e^\gamma$ (Faury et al., 2020). This dependency is instance-*independent*, in the sense that it arises from the worst-case $\dot{\mu}$ over the entry-wise domain $[-\gamma, \gamma]$, rather than adapting to the specific instance $\boldsymbol{\Theta}_\star$.

Our contributions are as follows:

- We propose GL-LowPopArt, an extension of LowPopArt (Jang et al., 2024) to generalized low-rank trace regression, which requires careful bias control of one-sample estimators during matrix Catoni estimation (Minsker, 2018). We prove its *instance-wise statistical rate* for an arbitrary design $\pi \in \mathcal{P}(\mathcal{A})$ (Theorem 3.1): ignoring logarithmic factors,

$$\left\| \widehat{\boldsymbol{\Theta}} - \boldsymbol{\Theta}_\star \right\|_F^2 \lesssim \frac{r\,\mathrm{GL}(\pi)}{N} \lesssim \frac{r(d_1 \vee d_2)}{N\lambda_{\min}(\boldsymbol{H}(\pi; \boldsymbol{\Theta}_\star))},$$

where $\mathrm{GL}(\pi)$ (Eqn. (8)) is a new quantity that effectively captures the nonlinearity and the arm-set geometry, and $\lambda_{\min}(\boldsymbol{H}(\pi; \boldsymbol{\Theta}_\star))$ is the minimum eigenvalue of the Hessian of the negative log-likelihood loss at

$\boldsymbol{\Theta}_\star$. In the active scenario, one can directly optimize the error bound as $\min_{\pi \in \mathcal{P}(\mathcal{A})} \mathrm{GL}(\pi)$. (Section 3)

- We prove the ***first instance-wise minimax lower bound*** for generalized low-rank trace regression (Theorem 4.1): for a fixed design $\pi \in \mathcal{P}(\mathcal{A})$ and instance $\boldsymbol{\Theta}_\star$, there is a $\widetilde{\boldsymbol{\Theta}}_\star$ *near* $\boldsymbol{\Theta}_\star$ such that

$$\left\| \widehat{\boldsymbol{\Theta}} - \widetilde{\boldsymbol{\Theta}}_\star \right\|_F^2 \gtrsim \frac{r(d_1 \vee d_2)}{N\lambda_{\max}(\boldsymbol{H}(\pi; \boldsymbol{\Theta}_\star))},$$

where $\lambda_{\max}(\cdot)$ is the maximum eigenvalue. The above lower bound shows that our GL-LowPopArt is nearly instance-wise optimal, up to the condition number, $\lambda_{\max}(\boldsymbol{H}(\pi; \boldsymbol{\Theta}_\star))/\lambda_{\min}(\boldsymbol{H}(\pi; \boldsymbol{\Theta}_\star))$. (Section 4)

- As an application, we revisit the classical problem of *generalized linear matrix completion* (Davenport et al., 2014; Lafond, 2015; Klopp et al., 2015) and show that GL-LowPopArt attains an improved Frobenius error scaling with $(\min_{i,j} \dot{\mu}((\boldsymbol{\Theta}_\star)_{i,j}))^{-1}$, adapting to the instance at hand. This improves upon prior results that depend on the instance-independent, worst-case curvature. (Section 5.1)

- As another application, we propose and tackle **bilinear dueling bandits**, a new variant of generalized linear dueling bandits involving the contextual bilinear preference model of Zhang et al. (2024b). We propose a GL-LowPopArt-based explore-then-commit algorithm and prove its *Borda regret* upper bound (Theorem 5.1): ignoring logarithmic factors,

$$\mathrm{Reg}^B(T) \lesssim (\mathrm{GL}_{\min}(\mathcal{A}))^{1/3} \left(\kappa_\star^B T\right)^{2/3},$$

where $\kappa_\star^B$ is a new curvature-dependent quantity specific to each bandit instance. (Section 5.2)

## 2. Technical Preliminaries

**Notations.** For a $\boldsymbol{A} \in \mathbb{R}^{m \times n}$ with singular values $\sigma_1 \geq \cdots \geq \sigma_{\min\{m,n\}}$, $\|\boldsymbol{A}\|_{\mathrm{nuc}} := \sum_{i=1}^{\min\{m,n\}} \sigma_i$ is its nuclear norm, and $\|\boldsymbol{A}\|_{\mathrm{op}} := \sigma_1$ is its operator (spectral) norm. For $\boldsymbol{B} \in \mathbb{R}^{m \times n}$, their Frobenius inner product is defined as $\langle \boldsymbol{A}, \boldsymbol{B} \rangle := \mathrm{tr}(\boldsymbol{A}^\top \boldsymbol{B})$. For a symmetric $\boldsymbol{A} \in \mathbb{R}^{m \times m}$, $\lambda_i(\boldsymbol{A})$ is its $i$-th largest eigenvalue, $\lambda_{\max} := \lambda_1$, and $\lambda_{\min} := \lambda_m$. On the positive semidefinite cone, define the Loewner order $\preceq$ as $\boldsymbol{A} \preceq \boldsymbol{B}$ if and only if $\boldsymbol{B} - \boldsymbol{A}$ is positive semidefinite. For a $S > 0$, let us denote $\mathcal{B}_i^{d_1 \times d_2}(S) := \{\boldsymbol{X} \in \mathbb{R}^{d_1 \times d_2} : \|\boldsymbol{X}\|_i \leq S\}$ for $i \in \{\mathrm{op}, \mathrm{nuc}, F\}$. $\mathrm{vec} : \mathbb{R}^{d_1 \times d_2} \to \mathbb{R}^{d_1 d_2}$ performs column-wise stacking of a matrix into a vector, and $\mathrm{vec}^{-1}$ is its inverse. $f(n) \lesssim g(n)$ and $f(n) \asymp g(n)$ indicates $f(n) \leq cg(n)$ and $cg(n) \leq f(n) \leq c'g(n)$ for some constants $c, c' > 0$, respectively. Denote $a \wedge b := \min(a, b)$ and $a \vee b := \max(a, b)$. For a $n \in \mathbb{N}$, let $[n] := \{1, 2, \ldots, n\}$. For a set $X$, $\mathcal{P}(X)$ is the set of all probability distributions on $X$.

**General Assumptions.** We now present some assumptions that we consider throughout this paper.

We assume the following for the parameter space $\Omega$:

**Assumption 1.** $\Omega$ is closed and convex, and it satisfies $\Theta \in \Omega \implies \mathrm{Proj}_r(\Theta) \in \Omega$, where $\mathrm{Proj}_r(\Theta)$ is the best rank-$r$ approximation[1] of $\Theta$.

Note that this encompasses $\mathbb{R}^{d_1 \times d_2}$ (unconstrained), $\{\Theta \in \mathbb{R}^{d_1 \times d_2} : \Theta^\top = -\Theta\}$ (skew-symmetric matrices with $r$ even), and $\mathcal{B}_{\mathrm{nuc}}^{d_1 \times d_2}(1)$ (nuclear norm unit ball; also assumed in Jang et al. (2024, Assumption A1)) to name a few.

We impose the following mild assumption on arm set $\mathcal{A}$:

**Assumption 2.** $\mathcal{A} \subseteq \mathcal{B}_{\mathrm{op}}^{d_1 \times d_2}(1)$ and $\mathrm{span}(\mathcal{A}) = \mathbb{R}^{d_1 \times d_2}$.

The first part is a mild assumption that has been considered before in the low-rank bandits (Jang et al., 2024). The second part is an essential assumption, as if not (i.e., if $\mathrm{span}(\mathcal{A}) \neq \mathbb{R}^{d_1 \times d_2}$), one cannot hope to recover $\Theta_\star$ in the direction of $\mathrm{span}(\mathcal{A})^\perp \neq \emptyset$. The matrix completion basis $\mathcal{X}$, for instance, satisfies this assumption.

We consider the following assumption on the log-partition function $m$, common in generalized linear bandits literature (Russac et al., 2021):

**Assumption 3.** $m : \mathbb{R} \to \mathbb{R}$ is three-times differentiable and convex. Moreover, the *inverse link function* $\mu := \dot{m}$ satisfies the following three conditions:

(a) $R_{\max} := \sup_{X \in \mathcal{A}, \Theta \in \Omega} \dot{\mu}(\langle X, \Theta \rangle) < \infty$,

(b) $R_s$-self-concordant for a known $R_s \in [0, \infty)$, i.e., $|\ddot{\mu}(z)| \leq R_s \dot{\mu}(z)$, $z \in \mathbb{R}$,

(c) $\kappa_\star := \min_{X \in \mathcal{A}} \dot{\mu}(\langle X, \Theta_\star \rangle) > 0$.

This includes Gaussian ($m(z) = \frac{1}{2}z^2$), Bernoulli ($m(z) = \log(1 + e^{-z})$), Poisson ($m(z) = e^z$), etc.

# 3. GL-LowPopArt: A Generalized Linear Low-Rank Matrix Estimator

**Additional Notations** We introduce additional notations to describe our algorithm. For $\pi \in \mathcal{P}(\mathcal{A})$ and $\Theta \in \mathbb{R}^{d_1 \times d_2}$, we define the *(vectorized) design/Hessian matrix* as

$$V(\pi) := \mathbb{E}_{X \sim \pi}[\mathrm{vec}(X)\mathrm{vec}(X)^\top], \tag{3}$$

$$H(\pi; \Theta) := \mathbb{E}_{X \sim \pi}[\dot{\mu}(\langle X, \Theta \rangle)\mathrm{vec}(X)\mathrm{vec}(X)^\top], \tag{4}$$

where $H(\pi; \Theta)$ is the Hessian of the population negative log-likelihood: $\Theta \mapsto -g(\tau)\mathbb{E}_{X \sim \pi}[\log p(y|X; \Theta)]$. Observe that $\kappa_\star V(\pi) \preceq H(\pi; \Theta)$, which we will often use, and that $V(\pi) = H(\pi; \Theta)$ when $\mu(z) = z$.

---

[1] Let $\Theta = U\Sigma V^\top$ be its SVD, ordered by its singular values in a decreasing manner. Then $\mathrm{Proj}_r(\Theta) := U_r \Sigma_r V_r^\top$, where the subscript $r$ denotes taking the first $r$ columns.

The following notations are for the matrix Catoni estimator (Catoni, 2012; Minsker, 2018). For any $f : \mathbb{R} \to \mathbb{R}$ and symmetric $M \in \mathbb{R}^{d \times d}$, we define $f(M)$ as $f(M) := U\mathrm{diag}(\{f(\lambda_i)\}_{i \in [d]})U^\top$, where $M = U\Lambda U^\top$ with $\Lambda = \mathrm{diag}(\{\lambda_i\}_{i \in [d]})$ being the eigenvalue decomposition of $M$, i.e., $f$ acts on its spectrum. The *Hermitian dilation* (Tropp, 2015) $\mathcal{H} : \mathbb{R}^{d_1 \times d_2} \to \mathbb{R}^{(d_1 + d_2) \times (d_1 + d_2)}$ is defined as

$$\mathcal{H}(A) := \begin{bmatrix} \mathbf{0}_{d_1 \times d_1} & A \\ A^\top & \mathbf{0}_{d_2 \times d_2} \end{bmatrix}. \tag{5}$$

The *influence function* (Catoni, 2012) is defined as

$$\psi(x) := \begin{cases} \log(1 + x + x^2/2), & x \geq 0, \\ -\log(1 - x + x^2/2), & x < 0. \end{cases} \tag{6}$$

We then define $\tilde{\psi}_\nu(A) := \frac{1}{\nu}\psi(\nu\mathcal{H}(A))_{\mathrm{ht}}$ for $\nu > 0$, where for $M \in \mathbb{R}^{(d_1 + d_2) \times (d_1 + d_2)}$, we define its *horizontal truncation* as $M_{\mathrm{ht}} := M_{1:d_1, d_1 + 1:d_1 + d_2}$.

**Organization.** Section 3.1 provides an overview of the algorithm, the main theorem that bounds the estimator's error guarantee and its discussions. Section 3.2 instantiates our algorithm and theorems for *adaptive* scenario by considering relevant optimal design objectives. Section 3.4 and Section 3.5 provide a proof sketch for the guarantee of Stage I and II, respectively.

### 3.1. Overview of GL-LowPopArt

We present GL-LowPopArt (Generalized Linear LOW-rank POPulation covariance regression with hARd Thresholding; Algorithm 1), a novel estimator for generalized low-rank trace regression. GL-LowPopArt consists of two stages: the first stage provides a rough, initial estimate, and the second stage refines it via matrix Catoni estimator (Minsker, 2018). It takes two designs $\pi_1$ and $\pi_2$ as inputs for Stage I and II, respectively. When the learner is in the adaptive learning scenario, she can (and will) choose $\pi_2$ dependent on the data collected during Stage I. If not, she simply inputs $\pi_1 = \pi_2 = \pi$, where $\pi$ is given to her.

Stage I uses the observations $\{(X_t, y_t)\}_{t=1}^{N_1}$ collected via $\pi_1$ to compute $\Theta_0$, the nuclear-norm regularized maximum likelihood estimator (Fan et al., 2019) (line 4). In Stage II, for each sample $(X_t, y_t)$ for $t = N_1 + 1, \cdots, N_1 + N_2$, GL-LowPopArt constructs one-sample estimator $\widetilde{\Theta}_t$ such that $\mathbb{E}[\widetilde{\Theta}_t] \approx \Theta_\star - \Theta_0$ (line 7). Then, the $\Omega$-projected matrix Catoni estimator $\Theta_1$ is computed (line 8). The final estimator $\widehat{\Theta}$ is obtained by singular value thresholding $\Theta_1$ (line 9). Note that by Assumption 1, we have $\widehat{\Theta} \in \Omega$.

We remark in advance that the final estimation error guarantee is mainly due to the use of matrix Catoni estimation (Minsker, 2018) in Stage II, yet unlike the linear trace

---

**Algorithm 1:** `GL-LowPopArt`

---

1 **Input:** Sample sizes $(N_1, N_2)$ and designs $\pi_1, \pi_2 \in \mathcal{P}(\mathcal{A})$ for Stage I and II, Regularization coefficient $\lambda_{N_1} > 0$;

   /* Stage I: Nuclear Norm-regularized Initial Estimator                   */

2 **for** $t = 1, 2, \cdots, N_1$ **do**

3      $\lfloor$ Pull $\boldsymbol{X}_t \sim \pi_1$ and receive $y_t \sim p(\cdot|\boldsymbol{X}_t; \boldsymbol{\Theta}_\star)$;

4 Compute the nuclear norm-regularized maximum likelihood estimator:

$$\boldsymbol{\Theta}_0 \leftarrow \underset{\boldsymbol{\Theta} \in \Omega}{\arg\min} \, \mathcal{L}_{N_1}(\boldsymbol{\Theta}) + \lambda_{N_1} \|\boldsymbol{\Theta}\|_*, \quad \mathcal{L}_{N_1}(\boldsymbol{\Theta}) := \frac{1}{N_1} \sum_{t=1}^{N_1} \frac{m(\langle \boldsymbol{X}_t, \boldsymbol{\Theta} \rangle) - y_t \langle \boldsymbol{X}_t, \boldsymbol{\Theta} \rangle}{g(\tau)} \tag{1}$$

   /* Stage II: Generalized Linear Matrix Catoni Estimation                       */

5 **for** $t = N_1 + 1, N_1 + 2, \cdots, N_1 + N_2$ **do**

6      Pull $\boldsymbol{X}_t \sim \pi_2$ and receive $y_t \sim p(\cdot|\boldsymbol{X}_t; \boldsymbol{\Theta}_\star)$;

7      Compute the matrix one-sample estimators:

$$\widetilde{\boldsymbol{\Theta}}_t \leftarrow \text{vec}^{-1}\left(\tilde{\boldsymbol{\theta}}_t\right), \quad \tilde{\boldsymbol{\theta}}_t \leftarrow \boldsymbol{H}(\pi_2; \boldsymbol{\Theta}_0)^{-1} \left(y_t - \mu(\langle \boldsymbol{X}_t, \boldsymbol{\Theta}_0 \rangle)\right) \text{vec}(\boldsymbol{X}_t) \tag{2}$$

8 $\boldsymbol{\Theta}_1 \leftarrow \text{Proj}_\Omega \left(\boldsymbol{\Theta}_0 + \frac{1}{N_2} \left(\sum_{t=N_1+1}^{N_1+N_2} \tilde{\psi}_\nu(\widetilde{\boldsymbol{\Theta}}_t)\right)_{\text{ht}}\right)$ with $\nu = \sqrt{\frac{2}{(1+R_s)\text{GL}(\pi_2; \boldsymbol{\Theta}_0)N_2} \log \frac{4(d_1+d_2)}{\delta}}$;

9 Let $\boldsymbol{\Theta}_1 = \boldsymbol{U}\boldsymbol{D}\boldsymbol{V}^\top$ be its SVD and $\widetilde{\boldsymbol{D}}$ be $\boldsymbol{D}$ after zeroing out singular values at most $\sqrt{\frac{8(1+R_s)\text{GL}(\pi_2; \boldsymbol{\Theta}_0)}{N_2} \log \frac{4(d_1+d_2)}{\delta}}$;

10 **Return:** $\widehat{\boldsymbol{\Theta}} := \boldsymbol{U}\widetilde{\boldsymbol{D}}\boldsymbol{V}^\top$;

---

regression (Jang et al., 2024), we *require* for the initial estimate $\boldsymbol{\Theta}_0$ to be asymptotically consistent in the rate of roughly $N_2^{-1/4}$. This was the main technical challenge for the algorithm design and analysis. We also note that Stage I only requires $\Theta(\sqrt{N_2})$ samples (ignoring other factors) for `GL-LowPopArt` to obtain the desired fast consistency rate, which is asymptotically negligible compared to $N_2$, the number of samples for the final estimator $\widehat{\boldsymbol{\Theta}}$.

We state the performance guarantee of `GL-LowPopArt`, which holds for *any* $\pi_1, \pi_2$, adaptive or nonadaptive:

**Theorem 3.1.** *Let* $\delta \in (0, 1)$. *For Stage I, set* $\lambda_{N_1} = f(\delta, d_1, d_2)\sqrt{\frac{1}{N_1}}$ *(see Lemma C.4) and*

$$N_1 \asymp \widetilde{N}_1 \vee \frac{R_s R_{\max} f(\delta, d_1, d_2)^2 r^2}{C_H(\pi_1)^2} \sqrt{\frac{(d_1 \vee d_2)N_2}{g(\tau)\kappa_\star^5 \log \frac{d}{\delta}}},$$

$$\widetilde{N}_1 \asymp \frac{r^2 R_{\max}^2}{C_H(\pi_1)^2} \left(|\text{supp}(\pi_1)| + \log \frac{1}{\delta} + \frac{R_s^2 r^2 f(\delta, d_1, d_2)^2}{C_H(\pi_1)^2}\right),$$

*with* $C_H(\pi_1) := \lambda_{\min}(\boldsymbol{H}(\pi_1; \boldsymbol{\Theta}_\star))$.

*Then,* `GL-LowPopArt` *outputs* $\widehat{\boldsymbol{\Theta}} \in \Omega$ *such that with probability at least* $1 - \delta$, $\text{rank}(\widehat{\boldsymbol{\Theta}}) \leq r$ *and*

$$\left\|\widehat{\boldsymbol{\Theta}} - \boldsymbol{\Theta}_\star\right\|_{\text{op}} \lesssim \sqrt{\frac{(1+R_s)g(\tau)\text{GL}(\pi_2)}{N_2} \log \frac{d_1 \vee d_2}{\delta}}, \tag{7}$$

*where* $\boldsymbol{\Theta}_0$ *is the initial estimator from Stage I, and*

$$\text{GL}(\pi_2) := \max\{H^{(\text{row})}(\pi_2), H^{(\text{col})}(\pi_2)\}, \tag{8}$$

*with*

$$H^{(\text{row})}(\pi_2) := \lambda_{\max}\left(\sum_{m=1}^{d_2} \boldsymbol{D}_m^{(\text{row})}(\pi_2)\right),$$

$$\boldsymbol{D}_m^{(\text{row})}(\pi_2) := [(\boldsymbol{H}(\pi_2; \boldsymbol{\Theta}_0)^{-1})_{jk}]_{j,k \in \{d_1(l-1)+m:l \in [d_2]\}},$$

$$H^{(\text{col})}(\pi_2) := \lambda_{\max}\left(\sum_{m=1}^{d_1} \boldsymbol{D}_m^{(\text{col})}(\pi_2)\right),$$

$$\boldsymbol{D}_m^{(\text{col})}(\pi_2) := [(\boldsymbol{H}(\pi_2; \boldsymbol{\Theta}_0)^{-1})_{jk}]_{j,k \in [d_1(m-1)+1:d_1 m]}.$$

*A nice illustration of* $\boldsymbol{D}_m^{(\text{row})}$ *and* $\boldsymbol{D}_m^{(\text{col})}$ *is provided in Figure 1 of Jang et al. (2024).*

**Remark 1.** *We remark that* `GL-LowPopArt` *is computationally tractable and readily implementable in practice. In Appendix J, we provide preliminary experimental results showing its efficacy, the necessity of Stage I, and more.*

$\text{GL}(\pi_2)$ captures two problem-specific characteristics: nonlinearity due to $\mu$ and the arm-set geometry of $\mathcal{A}$. The nonlinearity is captured by the use of the Hessian $\boldsymbol{H}(\pi_2; \boldsymbol{\Theta}_0)$ in the definition of $\text{GL}(\pi_2)$. Note that the "true" nonlinearity is actually $\boldsymbol{H}(\pi_2; \boldsymbol{\Theta}_\star)$, *but* given that the initial estimate $\boldsymbol{\Theta}_0$ is sufficiently close to $\boldsymbol{\Theta}_\star$, self-concordance implies that

---

**Algorithm 2: E-Carathéodory Optimal Design (ECaD)**

1   Compute $\pi_E \leftarrow \arg\max_{\pi_1 \in \mathcal{P}(\mathcal{A})} \lambda_{\min}(\boldsymbol{V}(\pi_1))$;

2   **if** $|\operatorname{supp}(\pi_E)| = \omega((d_1 d_2)^2)$ **then**

3     $\pi_{\mathrm{nuc}}^* \leftarrow \frac{1}{2(d_1 \vee d_2)}$-approximate Carathéodory solver;

4   **else**

5     $\pi_{\mathrm{nuc}}^* \leftarrow \pi_E$;

6   **Return:** $\pi_{\mathrm{nuc}}^*$;

---

$\boldsymbol{H}(\pi_2; \boldsymbol{\Theta}_0) \approx \boldsymbol{H}(\pi_2; \boldsymbol{\Theta}_\star)$ (Jun et al., 2021, Lemma 5), i.e., our design is essentially capturing the "true" nonlinearity of the problem. When $\mu(z) = z$, $\mathrm{GL}(\pi_2)$ reduces to the prior linear design objective (Jang et al., 2024, Theorem 3.4).

The intuition that $\mathrm{GL}(\pi_2)$ captures the arm-set geometry more effectively than the naïve worst-case $\frac{1}{\lambda_{\min}(\boldsymbol{H}(\pi_2; \boldsymbol{\Theta}_\star))}$ is shown in the following proposition, whose proof is deferred to Appendix E:

**Proposition 3.2.** *Suppose that $\mathcal{A} \subseteq \mathcal{B}_{\mathrm{op}}^{d_1 \times d_2}(1)$. Then, for any $\boldsymbol{\Theta}_0$ with $R_s \|\boldsymbol{\Theta}_\star - \boldsymbol{\Theta}_0\|_{\mathrm{nuc}} \leq 1$ and any $\pi \in \mathcal{P}(\mathcal{A})$,*

$$\frac{(d_1 \vee d_2)^2}{(1 + R_s)\overline{\kappa}(\pi_2; \boldsymbol{\Theta}_\star)} \leq \mathrm{GL}(\pi_2) \leq \frac{(1 + R_s)(d_1 \vee d_2)}{\lambda_{\min}(\boldsymbol{H}(\pi_2; \boldsymbol{\Theta}_\star))},$$

*where we define $\overline{\kappa}(\pi_2; \boldsymbol{\Theta}_\star) := \mathbb{E}_{\boldsymbol{X} \sim \pi_2}[\dot{\mu}(\langle \boldsymbol{X}, \boldsymbol{\Theta}_\star \rangle)]$. If $\mathcal{A} \subseteq \mathcal{B}_F^{d_1 \times d_2}(1)$, then the lower bound improves to*

$$\frac{d_1 d_2 (d_1 \vee d_2)}{(1 + R_s)\overline{\kappa}(\pi_2; \boldsymbol{\Theta}_\star)} \leq \mathrm{GL}(\pi_2).$$

Using the above proposition, we compare our result with the prior works under the assumption that $\mathcal{A} \subseteq \mathcal{B}_{\mathrm{op}}^{d_1 \times d_2}(1)$ and the GLM is 1-subGaussian. Our GL-LowPopArt achieves $\widetilde{\mathcal{O}}\left(\frac{r\mathrm{GL}(\pi_2)}{N_2}\right)$ (Theorem 3.1), while Fan et al. (2019, Theorem 1 & 2) achieve $\widetilde{\mathcal{O}}\left(\frac{r(d_1 \vee d_2)}{\lambda_{\min}(\boldsymbol{H}(\pi_2; \boldsymbol{\Theta}_\star))^2 N_2}\right)$, which is worse than ours from the above proposition. For the interest of space, we defer detailed comparison with Kang et al. (2022) to Appendix F, where we show improvements in dimension and curvature-dependent quantities. The improvement is similar in nature as to how Jang et al. (2024) improved over Koltchinskii et al. (2011) in linear trace regression.

### 3.2. Experimental Designs in the Adaptive Scenario

Theorem 3.1 induces two experimental design objectives, $C_H(\pi_1)$ and $\mathrm{GL}(\pi_2)$. Specifically, maximizing $C_H(\pi_1)$ and minimizing $|\operatorname{supp}(\pi_1)|$ results in less stringent sample size requirements for Stage I, while minimizing $\mathrm{GL}(\pi_2)$ directly minimizes the final error bound (Eqn. (7)). Because $\mathrm{GL}(\pi_2)$ depends on $\boldsymbol{\Theta}_0$ (the output of Stage I), its minimization necessitates consideration of the *adaptive scenario*.

**ECaD for Stage I.** We present **ECaD** (ee-ka-dee; Algorithm 2), an optimal design procedure for Stage I that combines E-optimal design and approximate Carathéodory solver. The outputted $\pi_{\mathrm{nuc}}^*$ is sufficiently close to the ground-truth E-optimal design while satisfying $|\operatorname{supp}(\pi_{\mathrm{nuc}}^*)| \lesssim K \wedge (d_1 d_2)^2$. We motivate the algorithm design below.

From Theorem 3.1, the straightforward design objective is as $\pi_H \leftarrow \arg\max_{\pi_1 \in \mathcal{P}(\mathcal{A})} \lambda_{\min}(\boldsymbol{H}(\pi_1; \boldsymbol{\Theta}_\star))$. However, as we do not have any prior knowledge about $\boldsymbol{\Theta}_\star$, we are forced to consider a naïve lower bound of $\lambda_{\min}(\boldsymbol{H}(\pi_1; \boldsymbol{\Theta}_\star)) \geq \kappa_\star \lambda_{\min}(\boldsymbol{V}(\pi_1))$. This motivates the following:

$$\pi_E \leftarrow \arg\max_{\pi_1 \in \mathcal{P}(\mathcal{A})} \left\{ C(\pi_1) \triangleq \lambda_{\min}(\boldsymbol{V}(\pi_1)) \right\}, \quad (9)$$

known as the *E-optimal design* (Pukelsheim, 2006), previously considered in sparse linear bandits (Hao et al., 2020) and bandit phase retrieval (Lattimore & Hao, 2021).

However, as the requirement on $N_1$ scales with $|\operatorname{supp}(\pi_1)|$, which may be quite large depending on $\mathcal{A}$, we want to minimize $|\operatorname{supp}(\pi_1)|$ as well, while retaining the E-optimality. For this, we utilize the $\epsilon$-approximate Carathéodory solver (Barman, 2015; Mirrokni et al., 2017; Combettes & Pokutta, 2023),[2][3] which outputs a $\pi_{\mathrm{nuc}}^*$ such that $\|\boldsymbol{V}(\pi_E) - \boldsymbol{V}(\pi_{\mathrm{nuc}}^*)\|_F \leq \epsilon$ and $|\operatorname{supp}(\pi_{\mathrm{nuc}}^*)| \lesssim \frac{(d_1 \wedge d_2)^2}{\epsilon^2}$.

We can control the approximation error in $C(\cdot)$ via the Hoffman-Wielandt inequality for eigenvalue perturbations (Hoffman & Wielandt, 1953), namely,

$$|C(\pi_E) - C(\pi_{\mathrm{nuc}}^*)| \leq \|\boldsymbol{V}(\pi_E) - \boldsymbol{V}(\pi_{\mathrm{nuc}}^*)\|_F \leq \epsilon.$$

As $C(\pi_E) \geq \frac{1}{d_1 \vee d_2}$ (Jang et al., 2024, Appendix D.2), it suffices to set $\epsilon = \frac{1}{2(d_1 \vee d_2)}$.

**Remark 2.** *If $\mathcal{A}$ is discrete, then one can use the polynomial-time algorithm of Allen-Zhu et al. (2021) to obtain $\pi_{\mathrm{nuc}}^*$ satisfying $|\operatorname{supp}(\pi_{\mathrm{nuc}}^*)| \lesssim d_1 d_2$ and $C(\pi_{\mathrm{nuc}}^*) \geq \frac{1}{2} C(\pi_E)$.*

**GL-Design for Stage II.** Here, we consider the optimization $\mathrm{GL}_{\min}(\mathcal{A}) := \min_{\pi_2 \in \mathcal{P}(\mathcal{A})} \mathrm{GL}(\pi_2)$. This can be efficiently solved, as $\mathrm{GL}(\pi_2)$ is convex in $\pi_2$. Implementation-wise, one can first formulate it into an epigraph form via Schur complement (Boyd & Vandenberghe, 2004) and use available convex optimization solver, e.g., CVXPY (Diamond & Boyd, 2016; Agrawal et al., 2018). For Frobenius/operator unit balls, we have the following crude upper bounds of $\mathrm{GL}_{\min}$:

**Corollary 3.3.** $\mathrm{GL}_{\min}\left(\mathcal{B}_F^{d_1 \times d_2}(1)\right) \lesssim \frac{(d_1 \vee d_2) d_1 d_2}{\kappa_\star}$ and $\mathrm{GL}_{\min}\left(\mathcal{B}_{\mathrm{op}}^{d_1 \times d_2}(1)\right) \lesssim \frac{(d_1 \vee d_2)^2}{\kappa_\star}$.

---

[2]Recently, Combettes & Pokutta (2023) showed that the Frank-Wolfe algorithm (Frank & Wolfe, 1956) is effective in solving the approximate Carathéodory problem, making it as efficient as solving the G-optimal design with bounded support (Todd, 2016).

[3]The approximate Carathéodory theorem (Barman, 2015, Theorem 2) states that $|\operatorname{supp}(\pi_{\mathrm{nuc}}^*)| \lesssim \epsilon^{-2} \operatorname{diam}(\operatorname{vec}(\mathcal{A}))^2$ where $\operatorname{vec}(\mathcal{A}) := \{\operatorname{vec}(\boldsymbol{X})\operatorname{vec}(\boldsymbol{X})^\top : \boldsymbol{X} \in \mathcal{A}\}$, and we have that $\operatorname{diam}(\operatorname{vec}(\mathcal{A}))^2 \leq 4(d_1 \wedge d_2)^2$ when $\mathcal{A} \subseteq \mathcal{B}_{\mathrm{op}}^{d_1 \times d_2}(1)$.

*Proof.* This follows directly from Proposition 3.2 and Jang et al. (2024, Appendix D) □

### 3.3. Knowledge of the GLM and Model Misspecification

Our algorithm design and analysis assume a well-specified GLM, a common assumption in the statistical and bandit literature. Addressing model misspecification typically requires fundamentally different techniques (Lattimore & Szepesvári, 2020, Chapter 24.4), as it can introduce challenges such as biased estimates and reduced efficiency; see Fortunati et al. (2017) for a survey. In particular, under misspecification, the Stage I MLE is known to converge not to the true $\boldsymbol{\Theta}_\star$, but to the KL projection of the assumed model class onto the true data-generating distribution (White, 1982). As a result, the Stage I initialization may be significantly biased, and this bias may not vanish even as $N_1$ increases. Consequently, the refined estimator from Stage II can suffer a persistent error due to this bias.

That said, our method may still tolerate mild forms of misspecification. For example, in the Gaussian case, an overestimation of the noise variance $\sigma^2$ leads to a larger choice of the regularization parameter $\lambda_{N_1}$ in Stage I, which results in a conservative but still statistically consistent estimate.[4] In such cases, the Stage I output may remain sufficiently close to $\boldsymbol{\Theta}_\star$ for Stage II to provide effective refinement.

We leave to future work exploring robustness to more general model misspecifications, or designing variants of `GL-LowPopArt` that explicitly account for GLM uncertainty – such as through Bayesian methods (Walker, 2013) or misspecification-robust estimators (Robins et al., 1994).

### 3.4. Theoretical Analysis of Stage I

**Theorem 3.4** (Guarantee for Stage I)**.** *Let* $\delta \in (0,1)$*. For Stage I, set* $\lambda_{N_1} = f(\delta, d_1, d_2)\sqrt{\frac{1}{N_1}}$ *(see Lemma C.4) and*

$$N_1 \asymp \frac{r^2 R_{\max}^2}{C_H(\pi_1)^2}\left(|\text{supp}(\pi_1)| + \log\frac{1}{\delta} + \frac{R_s^2 r^2 f(\delta, d_1, d_2)^2}{C_H(\pi_1)^2}\right),$$

*with* $C_H(\pi_1) := \lambda_{\min}(\boldsymbol{H}(\pi_1; \boldsymbol{\Theta}_\star))$*. Then, the following error bound holds with probability at least* $1 - \delta$*:*

$$\|\boldsymbol{\Theta}_0 - \boldsymbol{\Theta}_\star\|_F \lesssim \frac{f(\delta, d_1, d_2)}{C_H(\pi_1)}\sqrt{\frac{r}{N_1}}. \qquad (10)$$

*Proof Sketch.* We follow the general framework for analyzing high-dimensional M-estimators with decomposable regularizers, as established in the seminal works of Negahban & Wainwright (2011); Negahban et al. (2012); Fan et al.

---

[4]For certain applications, such as noisy matrix completion, one could utilize an alternate adaptive estimator, such as the square root LASSO-type estimator proposed in Klopp (2014, Section 4).

(2019). The proof proceeds by first establishing the Local Restricted Strong Convexity (LRSC) property of the loss function $\mathcal{L}_{N_1}$ within a nuclear norm-based constraint cone (Lemma C.2). Subsequently, leveraging a carefully chosen regularization parameter $\lambda_{N_1}$ (Lemma C.4), we derive a quadratic inequality in terms of $\|\boldsymbol{\Theta}_\star - \boldsymbol{\Theta}_0\|_F$ (proof of Theorem C.6). The complete proof is detailed in Appendix C.

We emphasize that this proof significantly improves (and arguably simplifies) upon Fan et al. (2019, Theorem 2) in the following ways:

**Relaxed Assumptions:** We do not require the crucial assumptions of Fan et al. (2019) of $\|\boldsymbol{\Theta}_\star\|_F \gtrsim \sqrt{d_1 \vee d_2}$ and $|\ddot{\mu}(z)| \leq \frac{1}{|z|}$ for $|z| > 1$ (conditions C4 and C5 in their Lemma 2). This broadens the applicability of our results, encompassing a wider range of GLMs such as Poisson.

**Improved Choice of** $\lambda_{N_1}$**:** Our Lemma C.4 introduces a novel approach for selecting $\lambda_{N_1}$ that goes beyond the double covering argument of Fan et al. (2019), which introduces a factor of $d_1 \vee d_2$. We leverage matrix Bernstein inequality (Tropp, 2015) and refined vector Hoeffding bounds for norm-sub-Gaussian and norm-sub-Poisson random vectors (Jin et al., 2019; Lee et al., 2024a). This leads to a tighter analysis for bounded GLMs, $\sigma$-subGaussian GLMs, and interestingly, enables the inclusion of Poisson distributions. Note that Fan et al. (2019) cannot cover the Poisson distribution due to their condition C5.

**Compatibility with Experimental Design:** In contrast to Fan et al. (2019), which assumes passively collected covariates $\boldsymbol{X}_t$ of bounded subGaussian norm (which they regarded as constant), our nonasymptotic analysis explicitly investigates the impact of different design $\pi_1$. □

**Remark 3.** *Our results for Stage I can be extended to the general* $\ell_q$*-constraint on the singular values of* $\boldsymbol{\Theta}_\star$ *for* $q \in [0,1)$ *as in Fan et al. (2019), and to the case where* $\Omega$ *is a smooth matrix manifold (Absil et al., 2007) using tools from manifold optimization (Boumal, 2023; Yang et al., 2014).*

### 3.5. Theoretical Analysis of Stage II – Proof Sketch of Theorem 3.1

The proof is inspired by Jang et al. (2024, Theorem 3.1), but some crucial differences make the extension non-trivial. For simplicity, let us denote $\boldsymbol{H} := \boldsymbol{H}(\pi; \boldsymbol{\Theta}_0)$ in this proof sketch with $\pi \triangleq \pi_2$, and let us ignore $\text{Proj}_\Omega$.

Recall the vectorized one-sample estimators (line 10):

$$\tilde{\boldsymbol{\theta}}_t = \boldsymbol{H}^{-1}\left(y_t - \mu(\langle \boldsymbol{X}_t, \boldsymbol{\Theta}_0\rangle)\right)\text{vec}(\boldsymbol{X}_t), \qquad (11)$$

which should satisfy $\mathbb{E}[\tilde{\boldsymbol{\theta}}_t] = \text{vec}(\boldsymbol{\Theta}_\star - \boldsymbol{\Theta}_0)$ for the matrix Catoni estimator's convergence rate (Minsker, 2018, Corollary 3.1) to be directly applicable. However, note that

$$\mathbb{E}[\tilde{\boldsymbol{\theta}}_t] = \boldsymbol{H}^{-1}\mathbb{E}_{\boldsymbol{X}\sim\pi}\left[(\mu(\langle \boldsymbol{X}, \boldsymbol{\Theta}_\star\rangle) - \mu(\langle \boldsymbol{X}, \boldsymbol{\Theta}_0\rangle))\text{vec}(\boldsymbol{X})\right].$$

When $\mu(z) = z$ as in Jang et al. (2024), above indeed reduces to $\text{vec}(\boldsymbol{\Theta}_\star - \boldsymbol{\Theta}_0)$, making $\tilde{\boldsymbol{\theta}}_t$ its unbiased estimator. When $\mu$ is nonlinear, $\tilde{\boldsymbol{\theta}}_t$ becomes *biased*.

The key technical novelty is appropriately dealing with this bias, inspired by recent progress in logistic and generalized linear bandits (Abeille et al., 2021; Jun et al., 2021; Lee et al., 2024a). Specifically, by the first-order Taylor expansion of $\mu$ with integral remainder and self-concordance (Assumption 3(b)), one can show the following (Eqn. (45) in Appendix D):

$$\left\| \mathbb{E}[\widetilde{\boldsymbol{\Theta}}_t] - (\boldsymbol{\Theta}_\star - \boldsymbol{\Theta}_0) \right\|_{\text{op}} \lesssim R_s \left\| \boldsymbol{\Theta}_\star - \boldsymbol{\Theta}_0 \right\|_{\text{nuc}}^2 \sqrt{\text{GL}(\pi)}.$$

Thus, the initial estimator $\boldsymbol{\Theta}_0$ must be asymptotically consistent at the rate of $\left\| \boldsymbol{\Theta}_\star - \boldsymbol{\Theta}_0 \right\|_{\text{nuc}} \lesssim N_2^{-1/4}$ (which requires $N_1 \gtrsim \sqrt{N_2}$) for the final error guarantee to match that of the matrix Catoni estimator. This is why we use the nuclear norm-regularized estimator in Stage I despite its sample inefficiency compared to the Catoni-style estimator. Indeed, the sample splitting approach[5] of `Warm-LowPopArt` (Jang et al., 2024, Algorithm 2) fails due to this bias.

We also remark that the experimental design objective $\text{GL}(\pi)$ arises from computing the matrix variance statistics for $\widetilde{\boldsymbol{\Theta}}_t$'s. Refer to Appendix D for the full proof. $\qquad\square$

# 4. Local Minimax Lower Bound for the Frobenius Estimation Error

In this section, we prove a *local (instance-wise)* minimax lower bound on the estimation error for generalized low-rank trace regression in the intersection of rank and nuclear norm balls. For each instance $\boldsymbol{\Theta}_\star$ with $\text{rank}(\boldsymbol{\Theta}_\star) \leq r$ and $\left\| \boldsymbol{\Theta}_\star \right\|_{\text{nuc}} \leq S_*$ for some $S_* > 0$, define its local neighborhood of radius $\varepsilon > 0$ as

$$\mathcal{N}(\boldsymbol{\Theta}_\star; \varepsilon, r, S_*) := \{ \boldsymbol{\Theta} \in \Theta(r, S_*) : \left\| \boldsymbol{\Theta} - \boldsymbol{\Theta}_\star \right\|_F \leq \varepsilon \},$$

$$\Theta(r, S_*) := \left\{ \boldsymbol{\Theta} \in \mathbb{R}^{d_1 \times d_2} : \text{rank}(\boldsymbol{\Theta}) \leq r, \left\| \boldsymbol{\Theta} \right\|_{\text{nuc}} \leq S_* \right\}.$$

$\Theta(r, S_*)$ has been considered before in the context of minimax lower bound by Rohde & Tsybakov (2011), similar to the minimax lower bound of sparse regression in the intersection of $\ell_0$ and $\ell_1$-ball constraints (Rigollet & Tsybakov, 2011, Theorem 5.3).

We now present our generic lower bound:

> **Theorem 4.1** (Local Minimax Lower Bound). *Let* $\mathcal{A} \subseteq \mathcal{B}_F^{d_1 \times d_2}(1)$ *and* $\pi \in \mathcal{P}(\mathcal{A})$. *Let* $S_* > 0, r \geq 1$ *such that* $\frac{S_*^2}{r} \geq \gamma$ *for some* $\gamma > 0$. *Also, suppose that* $N \geq \frac{R_s^2}{2^{10}} \frac{\log 2}{e} \frac{r(d_1 \vee d_2) g(\tau)}{\lambda_{\max}(\boldsymbol{H}(\pi; \boldsymbol{\Theta}_\star))}$. *Then, there exist universal constants* $C_1, C_2 = C_2(\gamma) > 0$[a] *and* $c \in$

---

> $(0, 1)$ *such that for any* $\boldsymbol{\Theta}_\star \in \Theta(r, S_*)$ *with* $\left\| \boldsymbol{\Theta}_\star \right\|_F^2 \geq \frac{9\gamma}{8}$, *there exists a small enough* $\varepsilon = \varepsilon(\boldsymbol{\Theta}_\star) > 0$ *such that the following holds:*
>
> $$\inf_{\widehat{\boldsymbol{\Theta}}} \sup_{\widetilde{\boldsymbol{\Theta}}_\star \in \mathcal{N}_\star} \mathbb{P}_{\pi, \widetilde{\boldsymbol{\Theta}}_\star} \left( E(\widehat{\boldsymbol{\Theta}}, \widetilde{\boldsymbol{\Theta}}_\star; \pi) \right) \geq c,$$
>
> $$E(\widehat{\boldsymbol{\Theta}}, \widetilde{\boldsymbol{\Theta}}_\star; \pi) := \left\{ \left\| \widehat{\boldsymbol{\Theta}} - \widetilde{\boldsymbol{\Theta}}_\star \right\|_F^2 \geq \frac{C_2 g(\tau) r(d_1 \vee d_2)}{N \lambda_{\max}(\boldsymbol{H}(\pi; \boldsymbol{\Theta}_\star)) S_*^2} \right\},$$
>
> *where* $\mathcal{N}_\star := \mathcal{N}(\boldsymbol{\Theta}_\star; \varepsilon, r, S_*)$, *and* $\mathbb{P}_{\pi, \widetilde{\boldsymbol{\Theta}}_\star}$ *is the probability measure of* $N$ *observations under* $\pi$ *and* $\widetilde{\boldsymbol{\Theta}}_\star$.
>
> [a] $C_2 = \frac{C_2' \gamma}{(1 + \sqrt{\gamma})^2}$ for an universal constant $C_2' > 0$.

*Proof Sketch.* We mainly utilize the many hypotheses technique of Tsybakov (2009, Chapter 2) for high-probability minimax lower bound; see also Yang & Barron (1999). One key technical novelty is the construction of a *local packing* $\Theta_{r, \varepsilon, \beta} \subset \Theta(r, S_*)$ around the given instance $\boldsymbol{\Theta}_\star$. Then, we carefully expand the $D_{\text{KL}}$ between two GLMs from the packing by utilizing its Bregman divergence form (Lee et al., 2024b) and self-concordance of $\mu$ (Assumption 3(b)), which leads to the instance-specific quantity $\lambda_{\max}(\boldsymbol{H}(\pi; \boldsymbol{\Theta}_\star))^{-1}$. Also, note that we don't explicitly require any restricted isometry assumption (Koltchinskii et al., 2011, Eqn. (2.4)). Refer to Appendix G for the full proof.

This significantly deviates from Rohde & Tsybakov (2011, Theorem 5), where they considered a packing around $\boldsymbol{\Theta}_\star = \boldsymbol{0}$ for linear trace regression. This still resulted in a tight lower bound, as when $\mu(z) = z$, the problem difficulty becomes uniform across all $\boldsymbol{\Theta}_\star \in \Theta(r, S_*)$. $\qquad\square$

**Instance-Specific Nature.** Our lower bound explicitly depends on the "optimistic" instance-specific curvature, $\lambda_{\max}(\boldsymbol{H}(\pi; \boldsymbol{\Theta}_\star))^{-1}$, thereby capturing the inherent variation in problem difficulty across different problem instances characterized by $\boldsymbol{\Theta}_\star$. To the best of our knowledge, this is the first time such an instance-wise dependency has been captured in the context of (generalized linear) trace regression and matrix completion. This behavior mirrors the local minimax lower bounds established for logistic bandits (Abeille et al., 2021, Theorem 2) and online LQR (Simchowitz & Foster, 2020, Theorem 1), which also account for instance-specific complexities. This contrasts with the worst-case minimax lower bounds (Koltchinskii et al., 2011; Rohde & Tsybakov, 2011; Davenport et al., 2014; Lafond, 2015; Taki et al., 2021), which cannot capture such instance-specific dependencies.

**Near Instance-wise Optimality.** Comparing our lower bound with the performance guarantee of `GL-LowPopArt` (Theorem 3.1), one can see that *for each fixed, nonrandom design* $\pi_2$, *the gap between the upper and lower bounds*

on the squared Frobenius error is $\mathrm{GL}(\pi_2)\lambda_{\max}(\pi_2; \boldsymbol{\Theta}_\star) \leq \frac{\lambda_{\max}(\pi_2;\boldsymbol{\Theta}_\star)}{\lambda_{\min}(\pi_2;\boldsymbol{\Theta}_\star)}$ (Proposition 3.2), i.e., at most the Hessian's condition number. Thus, GL-LowPopArt is nearly instance-wise optimal in the passive scenario where $\pi_1 = \pi_2$ is fixed in advance. A subtle but important point is that if $\pi_2$ is chosen using information gathered from Stage I (e.g., through experimental design as described in Section 3.2), then the upper bound is achieved via an *adaptive* procedure. However, our lower bound does not apply in this case, as it assumes i.i.d. samples drawn from a single fixed design. Extending our lower bound to the adaptive setting – analogous to the regret lower bounds in bandits (Lattimore & Szepesvári, 2020) –is an interesting future direction.

This stands in contrast to the nuclear norm-regularized estimator, which achieves at best a rate of $\widetilde{\mathcal{O}}\left(\frac{(d_1 \vee d_2)d_1 d_2 r}{\kappa^2 \lambda_{\min}(\boldsymbol{V}(\pi_2))N}\right)$ when using i.i.d. samples from $\pi_2$ (see Theorem 3.4 and Appendix F); note the additional factor of $1/\kappa$, which corresponds to the worst-case curvature. As a result, although the nuclear norm-regularized estimator is nearly instance-wise optimal in the linear setting (Rohde & Tsybakov, 2011; Koltchinskii et al., 2011), it fails to achieve such optimality in the nonlinear GLM case. This underscores the strength of our method, GL-LowPopArt, which is nearly instance-wise optimal across all GLMs satisfying Assumption 3.

**Requirement on $N$.** A keen reader may observe that our local minimax lower bound holds under the condition $N \gtrsim \frac{R_s^2 r(d_1 \vee d_2)}{\lambda_{\max}(\boldsymbol{H}(\pi;\boldsymbol{\Theta}_\star))}$. We emphasize that this requirement is not restrictive and actually provides an intuitive justification for Stage I as a warm-up phase; in fact, we believe that some condition of this form on $N$ is necessary—although we do not currently have a formal proof. The requirement on $N$ arises when bounding the KL divergence between the true model $\boldsymbol{\Theta}_\star$ and an alternative model from the constructed local packing. Intuitively, this stems from the necessity for the two models to be sufficiently close for self-concordance properties to take effect; this was also the case for prior local minimax lower bounds (Abeille et al., 2021, Theorem 2) (Simchowitz & Foster, 2020, Theorem 1), where the requirement on horizon length $T$ arises in a similar fashion. Finally, we point out that in the linear setting (i.e., $\mu(z) = z \Rightarrow R_s = 0$), our requirement on $N$ vanishes.

# 5. Applications of GL-LowPopArt

Here, we describe two applications of GL-LowPopArt. For the interest of space, we defer detailed discussions to the Appendix, and focus on the main results and intuitions.

## 5.1. Generalized Linear Matrix Completion under USR

In *generalized linear matrix completion under uniform sampling at random (USR)*, we assume $\mathcal{A} = \mathcal{X} =$ $\{\boldsymbol{e}_i(\boldsymbol{e}_j')^\top : (i,j) \in [d_1] \times [d_2]\}$, $\pi^U = \mathrm{Unif}(\mathcal{A})$, and $\max_{i,j} |(\boldsymbol{\Theta}_\star)_{i,j}| \leq \gamma$ for a $\gamma > 0$. Here, we focus on the *1-bit matrix completion* (Davenport et al., 2014) with $\mu(z) = (1 + e^{-z})^{-1}$ for simple calculations, although we emphasize that similar arguments can be made for generic (self-concordant) GLMs. Let us denote $\mathcal{E}_F := \left\|\widehat{\boldsymbol{\Theta}} - \boldsymbol{\Theta}_\star\right\|_F^2$.

We first compare the error bound of GL-LowPopArt (in passive scenario with $\pi_1 = \pi_2 = \pi^U$) with Davenport et al. (2014, Theorem 1) and Klopp et al. (2015, Corollary 2):

$$\mathcal{E}_F \lesssim \frac{1}{\min_{i,j} \dot{\mu}((\boldsymbol{\Theta}_\star)_{ij})} \frac{rd_1 d_2(d_1 \vee d_2)}{N}, \qquad \text{(ours)}$$

$$\mathcal{E}_F \lesssim \frac{1}{\min_{|z| \leq \gamma} \dot{\mu}(z)} \sqrt{\frac{r(d_1 d_2)^2(d_1 \vee d_2)}{N}}, \quad \text{(Davenport)}$$

$$\mathcal{E}_F \lesssim \left(\frac{1}{\min_{|z| \leq \gamma} \dot{\mu}(z)}\right)^2 \frac{rd_1 d_2(d_1 \vee d_2)}{N}. \qquad \text{(Klopp)}$$

Our bound obtains the known minimax optimal rate of $\frac{rd_1 d_2(d_1 \vee d_2)}{N}$, *and* captures the instance-specific difficulty via $\frac{1}{\min_{i,j} \dot{\mu}((\boldsymbol{\Theta}_\star)_{ij})}$. On the other hand, the other bounds depend on the worst-case curvature $\frac{1}{\min_{|z| \leq \gamma} \dot{\mu}(z)}$. In other words, if the current instance $\boldsymbol{\Theta}_\star$ is such that $\min_{i,j} \dot{\mu}((\boldsymbol{\Theta}_\star)_{ij}) \gg \min_{|z| \leq \gamma} \dot{\mu}(z)$, then the gap between our bound and theirs becomes larger.

Algorithm-wise, Davenport et al. (2014); Klopp et al. (2015), along with other approaches (Srebro & Salakhutdinov, 2010; Cai & Zhou, 2013; 2016; Lafond, 2015), requires the knowledge of $\gamma > 0$, to compute the nuclear-norm regularized estimator *with* the constraint of $\|\boldsymbol{\Theta}\|_\infty \leq \gamma$ or $\|\boldsymbol{\Theta}\|_{\max} \leq \gamma$. Interestingly, GL-LowPopArt does *not* require any knowledge about $\boldsymbol{\Theta}_\star$, yet it fully adapts to the given instance.

**Remark 4** (Comparing to BMF). *While the Burer-Monteiro Factorization (BMF) is a popular optimization-based approach to matrix completion, one cannot directly compare our work to BMF; see Appendix A.*

## 5.2. Bilinear Dueling Bandits

### 5.2.1. PROBLEM DESCRIPTION

In **bilinear dueling bandits**, let $\mathcal{A} \subseteq \mathcal{B}^d(1)$ be the given vector-valued arm-set satisfying the following:

**Assumption 4.** $\mathrm{span}(\mathcal{A}) = \mathbb{R}^d$, and $\mathcal{A}$ is compact.

At each timestep $t$, the learner chooses a pair of arms $(\boldsymbol{\phi}_{w,t}, \boldsymbol{\phi}_{l,t}) \in \mathcal{A} \times \mathcal{A}$, and receives a feedback sampled from the following generalized bilinear form:

$$o_t = \mathbb{1}[\boldsymbol{\phi}_{w,t} \succ \boldsymbol{\phi}_{l,t}] \sim \mathrm{Ber}(\mu(\boldsymbol{\phi}_{w,t}^\top \boldsymbol{\Theta}_\star \boldsymbol{\phi}_{l,t})), \quad (12)$$

for an *unknown*, skew-symmetric $\boldsymbol{\Theta}_\star$ of rank $2r$, and a *known* comparison function $\mu : \mathbb{R} \to [0,1]$. $\mathcal{A}$ may be infinite as in continuous dueling bandits (Kumagai, 2017).

---

**Algorithm 3:** `BETC-GLM-LR`

1 **for** $t = 1, 2, \cdots, N_1 + N_2$ **do**
2     Run `GL-LowPopArt`$(N_1, N_2)$ and obtain $\widehat{\Theta}$;
3 Obtain the estimated Borda winner:

$$\hat{\phi} \leftarrow \arg\max_{\phi \in \mathcal{A}} \left\{ \widehat{B}(\phi) \triangleq \mathbb{E}_{\phi' \sim \text{Unif}(\mathcal{A})} \left[ \mu\left(\phi^\top \widehat{\Theta} \phi'\right) \right] \right\}$$

4 **for** $t = N_1 + N_2 + 1, \cdots, T$ **do**
5     Pull $(\hat{\phi}, \hat{\phi})$;

---

We assume that $\mu$ satisfies the following (Wu et al., 2024):

**Assumption 5.** In addition to Assumption 3, $\mu : \mathbb{R} \to [0, 1]$ satisfies $\mu(z) + \mu(-z) = 1, \ z \in \mathbb{R}$.

Some examples of $\mu$ that satisfies the above include $\mu(z) = \frac{1+z}{2}$ and $\mu(z) = (1 + e^{-z})^{-1}$. Note that when $\mu(z) = (1 + e^{-z})^{-1}$, our model precisely becomes to Bernoulli.

The learner's goal is to minimize the Borda regret (Saha et al., 2021):

$$\text{Reg}^B(T) := \sum_{t=1}^{T} \left\{ B(\phi_\star) - \frac{B(\phi_{w,t}) + B(\phi_{l,t})}{2} \right\},$$

where

$$B(\phi) := \mathbb{E}_{\phi' \sim \text{Unif}(\mathcal{A})}[\mu(\phi^\top \Theta \phi')] \tag{13}$$

is the *(shifted) Borda score* of arm $\phi \in \mathcal{A}$, and $\phi_\star = \arg\max_{\phi \in \mathcal{A}} B(\phi)$ is the *Borda winner*. Note that when $\mathcal{A}$ is finite, it reduces to the usual definition of Borda regret/winner in the finite-armed dueling bandits (Jamieson et al., 2015; Saha et al., 2021). Unlike the Condorcet winner, the Borda winner always exists for any preference model (Bengs et al., 2021).

**Remark 5** (Significance of the Setting). *We emphasize that this is a **novel** dueling bandits setting not considered before. This is motivated by recent progress in general preference learning in RLHF, specifically Zhang et al. (2024b) where the authors have proposed Eqn. (12) that can express non-transitive preferences from item-wise features. We defer further discussions on the proposed setting, including its motivation, to Appendix H.*

Lastly, we introduce the following quantities, which are assumed to be strictly positive: denoting $\mathcal{U} := \text{Unif}$,

$$\kappa_\star := \min_{\phi, \phi' \in \mathcal{A}} \dot{\mu}\left(\phi^\top \Theta_\star \phi'\right), \ \kappa_\star^B := \mathbb{E}_{\phi' \sim \mathcal{U}(\mathcal{A})}[\dot{\mu}(\phi_\star^\top \Theta \phi')].$$

5.2.2. `BETC-GLM-LR` AND REGRET UPPER BOUND

We consider an explore-then-commit approach, where the exploration is done via our `GL-LowPopArt`. The full pseu-

docode is provided in Algorithm 3. It attains the following Borda regret bound:

**Theorem 5.1** (Informal). *With appropriate choices of $N_1$ and $N_2$ in `GL-LowPopArt` and large enough $T$, `BETC-GLM-LR` attains the following Borda regret bound with probability at least $1 - \delta$:*

$$\text{Reg}^B(T) \lesssim \left(\text{GL}_{\min}(\mathcal{A}) \log \frac{d}{\delta}\right)^{1/3} \left(\kappa_\star^B T\right)^{2/3}. \tag{14}$$

*Proof Sketch.* We deviate significantly from Wu et al. (2024) by using the self-concordance of $\mu$ as in Abeille et al. (2021, Theorem 1), allowing for the regret bound to scale *with* $\kappa_\star^B$. Refer to Appendix I.1 for the full proof. $\square$

Two quantities make our regret bound truly instance-specific. One is $\text{GL}_{\min}(\mathcal{A})$, which, as discussed previously, captures the geometry of $\mathcal{A}$ as well as the associated nonlinearity via the Hessian. In addition, the regret bound scales with $\kappa_\star^B$, the averaged curvature "centered" around the Borda winner, analogous to logistic and generalized linear bandits (Abeille et al., 2021; Liu et al., 2024; Lee et al., 2024a).

We believe $T^{2/3}$ dependency of the Borda regret is unavoidable. This stems from the fact that more general dueling bandit settings have shown $\Omega(T^{2/3})$ Borda regret lower bounds (omitting other dependencies) (Saha et al., 2021, Theorem 16) (Wu et al., 2024, Theorem 4.1). This naturally motivates our choice of the explore-then-commit (ETC) approach. Furthermore, our estimation procedure is not anytime-valid, making ETC an ideal choice for integrating our estimator within the bandit framework. We defer a more in-depth comparison with Wu et al. (2024) to Appendix I.2.

## 6. Conclusion and Future Work

This work addresses the critical gap in prior work by explicitly considering instance-specific curvature in generalized low-rank trace regression. We introduce `GL-LowPopArt`, a novel estimator that achieves state-of-the-art performance, adapting to both the nonlinearity of the model and the underlying arm-set geometry. We establish the first instance-wise minimax lower bound, demonstrating the near-optimality of `GL-LowPopArt`. We showcase its benefits through applications to generalized linear matrix completion and bilinear dueling bandits, a novel setting of independent interest for general preference learning (Zhang et al., 2024b).

Other than the future directions mentioned in the main text, another is deriving an instance-wise improved estimator for other structures, such as row (column)-wise sparsity (Zhao & Leng, 2014) or even their superposition (Yang & Ravikumar, 2013; Oymak et al., 2015; Richard et al., 2012; Zhao et al., 2017). A promising starting point for this is to extend `PopArt` (Jang et al., 2022) to the sparse trace regression.

## Acknowledgements

J. Lee thanks Hanseul Cho for reviewing the initial manuscript and providing valuable feedback on LaTeX typographical corrections. J. Lee also thanks Minchan Jeong for the initial discussions of this project regarding the 2nd-order tensor product spaces.

J. Lee and S.-Y. Yun were supported by the Institute of Information & Communications Technology Planning & Evaluation (IITP) grant funded by the Korea government(MSIT) (No. RS-2022-II220311, Development of Goal-Oriented Reinforcement Learning Techniques for Contact-Rich Robotic Manipulation of Everyday Objects, No. RS-2024-00457882, AI Research Hub Project, and No. RS-2019-II190075, Artificial Intelligence Graduate School Program (KAIST)). K. Jang was supported by the Institute of Information & Communications Technology Planning & Evaluation (IITP) grant funded by the Korea government (MSIT) [RS-2021-II211341, Artificial Intelligence Graduate School Program (Chung-Ang University)]. K.-S. Jun was supported in part by the National Science Foundation under grant CCF-2327013 and Meta Platforms, Inc.

## Impact Statement

This paper presents work whose goal is to advance the field of Machine Learning. There are many potential societal consequences of our work, none which we feel must be specifically highlighted here.

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

# A. Related Works

**Generalized Linear Matrix Completion.**    This has been extensively studied in the early 2010s under various noise assumptions: Gaussian (Rohde & Tsybakov, 2011; Koltchinskii et al., 2011), Bernoulli (Alquier et al., 2019), multinomial (Lafond et al., 2014; Klopp et al., 2015), general exponential family (Lafond, 2015), and even with the only assumption of bounded variance (Klopp, 2014). We refer interested readers to Davenport & Romberg (2016) for an overview of works on matrix completion. Note that our model implicitly implies that for each $(i,j) \in [d_1] \times [d_2]$ may be observed multiple times, which is often the case in recommender systems and bandits where the same item can be recommended multiple times for exploration, or it may be that "users are more active than others and popular items are rated more frequently." (Klopp et al., 2015). On a slightly different note, many works have explored the same setting under the assumption that each entry of $\Theta_\star$ can be sampled at most once (Candès & Plan, 2010; Cai & Zhou, 2013; Davenport et al., 2014; Gunasekar et al., 2014; Cao & Xie, 2016; Alaya & Klopp, 2019; McRae & Davenport, 2020). When $\Theta_\star$ is additionally is skew-symmetric ($\Theta_\star^\top = -\Theta_\star$), this is also related to learning the low-rank preference model (Gleich & Lim, 2011; Lu & Negahban, 2015; Rajkumar & Agarwal, 2016; Wu et al., 2024; Zhang et al., 2024b).

**Burer-Monteiro Factorization**    The Burer–Monteiro factorization (BMF, Burer & Monteiro (2003; 2005)) approach has been extensively studied for noiseless low-rank matrix recovery from deterministic linear measurements (Candès & Recht, 2009; Candès & Plan, 2011), primarily from an optimization perspective (Bi et al., 2022; Ge et al., 2017; Park et al., 2017; Zhang et al., 2024a; Boumal et al., 2016; Yalçın et al., 2022; Bhojanapalli et al., 2016; Stöger & Soltanolkotabi, 2021; Kim & Chung, 2023). In contrast, our work focuses on noisy matrix completion under a generalized linear model (GLM) framework, aiming to achieve accurate estimation with high probability as the sample size increases. This fundamental difference in problem settings implies that the optimization complexity measures used to analyze BMF methods, such as the optimization complexity metric (OCM) introduced by Yalçın et al. (2022) and Zhang et al. (2024a), are not directly comparable to our statistical analysis. Specifically, their OCM quantifies the non-convexity of the BMF landscape, which is related to the success of local search methods (e.g., gradient descent), while our "statistical complexity metric", arguably $\lambda_{\max}(\boldsymbol{H}(\pi; \Theta_\star))$ that pops up in our lower bound (Theorem 4.1), is information-theoretic and dictates the minimum sample size required for any estimator to obtain a desired accuracy with high probability.

While BMF methods offer computational efficiency and have been shown to perform well empirically, especially in large-scale problems, they all rely on some non-convex optimization, whose landscape is not always guaranteed to be benign, especially in the presence of noise (Ma & Fattahi, 2023). Our GL-LowPopArt only involves convex optimization subroutines and thus is computationally tractable, but inefficient: for instance, GL-LowPopArt requires computing the SVD and inverting $d^2 \times d^2$ matrices. Therefore, while BMF and our work both address low-rank matrix recovery, their respective advantages depend on the specific problem context.

**Low-Rank Matrix Bandits.**    Researchers in low-rank bandits have long focused on fundamental and specific models. For example, Katariya et al. (2017a;b); Trinh et al. (2020); Jedra et al. (2024); Sentenac et al. (2021) studied a bilinear bandit setting (which means $\mathcal{A} = \{xz^\top : x \in \mathcal{X} \subset \mathbb{R}^{d_1}, z \in \mathcal{Z} \subset \mathbb{R}^{d_2}\}$) with canonical basis ($\mathcal{X} = \{e_i : i \in [d_1]$ and $\mathcal{Z} = \{e_j : j \in [d_2]\}$). Katariya et al. (2017a;b); Trinh et al. (2020); Sentenac et al. (2021) added an assumption that $\text{rank}(\Theta_*) = 1$ over a bilinear bandit setting. Stojanovic et al. (2023) presents an entry-wise matrix estimation for low-rank reinforcement learning, including low-rank bandits. Another popular assumption on arm sets in low-rank bandits is a unit ball (or a unit sphere) assumption (Kotłowski & Neu, 2019; Lattimore & Hao, 2021; Huang et al., 2021). For bilinear bandits, Kotłowski & Neu (2019) assumed that $\mathcal{A} = \{\boldsymbol{x}\boldsymbol{x}^\top : \boldsymbol{x} \in \mathbb{S}^{d-1}\}$ and $\Theta_\star$ should be also symmetric. (Lattimore & Hao, 2021) even added an assumption that $\Theta_*$ is a symmetric rank-1 matrix. For low-rank bandits, Huang et al. (2021) assumed $\mathcal{A} = \mathcal{B}_F^{d \times d}$. These tailored algorithms often outperform general approaches significantly, yet extending these algorithms to other settings has generally proven challenging due to the highly specialized nature of their settings.

The first study on low-rank bandits with general arm sets is Jun et al. (2019). This work introduced the first general bilinear low-rank linear bandit algorithm that could be applied flexibly to any $d$-dimensional arm set $\mathcal{X}$ and $\mathcal{Z}$. Subsequently, Lu et al. (2021) extended this approach beyond bilinear settings, proposing a generalized low-rank linear bandit algorithm applicable to all matrix arm sets. Later, Kang et al. (2022) introduced a novel method leveraging Stein's method, and Li et al. (2022) developed a general framework for high-dimensional linear bandits, including low-rank bandits. However, none of these studies explicitly addressed experimental design; rather, they handled the issue of experimental designs by assuming that their arm sets are sufficiently well-distributed in all directions. As a result, they failed to fully capture how the regret bound varies with the geometry of the arm set. For example, (Jun et al., 2019) and (Lu et al., 2021) conjectured

that the lower bound for the bilinear low-rank bandit problem should be $\Omega(\sqrt{rd^3T})$, based on results from trace regression. However, Jang et al. (2021) later demonstrated that by considering the structure of the arm set in the bilinear setting, this bound could be further improved, highlighting the importance of optimal design tailored to the arm set. In Appendix F, we thoroughly compare our results with Kang et al. (2022).

Recent work by Jang et al. (2024) systematically addresses arm set geometry and experimental design in the low-rank linear bandits. This work applied thresholding at the subspace level called `LowPopArt` and proposed a novel experimental design for this new regression method. They then analyzed the experimental design assumptions underlying previous studies and successfully proved that their `LowPopArt` with their experimental design outperforms the previous works, even order-wise improvements in some cases. Our paper further extends the `LowPopArt` to the generalized linear scenario and provides performance guarantees in both upper and lower bounds that are nearly optimal even in terms of instance-specific, curvature-dependent quantities.

**Generalized Linear Bandits (GLBs).** **GLB** is a natural nonlinear extension of linear bandits, first proposed by Filippi et al. (2010), and later studied by much works (Lee et al., 2024a; Sawarni et al., 2024; Jun et al., 2017; Li et al., 2017). **GLB**s encompass a wide range of bandits, including linear, logistic, Poisson, logit, and more. Out of these, especially **logistic bandits (LogB)** (Faury et al., 2020; 2022; Mason et al., 2022; Abeille et al., 2021; Lee et al., 2024b) has garnered much attention, as it can naturally model binary feedback ('click' or 'no click'; Li et al. (2012)). Also, owing to its similarity to the Bradley-Terry model-based RLHF, the confidence sets of logistic bandits have been used for quantifying the uncertainty of the linear reward model (Das et al., 2024; Xiong et al., 2024; Zhong et al., 2024). In **GLB**s, the key quantity describing the problem difficulty is[6] $\kappa_\star^{-1} := \dot{\mu}(\langle \boldsymbol{x}_\star, \boldsymbol{\theta}_\star \rangle)$, where $\boldsymbol{\theta}_\star$ is the unknown vector and $\boldsymbol{x}_\star$ is the optimal arm vector. (Abeille et al., 2021) showed a regret lower bound of $\Omega(d\sqrt{T\kappa_\star})$ for **LogB**s, which was matched by various UCB-type algorithms (Abeille et al., 2021; Faury et al., 2022; Lee et al., 2024b). Despite the lack of a generic lower bound for general **GLB**s, recent breakthroughs (Sawarni et al., 2024; Lee et al., 2024a; Liu et al., 2024) showed that for self-concordant **GLB**s, regret upper bound of $\tilde{\mathcal{O}}(d\sqrt{T\kappa_\star})$ can be attained.

**Remark 6.** *In the optimization literature, the original definition of the self-concordance takes the form of $|\dddot{\mu}(z)| \leq 2\ddot{\mu}(z)^{3/2}$ $\forall z \in \mathbb{R}$, originally motivated for convergence analysis of Newton's method by Nesterov (1988). Bach (2010) was the first to adapt the concept to extend the M-estimator results of squared loss to logistic loss. Later, people from the bandit community further adapted it for logistic and generalized linear bandits (Faury et al., 2020; Abeille et al., 2021; Russac et al., 2021), which is the form we consider here (Assumption 3(b))*

---

[6]In the mentioned literature, the quantity is denoted as $\kappa_\star$. To keep our notation consistent with the dueling bandits' literature, we chose to denote this as $\kappa_\star^{-1}$.

# B. Notation Table

Table 1. Summary of notation used in this paper.

| Notation | Description |
|---|---|
| $\|\cdot\|_{\mathrm{nuc}}$ | Nuclear norm |
| $\|\cdot\|_{\mathrm{op}}$ | Operator (spectral) norm |
| $\langle \boldsymbol{A}, \boldsymbol{B} \rangle$ for $\boldsymbol{A}, \boldsymbol{B} \in \mathbb{R}^{m \times n}$ | $\mathrm{tr}(\boldsymbol{A}^\top \boldsymbol{B})$ |
| $\lambda_i(\boldsymbol{A})$ | The $i$-th largest eigenvalue of a symmetric matrix $\boldsymbol{A}$ |
| $\lambda_{\max}$ | The largest eigenvalue, same as $\lambda_1$ |
| $\lambda_{\min}$ | The smallest eigenvalue, same as $\lambda_m$ |
| $\mathcal{B}_i^{d_1 \times d_2}(S)$ for $i \in \{\mathrm{op}, \mathrm{nuc}, F\}$ | $\{\boldsymbol{X} \in \mathbb{R}^{d_1 \times d_2} : \|\boldsymbol{X}\|_i \leq S\}$ |
| $\mathrm{vec} : \mathbb{R}^{d_1 \times d_2} \to \mathbb{R}^{d_1 d_2}$ | Column-wise stacking operation of a matrix into a vector |
| $\mathrm{vec}^{-1} : \mathbb{R}^{d_1 d_2} \to \mathbb{R}^{d_1 \times d_2}$ | Reshape operation of a vector to a matrix |
| $[n]$ for $n \in \mathbb{N}$ | $\{1, 2, \ldots, n\}$ |
| $\mathcal{P}(X)$ | The set of all probability distributions on $X$ |
| $\Omega$ | Parameter space |
| $\boldsymbol{\Theta}_\star \in \mathbb{R}^{d_1 \times d_2}$ | An unknown reward matrix of rank at most $r \ll d_1 \wedge d_2$ |
| $\mathcal{A} \subseteq \mathbb{R}^{d_1 \times d_2}$ | Arm-set (e.g., sensing matrices). |
| $p(y|\boldsymbol{X}; \boldsymbol{\Theta}_\star)$ | Probability density function of the generalized linear model of the reward $y$ when $X$ is chosen by the learner, $\propto \exp\left( \frac{y\langle \boldsymbol{X}, \boldsymbol{\Theta}_\star \rangle - m(\langle \boldsymbol{X}, \boldsymbol{\Theta}_\star \rangle)}{g(\tau)} \right)$ |
| $m : \mathbb{R} \to \mathbb{R}$ | log-partition function of GLM |
| $\tau$ | Dispersion parameter |
| $\mu$ | $\dot{m}$, Inverse link function. |
| $\pi \in \mathcal{P}(\mathcal{A})$ | Sampling policy (design) |
| $\boldsymbol{V}(\pi)$ | Design matrix, $\mathbb{E}_{\boldsymbol{X} \sim \pi}[\mathrm{vec}(\boldsymbol{X})\mathrm{vec}(\boldsymbol{X})^\top]$ |
| $\boldsymbol{H}(\pi; \boldsymbol{\Theta})$ | Hessian matrix $\mathbb{E}_{\boldsymbol{X} \sim \pi}[\dot{\mu}(\langle \boldsymbol{X}, \boldsymbol{\Theta} \rangle)\mathrm{vec}(\boldsymbol{X})\mathrm{vec}(\boldsymbol{X})^\top]$ |
| $R_{\max}, R_s, \kappa_*$ | Parameters on $\mu$, check Assumption 3 |
| $\mathcal{H}$ | Hermitian Dilation (Check Eq. (5)) |
| $\psi$ | Influence function (Check Eq. (6) |
| $\tilde{\psi}_\nu(\boldsymbol{A})$ | $\frac{1}{\nu}\psi(\nu\mathcal{H}(\boldsymbol{A}))_{\mathrm{ht}}$, where for $\boldsymbol{M} \in \mathbb{R}^{(d_1+d_2) \times (d_1+d_2)}$, $\boldsymbol{M}_{\mathrm{ht}} := \boldsymbol{M}_{1:d_1, d_1+1:d_1+d_2}$ |
| $\mathrm{GL}(\pi)$ | Our new experimental design objective (See Eq. (8) |
| $\overline{\kappa}(\pi; \boldsymbol{\Theta})$ | $\mathbb{E}_{\boldsymbol{X} \sim \pi}[\dot{\mu}(\langle \boldsymbol{X}, \boldsymbol{\Theta} \rangle)]$ |

# C. Proof of Theorem 3.4 – Error Bound of Stage I

In this Appendix, let us denote $N = N_1$ for notational simplicity, and we introduce the following notations:

$$\mathcal{L}_N(\boldsymbol{\Theta}) := \frac{1}{N} \sum_{t=1}^{N} \frac{m(\langle \boldsymbol{X}_t, \boldsymbol{\Theta} \rangle) - y_t \langle \boldsymbol{X}_t, \boldsymbol{\Theta} \rangle}{g(\tau)} \tag{15}$$

$$\boldsymbol{\Theta}_0 := \arg\min_{\boldsymbol{\Theta} \in \Omega} \left\{ \mathcal{L}_N(\boldsymbol{\Theta}) + \lambda_N \|\boldsymbol{\Theta}\|_* \right\} \tag{16}$$

$$\boldsymbol{H}(\pi; \boldsymbol{\Theta}) := \mathbb{E}_{\boldsymbol{X} \sim \pi} \left[ \dot{\mu}(\langle \boldsymbol{X}, \boldsymbol{\Theta} \rangle) \mathrm{vec}(\boldsymbol{X}) \mathrm{vec}(\boldsymbol{X})^\top \right]. \tag{17}$$

## C.1. Definition of RSC and Constraint Cone $\mathcal{C}$

We first recall the definition of local restricted strong convexity (LRSC) (Negahban & Wainwright, 2011; Negahban et al., 2012; Fan et al., 2018; 2019):

**Definition C.1.** Let $\boldsymbol{\Theta}_\star \in \Omega \subseteq \mathbb{R}^{d_1 \times d_2}$ be the ground-truth parameter of rank $r \leq d_1 \wedge d_2$, and let us denote $\mathcal{B}_F^{d_1 \times d_2}(W) := \{\boldsymbol{\Theta} \in \mathbb{R}^{d_1 \times d_2} : \|\boldsymbol{\Theta}\|_F \leq W\}$. Let $\mathcal{C} \subseteq \mathbb{R}^{d_1 \times d_2}$ be a constraint cone, $W, \xi > 0$ and $\tau \geq 0$. A loss function $\mathcal{L}(\cdot)$ satisfies $\mathrm{LRSC}(\mathcal{C}, W, \xi, \tau)$ at $\boldsymbol{\Theta}_\star$ if the following holds:

$$B_{\mathcal{L}}^s(\boldsymbol{\Theta}_\star + \Delta, \boldsymbol{\Theta}_\star) \triangleq \frac{1}{2} \langle \nabla \mathcal{L}(\boldsymbol{\Theta}_\star + \Delta) - \nabla \mathcal{L}(\boldsymbol{\Theta}_\star), \Delta \rangle \geq \xi \|\Delta\|_F^2 - \tau, \quad \forall \Delta \in \mathcal{C} \cap \mathcal{B}_F^{d_1 \times d_2}(W), \tag{18}$$

where $B_{\mathcal{L}}^s(\cdot, \cdot)$ is the symmetric Bregman divergence induced by $\mathcal{L}$.

**Remark 7.** *The "original" definition of LRSC is in terms of the unsymmetric Bregman divergence and must hold for all points near $\boldsymbol{\Theta}_\star$, namely, for some neighborhood $\mathcal{N}$ of $\boldsymbol{\Theta}_\star$,*

$$B_{\mathcal{L}}(\boldsymbol{\Theta} + \Delta, \boldsymbol{\Theta}) \triangleq \mathcal{L}(\boldsymbol{\Theta} + \Delta) - \mathcal{L}(\boldsymbol{\Theta}) - \langle \nabla \mathcal{L}(\boldsymbol{\Theta}), \Delta \rangle \geq \xi \|\Delta\|_F^2 - \tau, \quad \forall \Delta \in \mathcal{C}, \forall \boldsymbol{\Theta} \in \mathcal{N}. \tag{19}$$

*As one can see later, we only require the symmetric version for the final proof, and we only need the above to hold for $\boldsymbol{\Theta} = \boldsymbol{\Theta}_\star$. Indeed, this is also the case in the proof of Theorem 1 of Fan et al. (2019).*

We follow the proof strategy for Lemma 1 of Negahban & Wainwright (2011), part of which dates back to Recht et al. (2010). Let $\boldsymbol{\Theta}_\star = \boldsymbol{U} \boldsymbol{D} \boldsymbol{V}^\top$ be its SVD, $\boldsymbol{U}_r$ be the first $r$ columns of $\boldsymbol{U}$, and $\boldsymbol{U}_r^\perp$ be the remaining columns. We define $\boldsymbol{V}_r$ and $\boldsymbol{V}_r^\perp$ analogously. Note that as $\mathrm{rank}(\boldsymbol{\Theta}_\star) = r$, the singular values corresponding to $\boldsymbol{U}_r^\perp$ and $\boldsymbol{V}_r^\perp$ are zero. Define the two subspaces

$$\mathcal{M} := \left\{ \boldsymbol{\Theta} \in \mathbb{R}^{d_1 \times d_2} : \mathrm{row}(\boldsymbol{\Theta}) \subseteq \mathrm{row}(\boldsymbol{V}_r), \mathrm{col}(\boldsymbol{\Theta}) \subseteq \mathrm{col}(\boldsymbol{U}_r) \right\}, \tag{20}$$

$$\overline{\mathcal{M}}^\perp := \left\{ \boldsymbol{\Theta} \in \mathbb{R}^{d_1 \times d_2} : \mathrm{row}(\boldsymbol{\Theta}) \perp \mathrm{row}(\boldsymbol{V}_r), \mathrm{col}(\boldsymbol{\Theta}) \perp \mathrm{col}(\boldsymbol{U}_r) \right\}, \tag{21}$$

where $\mathrm{row}(\cdot)$ and $\mathrm{col}(\cdot)$ denote row and column spaces, respectively.

For any $\Delta \in \mathbb{R}^{d_1 \times d_2}$, let $\boldsymbol{U}^\top \Delta \boldsymbol{V} = \begin{bmatrix} \boldsymbol{\Gamma}_{11}(\Delta) & \boldsymbol{\Gamma}_{12}(\Delta) \\ \boldsymbol{\Gamma}_{21}(\Delta) & \boldsymbol{\Gamma}_{22}(\Delta) \end{bmatrix}$, where $\boldsymbol{\Gamma}_{11}(\Delta) \in \mathbb{R}^{r \times r}$, $\boldsymbol{\Gamma}_{22}(\Delta) \in \mathbb{R}^{(d-r) \times (d-r)}$, $\boldsymbol{\Gamma}_{12}(\Delta) \in \mathbb{R}^{r \times (d-r)}$, and $\boldsymbol{\Gamma}_{21}(\Delta) \in \mathbb{R}^{(d-r) \times r}$. Then, one could consider the following decomposition:

$$\Delta = \underbrace{\boldsymbol{U} \begin{bmatrix} \boldsymbol{\Gamma}_{11}(\Delta) & \boldsymbol{\Gamma}_{12}(\Delta) \\ \boldsymbol{\Gamma}_{21}(\Delta) & \boldsymbol{0} \end{bmatrix} \boldsymbol{V}^\top}_{\triangleq \Delta_{\overline{\mathcal{M}}}} + \boldsymbol{U} \begin{bmatrix} \boldsymbol{0} & \boldsymbol{0} \\ \boldsymbol{0} & \boldsymbol{\Gamma}_{22}(\Delta) \end{bmatrix} \boldsymbol{V}^\top = \Delta_{\overline{\mathcal{M}}} + \begin{bmatrix} \boldsymbol{0} & \boldsymbol{0} \\ \boldsymbol{0} & \Delta_{\overline{\mathcal{M}}^\perp} \triangleq \boldsymbol{Q}_{d-2r} \boldsymbol{\Gamma}_{22}(\Delta) \boldsymbol{Q}_{d-2r}^\top \end{bmatrix}. \tag{22}$$

Note that $\mathrm{rank}(\Delta_{\overline{\mathcal{M}}}) \leq 2r$.

We then consider the following constraint cone:

$$\mathcal{C}(\boldsymbol{\Theta}_\star) := \left\{ \Delta \in \mathbb{R}^{d_1 \times d_2} : \left\| \Delta_{\overline{\mathcal{M}}^\perp} \right\|_{\mathrm{nuc}} \leq 3 \left\| \Delta_{\overline{\mathcal{M}}} \right\|_{\mathrm{nuc}} \right\}. \tag{23}$$

### C.2. $\mathcal{L}_N$ Satisfies LRSC With High Probability

We will now show that $\mathcal{L}_N$ satisfies LRSC with high probability:

**Lemma C.2.** *Let $W > 0$ be fixed, and suppose that $|\mathrm{supp}(\pi)| < \infty$. Then, with probability at least $1 - \frac{\delta}{2}$, $\mathcal{L}_N(\cdot)$ satisfies*
$\mathrm{LRSC}(\mathcal{C}, W, \lambda_{\min}(\boldsymbol{H}_A(\pi; \boldsymbol{\Theta}_\star)), \tau(W))$ *with* $\tau(W) := 16rW^2 R_{\max} \left( \sqrt{\frac{|\mathrm{supp}(\pi)| \log 2 + \log \frac{2}{\delta}}{N}} + 4\sqrt{2r} W R_s \right)$.

*Proof.* Let $\Delta \in \mathcal{C}(\boldsymbol{\Theta}_\star) \cap \mathcal{B}_F^{\mathrm{Skew}(d)}(W)$ be arbitrary, and denote $\boldsymbol{\Theta} = \boldsymbol{\Theta}_\star + \Delta$.

Note that

$$
\begin{aligned}
\langle \nabla \mathcal{L}_N(\boldsymbol{\Theta}) - \nabla \mathcal{L}_N(\boldsymbol{\Theta}_\star), \Delta \rangle &= \left\langle \frac{1}{N} \sum_{t=1}^N (\mu(\langle \boldsymbol{X}_t, \boldsymbol{\Theta} \rangle) - \mu_t(\langle \boldsymbol{X}_t, \boldsymbol{\Theta}_\star \rangle)) \boldsymbol{X}_t, \Delta \right\rangle \\
&= \left\langle \sum_{\boldsymbol{X} \in \mathrm{supp}(\pi)} \frac{N(\boldsymbol{X})}{N} (\mu(\langle \boldsymbol{X}, \boldsymbol{\Theta} \rangle) - \mu(\langle \boldsymbol{X}, \boldsymbol{\Theta}_\star \rangle)) \mathrm{vec}(\boldsymbol{X}), \mathrm{vec}(\Delta) \right\rangle \\
&\qquad\qquad\qquad\qquad\qquad (N(\boldsymbol{X}) := \textstyle\sum_{t=1}^N \mathbb{1}[\boldsymbol{X}_t = \boldsymbol{X}]) \\
&= \sum_{\boldsymbol{X} \in \mathrm{supp}(\pi)} \frac{N(\boldsymbol{X})}{N} (\dot{\mu}(\langle \boldsymbol{X}, \boldsymbol{\Theta}_\star \rangle) + G(\boldsymbol{\Theta}_\star, \boldsymbol{\Theta}; \boldsymbol{X}) \langle \mathrm{vec}(\boldsymbol{X}), \mathrm{vec}(\Delta) \rangle) \langle \mathrm{vec}(\boldsymbol{X}), \mathrm{vec}(\Delta) \rangle^2,
\end{aligned}
$$

(first-order Taylor expansion, $\mathrm{vec}(\Delta) = \mathrm{vec}(\boldsymbol{\Theta}_\star - \boldsymbol{\Theta})$)

where we define

$$
G(\boldsymbol{\Theta}_\star, \boldsymbol{\Theta}; \boldsymbol{X}) := \int_0^1 (1-z) \ddot{\mu}(\langle \boldsymbol{X}, z\boldsymbol{\Theta} + (1-z)\boldsymbol{\Theta}_\star \rangle) dz. \tag{24}
$$

Note that

$$
\begin{aligned}
|G(\boldsymbol{\Theta}_\star, \boldsymbol{\Theta}; \boldsymbol{X})| &\leq \int_0^1 (1-z) |\ddot{\mu}(\langle \boldsymbol{X}, z\boldsymbol{\Theta} + (1-z)\boldsymbol{\Theta}_\star \rangle)| \, dz \\
&\leq R_s \int_0^1 (1-z) \dot{\mu}(\langle \boldsymbol{X}, z\boldsymbol{\Theta} + (1-z)\boldsymbol{\Theta}_\star \rangle) dz &\text{(self-concordance)} \\
&\leq R_s R_{\max} \int_0^1 (1-z) dz &(\dot{\mu} \leq R_{\max}) \\
&= \frac{1}{2} R_s R_{\max}.
\end{aligned}
$$

Let us also define the empirical Hessian:

$$
\widehat{\boldsymbol{H}}(\pi; \boldsymbol{\Theta}_\star) := \sum_{\boldsymbol{X} \in \mathrm{supp}(\pi)} \frac{N(\boldsymbol{X})}{N} \dot{\mu}(\langle \boldsymbol{X}, \boldsymbol{\Theta}_\star \rangle) \mathrm{vec}(\boldsymbol{X}) \mathrm{vec}(\boldsymbol{X})^\top. \tag{25}
$$

Then, we can bound as

$$
\begin{aligned}
\langle \nabla \mathcal{L}_N(\boldsymbol{\Theta}) - \nabla \mathcal{L}_N(\boldsymbol{\Theta}_\star), \Delta \rangle &= \mathrm{vec}(\Delta)^\top \widehat{\boldsymbol{H}}(\pi; \boldsymbol{\Theta}_\star) \mathrm{vec}(\Delta) + \sum_{\boldsymbol{X} \in \mathrm{supp}(\pi)} \frac{N(\boldsymbol{X})}{N} G(\boldsymbol{\Theta}_\star, \boldsymbol{\Theta}_0; \boldsymbol{X}) \langle \mathrm{vec}(\boldsymbol{X}), \mathrm{vec}(\Delta) \rangle^3 \\
&\geq \mathrm{vec}(\Delta)^\top \widehat{\boldsymbol{H}}(\pi; \boldsymbol{\Theta}_\star) \mathrm{vec}(\Delta) - \frac{1}{2} R_s R_{\max} \sum_{\boldsymbol{X} \in \mathrm{supp}(\pi)} \frac{N(\boldsymbol{X})}{N} |\langle \boldsymbol{X}, \Delta \rangle|^3 \\
&= \mathrm{vec}(\Delta)^\top \widehat{\boldsymbol{H}}(\pi; \boldsymbol{\Theta}_\star) \mathrm{vec}(\Delta) - \frac{1}{2} R_s R_{\max} \|\Delta\|_{\mathrm{nuc}}^3.
\end{aligned}
$$

(matrix Hölder's inequality, $\|\boldsymbol{X}\|_{\mathrm{op}} \leq 1$ by Assumption 2)

The first term is bounded as

$$
\mathrm{vec}(\Delta)^\top \widehat{\boldsymbol{H}}(\pi; \boldsymbol{\Theta}_\star) \mathrm{vec}(\Delta)
$$

$$= \text{vec}(\Delta)^\top \boldsymbol{H}(\pi; \boldsymbol{\Theta}_\star)\text{vec}(\Delta) + \text{vec}(\Delta)^\top (\widehat{\boldsymbol{H}}(\pi; \boldsymbol{\Theta}_\star) - \boldsymbol{H}_A(\pi; \boldsymbol{\Theta}_\star))\text{vec}(\Delta_0)$$

$$\geq \|\Delta\|_F^2 \, \lambda_{\min}(\boldsymbol{H}(\pi; \boldsymbol{\Theta}_\star)) + \text{vec}(\Delta)^\top \underbrace{\left( \sum_{\boldsymbol{X} \in \text{supp}(\pi)} \left( \frac{N(\boldsymbol{X})}{N} - \pi(\boldsymbol{X}) \right) \dot{\mu}(\langle \boldsymbol{X}, \boldsymbol{\Theta}_\star \rangle) \text{vec}(\boldsymbol{X}) \text{vec}(\boldsymbol{X})^\top \right)}_{\triangleq E} \text{vec}(\Delta).$$

Let us now lower bound $E$:

$$E = \sum_{\boldsymbol{X} \in \text{supp}(\pi)} \left( \frac{N(\boldsymbol{X})}{N} - \pi(\boldsymbol{X}) \right) \dot{\mu}(\langle \boldsymbol{X}, \boldsymbol{\Theta}_\star \rangle) \langle \boldsymbol{X}, \Delta \rangle^2$$

$$\geq - \|\Delta\|_{\text{nuc}}^2 \sum_{\boldsymbol{X} \in \text{supp}(\pi)} \left| \frac{N(\boldsymbol{X})}{N} - \pi(\boldsymbol{X}) \right| \dot{\mu}(\langle \boldsymbol{X}, \boldsymbol{\Theta}_\star \rangle) \qquad \text{(matrix Hölder's inequality, } \|\boldsymbol{X}\|_{\text{op}} \leq 1\text{)}$$

$$\geq - \frac{R_{\max}}{4} \|\Delta\|_{\text{nuc}}^2 \sum_{\boldsymbol{X} \in \text{supp}(\pi)} \left| \frac{N(\boldsymbol{X})}{N} - \pi(\boldsymbol{X}) \right|. \qquad (\dot{\mu} \leq R_{\max})$$

For the last term, we utilize the following concentration for learning discrete distributions (of finite support) in $\ell_1$-distance:

**Lemma C.3** (Theorem 1 of Canonne (2020)). *Let $\mathcal{X}$ be a finite space, $\pi \in \mathcal{P}(\mathcal{X})$, and $\delta \in (0, 1)$. We are given $\{X_i\}_{i \in [N]}$ with $X_i \overset{i.i.d.}{\sim} \pi$. Let $\hat{\pi}_N \in \mathcal{P}(\mathcal{X})$ be defined as $\hat{\pi}_N(X) := \frac{1}{N} \sum_{i \in [N]} \mathbb{1}[X_i = X]$. Then, we have the following:*

$$\mathbb{P}\left( \|\pi - \hat{\pi}_N\|_1 := \sum_{X \in \mathcal{X}} |\pi(X) - \hat{\pi}_N(X)| \geq 2\sqrt{\frac{|\text{supp}(\pi)| \log 2 + \log \frac{2}{\delta}}{N}} \right) \leq \frac{\delta}{2}. \qquad (26)$$

Combining everything, we have that with probability at least $1 - \frac{\delta}{2}$,

$$\langle \nabla \mathcal{L}_N(\boldsymbol{\Theta}) - \nabla \mathcal{L}_N(\boldsymbol{\Theta}_\star), \Delta \rangle \geq \lambda_{\min}(\boldsymbol{H}(\pi; \boldsymbol{\Theta}_\star)) \|\Delta\|_F^2 - \frac{R_{\max}}{2} \left( \sqrt{\frac{|\text{supp}(\pi)| \log 2 + \log \frac{2}{\delta}}{N}} + R_s \|\Delta\|_{\text{nuc}} \right) \|\Delta\|_{\text{nuc}}^2.$$

As $\Delta \in \mathcal{C}(\boldsymbol{\Theta}_\star) \cap \mathcal{B}_F^{\text{Skew}(d)}(W)$, recalling the orthogonal subspace decompositions, $\overline{\mathcal{M}}$ and $\overline{\mathcal{M}}^\perp$:

$$\|\Delta\|_{\text{nuc}} \leq \|\Delta_{\overline{\mathcal{M}}}\|_{\text{nuc}} + \|\Delta_{\overline{\mathcal{M}}^\perp}\|_{\text{nuc}} \qquad \text{(triangle inequality)}$$

$$\leq 4 \|\Delta_{\overline{\mathcal{M}}}\|_{\text{nuc}} \qquad (\Delta \in \mathcal{C}(\boldsymbol{\Theta}_\star))$$

$$\leq 4\sqrt{2r} \|\Delta_{\overline{\mathcal{M}}}\|_F \qquad \text{(rank}(\Delta_{\overline{\mathcal{M}}}) \leq 2r, \text{ Cauchy-Schwartz inequality on the singular values)}$$

$$\leq 4\sqrt{2r} \|\Delta\|_F$$

$$\leq 4\sqrt{2r}W. \qquad (\Delta \in \mathcal{B}_F^{\text{Skew}(d)}(W))$$

Plugging it in, we have that

$$\langle \nabla \mathcal{L}_N(\boldsymbol{\Theta}) - \nabla \mathcal{L}_N(\boldsymbol{\Theta}_\star), \Delta \rangle \geq \lambda_{\min}(\boldsymbol{H}(\pi; \boldsymbol{\Theta}_\star)) \|\text{vec}(\Delta)\|_F^2 - 16rW^2 R_{\max} \left( \sqrt{\frac{|\text{supp}(\pi)| \log 2 + \log \frac{2}{\delta}}{N}} + 4\sqrt{2r}WR_s \right).$$

$$\square$$

**Remark 8** (Importance of $|\text{supp}(\pi)| < \infty$). *If $\pi$ is absolutely continuous w.r.t. the Lebesgue measure, than the usual empirical distribution $\widehat{\pi}_N := \frac{1}{N} \sum_{t=1}^N \delta_{\boldsymbol{X}_t}$ does not converge to $\pi$ in the total variational (TV) distance (Barron et al., 1992). Indeed, a stronger statement is possible: for any $\delta \in (0, 1/2)$ and for any sequence of distribution estimators $\{\pi_N\}$ on $\mathbb{R}$ (with Borel $\sigma$-algebra), there exists a probability measure $\pi$ such that $\inf_{N \geq 1} \|\pi_N - \pi\|_1 > \frac{1}{2} - \delta$, a.s. (Devroye & Györfi, 1990). Thus, to deal with $\pi$'s with continuous densities, one must consider an alternate form of empirical Hessian $\widehat{\boldsymbol{H}}$ via histogram or kernel density estimator (Tsybakov, 2009). We leave this to future work.*

## C.3. Choosing $\lambda_N$ such that $\|\nabla\mathcal{L}_N(\boldsymbol{\Theta}_\star)\|_{\mathrm{op}}$ is Well-Controlled

The following lemma explicitly characterizes (up to absolute constants!) the "correct" choice of $\lambda_{N_1}$:

**Lemma C.4** (Setting $\lambda_{N_1}$). *Let $\delta \in (0,1)$ and define $v(\delta, d_1, d_2) := \log(2\max(d_1, d_2)) + \min(d_1, d_2)\log\frac{5}{\delta}$. By setting $\lambda_{N_1} = f(\delta, d_1, d_2)\sqrt{\frac{1}{N}}$ with $f(\delta, d_1, d_2)$ as described below, we have $\mathbb{P}(\|\nabla\mathcal{L}_N(\boldsymbol{\Theta}_\star)\|_{\mathrm{op}} \leq \frac{\lambda_N}{2}) \geq 1 - \delta$:*

*(i) When $|y - \mu(\langle\boldsymbol{X}, \boldsymbol{\Theta}_\star\rangle)| \leq M$ a.s.: $f(\delta, d_1, d_2) = \sqrt{\frac{8R_{\max}}{g(\tau)}\log\frac{d_1+d_2}{\delta}}$, given that $N \geq \frac{2M^2}{9R_{\max}g(\tau)}\log\frac{d_1+d_2}{\delta}$,*

*(ii) When GLM is $\sigma$-subGaussian: $f(\delta, d_1, d_2) = \frac{16\pi\sigma}{g(\tau)}\sqrt{v(\delta)}$,*

*(iii) When Poisson: if $R_{\max} > e$, $f(\delta, d_1, d_2) = g_1(R_{\max}) + \frac{4}{1-2R_{\max}^{-1}}v(\delta, d_1, d_2)$ with $g_1(R_{\max}) := \frac{1}{2}(1-2R_{\max}^{-1})(R_{\max} + 2\log R_{\max} + 2\log\frac{2(1-2R_{\max}^{-1})}{e}) + 4R_{\max}\log R_{\max}$; otherwise, $f(\delta, d_1, d_2) = g_2(R_{\max}) + 8v(\delta, d_1, d_2)$ with $g_2(R_{\max}) := \frac{1}{8}(R_{\max} + 4\log R_{\max} + 4\log(8 + 2R_{\max})) + 4R_{\max}\log R_{\max}$.*

*Proof.* The proof is heavily inspired by Appendix C of Lee et al. (2024a), where the authors compute a high-probability bound for the global Lipschitz constant of $\mathcal{L}_N$. Here, we only need to bound it at $\boldsymbol{\Theta}_\star$, making our guarantee a bit tighter. During the proof, we also identify and improve suboptimal dependencies in Lee et al. (2024a), correctly leading to $\lambda_N$ scaling as $\sqrt{1/N}$ for all considered GLMs.

Let us prove each part separately:

### C.3.1. PROOF OF (I) – GLM BOUNDED BY $M$

Here, "bounded by $M$" means $|y - \langle\boldsymbol{X}, \boldsymbol{\Theta}_\star\rangle| \leq M$ a.s. The original proof of Lee et al. (2024a) is too loose, and thus we instead utilize the matrix Bernstein inequality (Tropp, 2015, Theorem 6.6.1), which we recall here:

**Theorem C.5** (Restatement of Theorem 6.1.1 of Tropp (2015)). *Let $\{\boldsymbol{A}_t\}_{t=1}^N \subset \mathbb{R}^{d_1 \times d_2}$ be independent with $\|\boldsymbol{A}_t\|_{\mathrm{op}} \leq L$ and $\mathbb{E}[\boldsymbol{A}_t] = \boldsymbol{A}$, and define their matrix variance statistics as*

$$\sigma_N^2 := \max\left\{\left\|\sum_{t=1}^N\mathbb{E}[\boldsymbol{A}_t\boldsymbol{A}_t^\top]\right\|_{\mathrm{op}}, \left\|\sum_{t=1}^N\mathbb{E}[\boldsymbol{A}_t^\top\boldsymbol{A}_t]\right\|_{\mathrm{op}}\right\}.$$

*Then we have that for any $\delta \in (0,1)$, as long as $b(N)^2 \geq \sigma_N^2 \geq \frac{2L^2}{9}\log\frac{d_1+d_2}{\delta}$ for a $b : \mathbb{N} \to \mathbb{R}_{>0}$,*

$$\mathbb{P}\left(\left\|\frac{1}{N}\sum_{t=1}^N\boldsymbol{A}_t - \boldsymbol{A}\right\|_{\mathrm{op}} \leq \frac{2b(N)}{N}\sqrt{2\log\frac{d_1+d_2}{\delta}}\right) \geq 1 - \delta. \tag{27}$$

As $\|\nabla\mathcal{L}_N(\boldsymbol{\Theta}_\star)\|_{\mathrm{op}} = \left\|\frac{1}{N}\sum_{t=1}^N\frac{\mu_t(\boldsymbol{\Theta}_\star)-y_t}{g(\tau)}\boldsymbol{X}_t\right\|_{\mathrm{op}}$, we set $\boldsymbol{A}_t = \frac{\mu_t(\boldsymbol{\Theta}_\star)-y_t}{g(\tau)}\boldsymbol{X}_t$, which satisfies $\boldsymbol{A} = \mathbb{E}[\boldsymbol{A}_t] = \boldsymbol{0}$. Its maximum deviation is bounded as

$$\left\|\frac{\mu_t(\boldsymbol{\Theta}_\star) - y_t}{g(\tau)}\boldsymbol{X}_t\right\|_{\mathrm{op}} \leq \frac{M}{g(\tau)}.$$

Its matrix variance statistics is bounded as

$$\sigma_N^2 = \frac{1}{g(\tau)^2}\max\left\{\left\|\sum_{t=1}^N\mathbb{E}_{\boldsymbol{X}\sim\pi}[\boldsymbol{X}\boldsymbol{X}^\top\mathbb{E}[(\mu(\langle\boldsymbol{X}, \boldsymbol{\Theta}_\star\rangle)-y)^2]]\right\|_{\mathrm{op}}, \left\|\sum_{t=1}^N\mathbb{E}_{\boldsymbol{X}\sim\pi}[\boldsymbol{X}^\top\boldsymbol{X}\mathbb{E}[(\mu(\langle\boldsymbol{X}, \boldsymbol{\Theta}_\star\rangle)-y)^2]]\right\|_{\mathrm{op}}\right\}$$

$$\leq \frac{1}{g(\tau)}\sum_{t=1}^N\dot\mu(\langle\boldsymbol{X}, \boldsymbol{\Theta}_\star\rangle) \qquad (\mathbb{E}[(\mu(\langle\boldsymbol{X}, \boldsymbol{\Theta}_\star\rangle)-y)^2] = \mathrm{Var}[y|\boldsymbol{X}] = g(\tau)\dot\mu(\langle\boldsymbol{X}, \boldsymbol{\Theta}_\star\rangle), \|\boldsymbol{X}\|_{\mathrm{op}} \leq 1)$$

$$\leq \frac{NR_{\max}}{g(\tau)}.$$

We then conclude by applying the matrix Bernstein inequality.

### C.3.2. PROOF OF (II) – $\sigma$-SUBGAUSSIAN GLM

Here, we first utilize a covering argument to reduce the problem to $\sigma$-norm-subGaussian vector concentration, where we utilize the results of Jin et al. (2019), refined in Appendix C.2 of Lee et al. (2024a).

Let $\widehat{\mathcal{B}}^{d_2}(1)$ be a $\frac{1}{2}$-cover of $\mathcal{B}^{d_2}(1) := \{\boldsymbol{\theta} \in \mathbb{R}^{d_2} : \|\boldsymbol{\theta}\|_2 \leq 1\}$. By Corollary 4.2.13 of Vershynin (2018), we can find a cover with $|\widehat{\mathcal{B}}^{d_2}(1)| \leq 5^{d_2}$. For each $\boldsymbol{u} \in \mathcal{B}^{d_2}(1)$, let $\hat{\boldsymbol{u}} \in \widehat{\mathcal{B}}^{d_2}(1)$ be such that $\|\boldsymbol{u} - \hat{\boldsymbol{u}}\|_2 \leq \varepsilon_N$. Then, we have that

$$
\begin{aligned}
\|\nabla \mathcal{L}_N(\boldsymbol{\Theta}_\star)\|_{\mathrm{op}} &= \sup_{\|\boldsymbol{u}\| \leq 1} \left\| \frac{1}{N} \sum_{t=1}^N \frac{\mu_t(\boldsymbol{\Theta}_\star) - y_t}{g(\tau)} \boldsymbol{X}_t \boldsymbol{u} \right\|_2 \\
&\leq \sup_{\|\boldsymbol{u}\| \leq 1} \left\{ \left\| \frac{1}{N} \sum_{t=1}^N \frac{\mu_t(\boldsymbol{\Theta}_\star) - y_t}{g(\tau)} \boldsymbol{X}_t (\boldsymbol{u} - \hat{\boldsymbol{u}}) \right\|_2 + \left\| \frac{1}{N} \sum_{t=1}^N \frac{\mu_t(\boldsymbol{\Theta}_\star) - y_t}{g(\tau)} \boldsymbol{X}_t \hat{\boldsymbol{u}} \right\|_2 \right\} \quad \text{(triangle inequality)} \\
&\leq \frac{1}{2} \|\nabla \mathcal{L}_N(\boldsymbol{\Theta}_\star)\|_{\mathrm{op}} + \sup_{\hat{\boldsymbol{u}} \in \widehat{\mathcal{B}}^{d_2}(1)} \left\| \frac{1}{N} \sum_{t=1}^N \frac{\mu_t(\boldsymbol{\Theta}_\star) - y_t}{g(\tau)} \boldsymbol{X}_t \hat{\boldsymbol{u}} \right\|_2,
\end{aligned}
$$

and thus,

$$
\|\nabla \mathcal{L}_N(\boldsymbol{\Theta}_\star)\|_{\mathrm{op}} \leq 2 \sup_{\hat{\boldsymbol{u}} \in \widehat{\mathcal{B}}^{d_2}(1)} \left\| \frac{1}{N} \sum_{t=1}^N \frac{\mu_t(\boldsymbol{\Theta}_\star) - y_t}{g(\tau)} \boldsymbol{X}_t \hat{\boldsymbol{u}} \right\|_2.
$$

For each fixed $\hat{\boldsymbol{u}}$ and $\delta' \in (0, 1)$, applying Corollary 7 of Jin et al. (2019)[7] gives

$$
\mathbb{P}\left( \left\| \frac{1}{N} \sum_{t=1}^N \frac{\mu_t(\boldsymbol{\Theta}_\star) - y_t}{g(\tau)} \boldsymbol{X}_t \hat{\boldsymbol{u}} \right\|_2 \leq \frac{4\pi\sigma}{g(\tau)} \sqrt{\frac{1}{N} \log \frac{2d_1}{\delta'}} \right) \geq 1 - \delta'.
$$

By the union bound, we finally have that

$$
\mathbb{P}\left( \|\nabla \mathcal{L}_N(\boldsymbol{\Theta}_\star)\|_{\mathrm{op}} \leq \frac{8\pi\sigma}{g(\tau)} \sqrt{\frac{1}{N} \left( \log(2d_1) + d_2 \log \frac{5}{\delta} \right)} \right) \geq 1 - \delta.
$$

By a symmetric argument with $\boldsymbol{X}_t^\top$, we can take the term in the square root as $\log(2 \max(d_1, d_2)) + \min(d_1, d_2) \log \frac{5}{\delta}$, and we are done.

### C.3.3. PROOF OF (III) – POISSON DISTRIBUTION

Note that $g(\tau) = 1$ for Poisson distribution. We again observe that the original proof of Lee et al. (2024a) is too loose.

First, via the same covering argument, it suffices to bound (with high probability) $\left\| \frac{1}{N} \sum_{t=1}^N (\mu_t(\boldsymbol{\Theta}_\star) - y_t) \boldsymbol{X}_t \hat{\boldsymbol{u}} \right\|_2$. Then we have from Appendix C.3 of Lee et al. (2024a) that

$$
\mathbb{P}\left( \left\| \frac{1}{N} \sum_{t=1}^N \frac{\mu_t(\boldsymbol{\Theta}_\star) - y_t}{g(\tau)} \boldsymbol{X}_t \hat{\boldsymbol{u}} \right\|_2 \leq \frac{1}{N} \inf_{\theta \in (0, 1/2)} \left\{ \theta \sum_{t=1}^N F(\theta, e^{\langle \boldsymbol{X}, \boldsymbol{\Theta}_\star \rangle}) + \frac{1}{\theta} \log \frac{2d_2}{\delta} \right\} \right) \geq 1 - \delta, \tag{28}
$$

where $F(\theta, v) := v\theta + \log(2\theta) + \log\left( \frac{e^{-\frac{v}{2}}}{\frac{1}{2} - \theta} + v \right)$ for $\theta > 0$.

Recall from Assumption 3 that $\max_{\boldsymbol{X} \in \mathcal{A}} e^{\langle \boldsymbol{X}, \boldsymbol{\Theta}_\star \rangle} \leq R_{\max}$. We choose $\theta = \frac{1}{\sqrt{N}} \left( \frac{1}{2} - \frac{1}{R_{\max}} \right)$ when $R_{\max} > e$ and $\frac{1}{4\sqrt{N}}$ otherwise. Then, applying the same argument symmetrically as previous, we have the desired result. $\qquad \square$

**Remark 9.** *Lafond (2015); Klopp (2014); Klopp et al. (2015) have utilized similar proof techniques involving (non-commutative) matrix concentration inequalities.*

---

[7] see Lemma C.1 of Lee et al. (2024a) for the version with explicit constants.

## C.4. Proof of Theorem 3.4 – LRSC and Our $\lambda_N$ Implies Good Rate

We now present the full version of Theorem 3.4 and its proof:

**Theorem C.6.** *Let* $\delta \in (0, 1)$ *and set* $\lambda_N = f(\delta, d_1, d_2)\sqrt{\frac{1}{N}}$ *as in Lemma C.4. Then, with*

$$N > \frac{2^{13}r^2 R_{\max}^2}{\lambda_{\min}(\boldsymbol{H}(\pi; \boldsymbol{\Theta}_\star))^2}\left(|\mathrm{supp}(\pi)|\log 2 + \log \frac{2}{\delta} + \frac{400 R_s^2 r^2 f(\delta, d_1, d_2)^2}{\lambda_{\min}(\boldsymbol{H}(\pi; \boldsymbol{\Theta}_\star))^2}\right), \tag{29}$$

*the following holds:*

$$\mathbb{P}\left(\|\boldsymbol{\Theta}_0 - \boldsymbol{\Theta}_\star\|_F \le \frac{5f(\delta, d_1, d_2)}{\sqrt{2}\lambda_{\min}(\boldsymbol{H}(\pi; \boldsymbol{\Theta}_\star))}\sqrt{\frac{r}{N}}\right) \ge 1 - \delta. \tag{30}$$

*Proof.* Similar to Fan et al. (2019), we will follow the localized analysis technique as introduced in Fan et al. (2018); see their Appendix B.3.2 and Figure 1 for a geometric intuition of the proof idea.

Let us denote $\Delta_0 := \boldsymbol{\Theta}_0 - \boldsymbol{\Theta}_\star$. We start by constructing a middle point $\widetilde{\boldsymbol{\Theta}}_\eta = \boldsymbol{\Theta}_\star + \eta\Delta_0$, where $\eta = 1$ if $\|\Delta_0\|_F \le W$ and $\eta = \frac{W}{\|\Delta_0\|_F}$ otherwise. We will choose an appropriate $W$ at the end.

Recall the definition of the constraint cone $\mathcal{C}(\boldsymbol{\Theta}_\star)$:

$$\mathcal{C}(\boldsymbol{\Theta}_\star) = \left\{\Delta \in \mathbb{R}^{d_1 \times d_2} : \left\|\Delta_{\overline{\mathcal{M}}^\perp}\right\|_{\mathrm{nuc}} \le 3\left\|\Delta_{\overline{\mathcal{M}}}\right\|_{\mathrm{nuc}}\right\}. \tag{31}$$

By Lemma 1(b) of Negahban & Wainwright (2011), $\Delta_0 \in \mathcal{C}$ is *implied* by $\|\nabla\mathcal{L}_N(\boldsymbol{\Theta}_\star)\|_{\mathrm{op}} \le \frac{\lambda_N}{2}$, which holds with probability at least $1 - \frac{\delta}{2}$ by Lemma C.4. Combining the above with Lemma C.2, we have that

$$\mathbb{P}(\Delta_0 \in \mathcal{C}(\boldsymbol{\Theta}_\star), \mathrm{LRSC}(\mathcal{C}(\boldsymbol{\Theta}_\star), W, \xi, \tau(W))) \ge 1 - \delta, \tag{32}$$

where $\xi = \lambda_{\min}(\boldsymbol{H}(\pi; \boldsymbol{\Theta}_\star))$ and $\tau(W) = 16rW^2 R_{\max}\left(\sqrt{\frac{|\mathrm{supp}(\pi)|\log 2 + \log\frac{2}{\delta}}{N}} + 4\sqrt{2r}WR_s\right)$, which we will assume to hold throughout the proof.

As LRSC holds and $\widetilde{\boldsymbol{\Theta}}_\eta - \boldsymbol{\Theta}_\star = \eta\Delta_0 \in \mathcal{C}(\boldsymbol{\Theta}_\star) \cap \mathcal{B}_F^{d_1 \times d_2}(W)$,

$$\xi\|\eta\Delta_0\|_F^2 - \tau(W) \le \frac{1}{2}B_{\mathcal{L}_N}^s(\widetilde{\boldsymbol{\Theta}}_\eta, \boldsymbol{\Theta}_\star) \overset{(*)}{\le} \frac{\eta}{2}B_{\mathcal{L}_N}^s(\boldsymbol{\Theta}_0, \boldsymbol{\Theta}_\star) = \frac{1}{2}\langle\nabla\mathcal{L}_N(\boldsymbol{\Theta}_0) - \nabla\mathcal{L}_N(\boldsymbol{\Theta}_\star), \eta\Delta_0\rangle, \tag{33}$$

where $(*)$ follows from Lemma F.4 of Fan et al. (2018).

As $\boldsymbol{\Theta}_0$ is the solution to the nonsmooth convex optimization (Eqn. (16)), its first-order optimality condition (Rockafellar, 1970) implies the following:

$$\exists \boldsymbol{\Xi} \in \partial\|\cdot\|_{\mathrm{nuc}}|_{\boldsymbol{\Theta}_0}, \exists \boldsymbol{V} \in N_\Omega(\boldsymbol{\Theta}_0): \quad \nabla\mathcal{L}_N(\boldsymbol{\Theta}_0) + \lambda_N\boldsymbol{\Xi} + \boldsymbol{V} = \boldsymbol{0}, \tag{34}$$

where $\partial\|\cdot\|_{\mathrm{nuc}}$ is the (Clarke) subdifferential of the nuclear norm, and $N_\Omega(\boldsymbol{\Theta}_0) := \{\boldsymbol{V} \in \mathbb{R}^{d_1 \times d_2} : \langle\boldsymbol{V}, \boldsymbol{Y} - \boldsymbol{\Theta}_0\rangle \le 0, \forall\boldsymbol{Y} \in \Omega\}$ is the normal cone of $\Omega$ at $\boldsymbol{\Theta}_0$.

It can be deduced from the closed form of $\partial\|\cdot\|_{\mathrm{nuc}}$ (see Example 2 of Watson (1992)) that $\|\boldsymbol{\Xi}\|_{\mathrm{op}} \le 2$. Thus, we have that

$$\begin{aligned}
\xi\|\eta\Delta_0\|_F^2 - \tau(W) &\le \frac{1}{2}\langle\nabla\mathcal{L}_N(\boldsymbol{\Theta}_0) - \nabla\mathcal{L}_N(\boldsymbol{\Theta}_\star), \eta\Delta_0\rangle \\
&= -\frac{1}{2}\langle\lambda_N\boldsymbol{\Xi} + \boldsymbol{V} + \nabla\mathcal{L}_N(\boldsymbol{\Theta}_\star), \eta\Delta_0\rangle \\
&= -\frac{1}{2}\langle\lambda_N\boldsymbol{\Xi} + \nabla\mathcal{L}_N(\boldsymbol{\Theta}_\star), \eta\Delta_0\rangle + \frac{\eta}{2}\langle\boldsymbol{V}, \boldsymbol{\Theta}_\star - \boldsymbol{\Theta}_0\rangle \quad \text{(Definition of } \Delta_0) \\
&\le \frac{1}{2}(\lambda_N\|\boldsymbol{\Xi}\|_{\mathrm{op}} + \|\nabla\mathcal{L}_N(\boldsymbol{\Theta}_\star)\|_{\mathrm{op}})\|\eta\Delta_0\|_{\mathrm{nuc}} \\
&\quad \text{(matrix Hölder's inequality, triangle inequality, definition of normal cone \& } \boldsymbol{\Theta}_\star \in \Omega)
\end{aligned}$$

$$\leq \frac{5}{4}\lambda_N \left\|\eta\Delta_0\right\|_{\mathrm{nuc}}. \qquad\qquad (\left\|\Xi\right\|_{\mathrm{op}} \leq 2, \text{ Lemma C.4})$$

Again recalling the orthogonal subspace decompositions, $\overline{\mathcal{M}}$ and $\overline{\mathcal{M}}^{\perp}$:

$$
\begin{aligned}
\left\|\Delta_0\right\|_{\mathrm{nuc}} &\leq \left\|(\Delta_0)_{\overline{\mathcal{M}}}\right\|_{\mathrm{nuc}} + \left\|(\Delta_0)_{\overline{\mathcal{M}}^{\perp}}\right\|_{\mathrm{nuc}} && \text{(triangle inequality)}\\
&\leq 4\left\|(\Delta_0)_{\overline{\mathcal{M}}}\right\|_{\mathrm{nuc}} && (\Delta_0 \in \mathcal{C}(\boldsymbol{\Theta}_\star))\\
&\leq 4\sqrt{2r}\left\|(\Delta_0)_{\overline{\mathcal{M}}}\right\|_F && \text{(Cauchy-Schwartz inequality on the singular values)}\\
&\leq 4\sqrt{2r}\left\|\Delta_0\right\|_F.
\end{aligned}
$$

Combining everything, we have that

$$\xi\left\|\eta\Delta_0\right\|_F^2 - \tau(W) \leq 5\sqrt{2r}\lambda_N \left\|\eta\Delta_0\right\|_F.$$

Solving this quadratic inequality gives

$$\left\|\widetilde{\boldsymbol{\Theta}}_\eta - \boldsymbol{\Theta}_\star\right\|_F = \left\|\eta\Delta_0\right\|_F \leq \frac{5\sqrt{r}\lambda_N}{\sqrt{2}\xi} + \sqrt{\frac{\tau(W)}{\xi} + \frac{25r\lambda_N^2}{2\xi^2}} \leq \underbrace{\frac{5\sqrt{2r}\lambda_N}{\xi} + \sqrt{\frac{\tau(W)}{\xi}}}_{\mathrm{RHS}},$$

where the last inequality follows from $\sqrt{a+b} \leq \sqrt{a} + \sqrt{b}$.

We will now choose $W$ such that RHS $< W$ (forcing a contraction into $\mathcal{B}_F^{\mathrm{Skew}(d)}(W)$, which implies that $\eta = 1$ and thus $\widetilde{\boldsymbol{\Theta}}_\eta = \boldsymbol{\Theta}_0$: if not (i.e., if RHS $< W$ and $\eta < 1$), then $W = \left\|\widetilde{\boldsymbol{\Theta}}_\eta - \boldsymbol{\Theta}_\star\right\| < W$, a contradiction.

Set[8] $W = \frac{5\sqrt{r}\lambda_N}{\sqrt{2}\xi} = \frac{5f(\delta,d_1,d_2)}{\sqrt{2}\xi}\sqrt{\frac{r}{N}}$. We then conclude by deriving a condition on $N$ for RHS $< W$. Although the computation is a bit tedious, we provide the details for completeness.

First, RHS $< W$ writes

$$\frac{W}{2} + 4W\sqrt{\frac{rR_{\max}}{\xi}\left(\sqrt{\frac{|\mathrm{supp}(\pi)|\log 2 + \log\frac{2}{\delta}}{N}} + 4\sqrt{2r}R_sW\right)} < W.$$

Canceling $W$ on both sides, plugging in our choice of $W$ and rearranging give

$$\frac{64rR_{\max}}{\xi}\left(\sqrt{\frac{|\mathrm{supp}(\pi)|\log 2 + \log\frac{2}{\delta}}{N}} + \frac{20R_srf(\delta,d_1,d_2)}{\xi}\sqrt{\frac{1}{N}}\right) < 1.$$

To avoid any cross terms, we use $(\sqrt{a} + \sqrt{b})^2 \leq 2(a+b)$ and solve for $N$, which gives

$$N > \frac{2^{13}r^2R_{\max}^2}{\xi^2}\left(|\mathrm{supp}(\pi)|\log 2 + \log\frac{2}{\delta} + \frac{400R_s^2r^2f(\delta,d_1,d_2)^2}{\xi^2}\right). \qquad (35)$$

$\square$

---

[8]Here, we did not make any effort to optimize the constants.

# D. Proof of Theorem 3.1 – Error Bound of Stage II

We first recall the following result on the robust estimation of matrix mean due to Minsker (2018), which is a generalization of the seminal result of Catoni (2012) to matrices:

**Lemma D.1** (Corollary 3.1 of Minsker (2018)). *Let $\{A_i\}_{i=1}^n \subset \mathbb{R}^{d_1 \times d_2}$ be independent with $\mathbb{E}[A_i] = A$, and define their matrix variance statistics as*

$$\sigma_n^2 := \max \left\{ \left\| \sum_{i=1}^n \mathbb{E}[A_i A_i^\top] \right\|_{\mathrm{op}}, \left\| \sum_{i=1}^n \mathbb{E}[A_i^\top A_i] \right\|_{\mathrm{op}} \right\}.$$

*Then we have that for any $\delta \in (0, 1)$,*

$$\mathbb{P} \left( \left\| \widehat{T} - A \right\|_{\mathrm{op}} \leq \sqrt{\frac{2\sigma_n^2}{n^2} \log \frac{2(d_1 + d_2)}{\delta}} \right) \geq 1 - \delta,$$

*where*

$$\widehat{T} := \frac{1}{n} \left( \sum_{i=1}^n \tilde{\psi}_\nu(A_i) \right)_{\mathrm{ht}}, \quad \nu := \sqrt{\frac{2}{\sigma_n^2} \log \frac{2(d_1 + d_2)}{\delta}}.$$

**Remark 10.** *The significance of the Catoni-type robust estimator is that the guarantee does not assume the boundedness of the matrices, yet it still gives a Bernstein-type concentration. This has been successfully utilized in obtaining tight, instance-specific guarantees for various reinforcement learning problems, such as sparse linear bandits (Jang et al., 2022), low-rank bandits (Jang et al., 2024), linear MDP (Wagenmaker et al., 2022), and more.*

For simplicity let us denote $\pi \triangleq \pi_2$. Recall the Hessian:

$$\boldsymbol{H}(\pi; \boldsymbol{\Theta}_0) := \mathbb{E}_{\boldsymbol{X} \sim \pi} \left[ \dot{\mu}(\langle \boldsymbol{X}, \boldsymbol{\Theta}_0 \rangle) \mathrm{vec}(\boldsymbol{X}) \mathrm{vec}(\boldsymbol{X})^\top \right], \tag{36}$$

and the one-sample estimators (line 9 of Algorithm 1): for each $t \in [N_1]$,

$$\widetilde{\boldsymbol{\Theta}}_t = \mathrm{vec}_{d \times d}^{-1} \left( \widetilde{\boldsymbol{\theta}}_t \right), \quad \widetilde{\boldsymbol{\theta}}_t := \boldsymbol{H}(\pi; \boldsymbol{\Theta}_0)^{-1} \left( y_t - \mu(\langle \boldsymbol{X}_t, \boldsymbol{\Theta}_0 \rangle) \right) \mathrm{vec}(\boldsymbol{X}_t), \tag{37}$$

We will utilize the above lemma to estimate $\boldsymbol{\Theta}_\star - \boldsymbol{\Theta}_0$ via $\widetilde{\boldsymbol{\Theta}}_t$'s. The key technical challenge lies in how to control the bias of those one-sample estimators, which we will see soon.

We first have that

$$
\begin{aligned}
&\mathbb{E}[\widetilde{\boldsymbol{\theta}}_t | \boldsymbol{X}_t = \boldsymbol{X}] \\
&= \boldsymbol{H}(\pi; \boldsymbol{\Theta}_0)^{-1} \left[ \mu(\langle \boldsymbol{X}, \boldsymbol{\Theta}_\star \rangle) - \mu(\langle \boldsymbol{X}, \boldsymbol{\Theta}_0 \rangle) \right] \mathrm{vec}(\boldsymbol{X}) \\
&\overset{(*)}{=} \boldsymbol{H}(\pi; \boldsymbol{\Theta}_0)^{-1} \left[ \dot{\mu}(\langle \boldsymbol{X}, \boldsymbol{\Theta}_0 \rangle) \langle \boldsymbol{\Theta}_\star - \boldsymbol{\Theta}_0, \boldsymbol{X} \rangle + \langle \boldsymbol{\Theta}_\star - \boldsymbol{\Theta}_0, \boldsymbol{X} \rangle^2 G(\boldsymbol{\Theta}_0, \boldsymbol{\Theta}_\star; \boldsymbol{X}) \right] \mathrm{vec}(\boldsymbol{X}) \\
&\qquad\qquad\qquad\qquad\qquad\qquad\qquad\qquad \text{(first-order Taylor expansion with integral remainder)} \\
&= \boldsymbol{H}(\pi; \boldsymbol{\Theta}_0)^{-1} \left[ \dot{\mu}(\langle \boldsymbol{X}, \boldsymbol{\Theta}_0 \rangle) \mathrm{vec}(\boldsymbol{X}) \mathrm{vec}(\boldsymbol{X})^\top \mathrm{vec}(\boldsymbol{\Theta}_\star - \boldsymbol{\Theta}_0) + \langle \boldsymbol{\Theta}_\star - \boldsymbol{\Theta}_0, \boldsymbol{X} \rangle^2 G(\boldsymbol{\Theta}_0, \boldsymbol{\Theta}_\star; \boldsymbol{X}) \mathrm{vec}(\boldsymbol{X}) \right] \\
&= \boldsymbol{H}(\pi; \boldsymbol{\Theta}_0)^{-1} \left[ \dot{\mu}(\langle \boldsymbol{X}, \boldsymbol{\Theta}_0 \rangle) \mathrm{vec}(\boldsymbol{X}) (\mathrm{vec}(\boldsymbol{X}))^\top \mathrm{vec}(\boldsymbol{\Theta}_\star - \boldsymbol{\Theta}_0) + \langle \boldsymbol{\Theta}_\star - \boldsymbol{\Theta}_0, \boldsymbol{X} \rangle^2 G(\boldsymbol{\Theta}_0, \boldsymbol{\Theta}_\star; \boldsymbol{X}) \mathrm{vec}(\boldsymbol{X}) \right] \\
&= \boldsymbol{H}(\pi; \boldsymbol{\Theta}_0)^{-1} \left[ \dot{\mu}(\langle \boldsymbol{X}, \boldsymbol{\Theta}_0 \rangle) (\mathrm{vec}(\boldsymbol{X})) (\mathrm{vec}(\boldsymbol{X}))^\top \mathrm{vec}(\boldsymbol{\Theta}_\star - \boldsymbol{\Theta}_0) + \langle \boldsymbol{\Theta}_\star - \boldsymbol{\Theta}_0, \mathrm{vec}(\boldsymbol{X}) \rangle^2 G(\boldsymbol{\Theta}_0, \boldsymbol{\Theta}_\star; \boldsymbol{X}) \mathrm{vec}(\boldsymbol{X}) \right],
\end{aligned}
$$

where at $(*)$, we define

$$G(\boldsymbol{\Theta}_0, \boldsymbol{\Theta}_\star; \boldsymbol{X}) := \int_0^1 (1 - z) \ddot{\mu}(\langle z\boldsymbol{\Theta}_\star + (1 - z)\boldsymbol{\Theta}_0, \boldsymbol{X} \rangle) dz. \tag{38}$$

By taking the expectation over $\boldsymbol{X} \sim \pi$, we have that

$$\mathbb{E}[\widetilde{\boldsymbol{\theta}}_t] = \mathrm{vec}(\boldsymbol{\Theta}_\star - \boldsymbol{\Theta}_0) + \mathbb{E}_{\boldsymbol{X} \sim \pi} \left[ \langle \boldsymbol{\Theta}_\star - \boldsymbol{\Theta}_0, \mathrm{vec}(\boldsymbol{X}) \rangle^2 G(\boldsymbol{\Theta}_0, \boldsymbol{\Theta}_\star; \boldsymbol{X}) \boldsymbol{H}(\pi; \boldsymbol{\Theta}_0)^{-1} \mathrm{vec}(\boldsymbol{X}) \right], \tag{39}$$

We will assume that $\|\boldsymbol{\Theta}_\star - \boldsymbol{\Theta}_0\|_{\mathrm{nuc}} \leq E \asymp \frac{rf(\delta,d_1,d_2)}{C_H(\pi_1)}\sqrt{\frac{1}{N_1}}$, which holds with probability at least $1 - \frac{\delta}{2}$ by Theorem 3.4 and the fact that $\|\boldsymbol{A}\|_{\mathrm{nuc}} \leq \sqrt{\mathrm{rank}(\boldsymbol{A})}\|\boldsymbol{A}\|_F$.

Note that $\widetilde{\boldsymbol{\theta}}_t$'s are *biased* estimators of $\mathrm{vec}(\boldsymbol{\Theta}_\star - \boldsymbol{\Theta}_0)$:

$$
\begin{aligned}
\left\|\mathbb{E}[\widetilde{\boldsymbol{\Theta}}_t] - (\boldsymbol{\Theta}_\star - \boldsymbol{\Theta}_0)\right\|_{\mathrm{op}} &= \left\|\mathbb{E}_{\boldsymbol{X}\sim\pi}\left[\langle\boldsymbol{\Theta}_\star - \boldsymbol{\Theta}_0, \mathrm{vec}(\boldsymbol{X})\rangle^2 G(\boldsymbol{\Theta}_0, \boldsymbol{\Theta}_\star; \boldsymbol{X})\mathrm{vec}^{-1}(\boldsymbol{H}(\pi;\boldsymbol{\Theta}_0)^{-1}\mathrm{vec}(\boldsymbol{X}))\right]\right\|_{\mathrm{op}} \\
&\leq \mathbb{E}_{\boldsymbol{X}\sim\pi}\left[\langle\boldsymbol{\Theta}_\star - \boldsymbol{\Theta}_0, \mathrm{vec}(\boldsymbol{X})\rangle^2|G(\boldsymbol{\Theta}_0, \boldsymbol{\Theta}_\star; \boldsymbol{X})|\left\|\mathrm{vec}^{-1}(\boldsymbol{H}(\pi;\boldsymbol{\Theta}_0)^{-1}\mathrm{vec}(\boldsymbol{X}))\right\|_{\mathrm{op}}\right] \\
&\hspace{8cm}\text{(Jensen's inequality)} \\
&\leq \frac{1}{2}R_s R_{\max}E^2\mathbb{E}_{\boldsymbol{X}\sim\pi}\left[\left\|\mathrm{vec}^{-1}(\boldsymbol{H}(\pi;\boldsymbol{\Theta}_0)^{-1}\mathrm{vec}(\boldsymbol{X}))\right\|_F\right] \\
&\hspace{4cm}(|G(\boldsymbol{\Theta}_0, \boldsymbol{\Theta}_\star; \boldsymbol{X})| \leq \tfrac{1}{2}R_s R_{\max} \text{ from proof of Lemma C.2}) \\
&= \frac{1}{2}R_s R_{\max}E^2\mathbb{E}_{\boldsymbol{X}\sim\pi}\left[\left\|\boldsymbol{H}(\pi;\boldsymbol{\Theta}_0)^{-1}\mathrm{vec}(\boldsymbol{X})\right\|_2\right] \\
&\leq \frac{1}{2}R_s R_{\max}E^2\sqrt{\mathbb{E}_{\boldsymbol{X}\sim\pi}\left[\mathrm{vec}(\boldsymbol{X})^\top\boldsymbol{H}(\pi;\boldsymbol{\Theta}_0)^{-2}\mathrm{vec}(\boldsymbol{X})\right]}. \hspace{1cm}\text{(Jensen's inequality)}
\end{aligned}
$$

We will control this bias at the end.

In order to apply the matrix Catoni estimator of Minsker (2018), we bound the matrix variance statistics of the one-sample estimators $\widetilde{\boldsymbol{\Theta}}_t$'s, whose proof is deferred to the end of this section:

**Lemma D.2.**

$$
\sigma_n^2 := \max\left\{\left\|\sum_{t=1}^{N_2}\mathbb{E}[\widetilde{\boldsymbol{\Theta}}_t\widetilde{\boldsymbol{\Theta}}_t^\top]\right\|_{\mathrm{op}}, \left\|\sum_{t=1}^{N_2}\mathbb{E}[\widetilde{\boldsymbol{\Theta}}_t^\top\widetilde{\boldsymbol{\Theta}}_t]\right\|_{\mathrm{op}}\right\} \leq \frac{1}{2}(1 + 2R_sE)\left(g(\tau) + \frac{E^2R_{\max}^2}{\kappa_\star}\right)\mathrm{GL}(\pi)N_2, \quad (40)
$$

*where* $\mathrm{GL}(\pi) := \max\{H^{(\mathrm{row})}(\pi), H^{(\mathrm{col})}(\pi)\}$ *with*

$$
H^{(\mathrm{row})}(\pi) := \lambda_{\max}\left(\sum_{m=1}^{d_2}\boldsymbol{D}_m^{(\mathrm{row})}(\pi)\right), \quad \boldsymbol{D}_m^{(\mathrm{row})}(\pi) := [(\boldsymbol{H}(\pi;\boldsymbol{\Theta}_0)^{-1})_{jk}]_{j,k\in\{\ell+d_1(m-1):\ell\in[d_1]\}}, \quad (41)
$$

*and*

$$
H^{(\mathrm{col})}(\pi) := \lambda_{\max}\left(\sum_{m=1}^{d_1}\boldsymbol{D}_m^{(\mathrm{col})}(\pi)\right), \quad \boldsymbol{D}_m^{(\mathrm{col})}(\pi) := [(\boldsymbol{H}(\pi;\boldsymbol{\Theta}_0)^{-1})_{jk}]_{j,k\in\{m+d_1(\ell-1):\ell\in[d_2]\}}. \quad (42)
$$

*A nice illustration of $\boldsymbol{D}_m^{(\mathrm{row})}$ and $\boldsymbol{D}_m^{(\mathrm{col})}$ is provided in Figure 1 of Jang et al. (2024).*

Then, recalling the definition of $\boldsymbol{\Theta}_1$ (line 14 of Algorithm 1) and denoting the matrix Catoni estimator for $\widetilde{\boldsymbol{\Theta}}_t$'s as $\widehat{T}_N$, we have that

$$
\begin{aligned}
\left\|(\boldsymbol{\Theta}_1 - \boldsymbol{\Theta}_0) - \mathrm{Proj}_\Omega(\mathbb{E}[\widetilde{\boldsymbol{\Theta}}_t])\right\|_{\mathrm{op}} &= \left\|\mathrm{Proj}_\Omega(\boldsymbol{\Theta}_0 + \widehat{T}_N) - \boldsymbol{\Theta}_0 - \mathrm{Proj}_\Omega(\mathbb{E}[\widetilde{\boldsymbol{\Theta}}_t])\right\|_{\mathrm{op}} \quad (43) \\
&\leq \left\|\widehat{T}_N - \mathbb{E}[\widetilde{\boldsymbol{\Theta}}_t]\right\|_{\mathrm{op}} \hspace{2cm}(\mathrm{Proj}_\Omega \text{ is a linear contraction mapping}) \\
&\leq \sqrt{\frac{\mathrm{GL}(\pi)}{N_2}(1 + 2R_sE)\left(g(\tau) + \frac{E^2R_{\max}^2}{\kappa_\star}\right)\log\frac{4(d_1 + d_2)}{\delta}} \\
&\hspace{3cm}\text{(with probability at least } 1 - \delta/2, \text{ by Lemma D.1 and D.2)}
\end{aligned}
$$

Let us now control the bias appropriately. To do that, we recall the following lemma that relates $\boldsymbol{H}(\pi;\boldsymbol{\Theta}_0)$ to $\boldsymbol{H}(\pi;\boldsymbol{\Theta}_\star)$:

**Lemma D.3** (Lemma 5 of Jun et al. (2021), adapted to our notations). *Suppose* $R_s\|\boldsymbol{\Theta}_\star - \boldsymbol{\Theta}_0\|_{\mathrm{nuc}} \leq R_sE \leq 1$. *Then, we have that*

$$
\frac{1}{1 + 2R_sE}\boldsymbol{H}(\pi;\boldsymbol{\Theta}_\star) \preceq \boldsymbol{H}(\pi;\boldsymbol{\Theta}_0) \preceq (1 + 2R_sE)\boldsymbol{H}(\pi;\boldsymbol{\Theta}_\star). \quad (44)
$$

Thus,

$$
\begin{aligned}
&\left\| \mathrm{Proj}_\Omega(\mathbb{E}[\widetilde{\boldsymbol{\Theta}}_t]) - (\boldsymbol{\Theta}_\star - \boldsymbol{\Theta}_0) \right\|_{\mathrm{op}} \\
&= \left\| \mathrm{Proj}_\Omega \left( \mathbb{E}[\widetilde{\boldsymbol{\Theta}}_t] - (\boldsymbol{\Theta}_\star - \boldsymbol{\Theta}_0) \right) \right\|_{\mathrm{op}} && (\boldsymbol{\Theta}_\star, \boldsymbol{\Theta}_0 \in \Omega,\ \mathrm{Proj}_\Omega \text{ is linear}) \\
&\leq \left\| \mathbb{E}[\widetilde{\boldsymbol{\Theta}}_t] - (\boldsymbol{\Theta}_\star - \boldsymbol{\Theta}_0) \right\|_{\mathrm{op}} && (\mathrm{Proj}_\Omega \text{ is a contraction}) \\
&\leq \frac{1}{2} R_s R_{\max} E^2 \sqrt{\mathbb{E}_{\boldsymbol{X} \sim \pi}\left[ \mathrm{vec}(\boldsymbol{X})^\top \boldsymbol{H}(\pi; \boldsymbol{\Theta}_0)^{-2} \mathrm{vec}(\boldsymbol{X}) \right]} \\
&= \frac{1}{2} R_s R_{\max} E^2 \sqrt{\mathrm{tr}(\mathbb{E}_{\boldsymbol{X} \sim \pi}\left[ \mathrm{vec}(\boldsymbol{X}) \mathrm{vec}(\boldsymbol{X})^\top \right] \boldsymbol{H}(\pi; \boldsymbol{\Theta}_0)^{-2})} && (\text{cyclic property \& linearity of } \mathrm{tr}(\cdot)) \\
&\leq \frac{1}{2} R_s R_{\max} E^2 \sqrt{\frac{1 + 2R_s E}{\kappa_\star} \mathrm{tr}(\boldsymbol{H}(\pi; \boldsymbol{\Theta}_0)^{-1})} && (\tfrac{\kappa_\star}{1+2R_s E} \boldsymbol{V}(\pi) \preceq \tfrac{1}{1+2R_s E} \boldsymbol{H}(\pi; \boldsymbol{\Theta}_\star) \preceq \boldsymbol{H}(\pi; \boldsymbol{\Theta}_0) \text{ by Lemma D.3}) \\
&= \frac{1}{2} R_s R_{\max} E^2 \sqrt{\frac{1 + 2R_s E}{\kappa_\star} \max\left\{ \mathrm{tr}\left( \sum_{m=1}^d \boldsymbol{D}_m^{(\mathrm{row})} \right), \mathrm{tr}\left( \sum_{m=1}^d \boldsymbol{D}_m^{(\mathrm{col})} \right) \right\}} \\
&\leq \frac{1}{2} R_s R_{\max} E^2 \sqrt{\frac{(d_1 \vee d_2)(1 + 2R_s E)}{\kappa_\star} \max\left\{ \lambda_{\max}\left( \sum_{m=1}^d \boldsymbol{D}_m^{(\mathrm{row})} \right), \lambda_{\max}\left( \sum_{m=1}^d \boldsymbol{D}_m^{(\mathrm{col})} \right) \right\}} \\
&&& (\text{for a } d \times d \text{ square matrix } \boldsymbol{A} \succeq \boldsymbol{0},\ \mathrm{tr}(\boldsymbol{A}) \leq d\lambda_{\max}(\boldsymbol{A})) \\
&= \frac{1}{2} R_s R_{\max} E^2 \sqrt{\frac{(d_1 \vee d_2)(1 + 2R_s E)}{\kappa_\star} \mathrm{GL}(\pi)}.
\end{aligned}
\tag{45}
$$

Combining everything we have that:

$$
\begin{aligned}
\|\boldsymbol{\Theta}_1 - \boldsymbol{\Theta}_\star\|_{\mathrm{op}} &\leq \left\| (\boldsymbol{\Theta}_1 - \boldsymbol{\Theta}_0) - \mathrm{Proj}_\Omega(\mathbb{E}[\widetilde{\boldsymbol{\Theta}}_t]) \right\|_{\mathrm{op}} + \left\| \mathrm{Proj}_\Omega(\mathbb{E}[\widetilde{\boldsymbol{\Theta}}_t]) - (\boldsymbol{\Theta}_\star - \boldsymbol{\Theta}_0) \right\|_{\mathrm{op}} \\
&\leq \sqrt{(1 + 2R_s E)\mathrm{GL}(\pi)} \left( \sqrt{\frac{1}{N_2}\left( g(\tau) + \frac{E^2 R_{\max}^2}{\kappa_\star} \right) \log \frac{4(d_1 + d_2)}{\delta}} + \frac{1}{2} R_s R_{\max} E^2 \sqrt{\frac{d_1 \vee d_2}{\kappa_\star}} \right).
\end{aligned}
\tag{46}
$$

Combining above with Theorem 3.4 (Guarantee for Stage I), it can be deduced that with

$$
N_1 \gtrsim \max\left\{ \widetilde{N}_1, \frac{R_s R_{\max} f(\delta, d_1, d_2)^2 r^2}{C_H(\pi_1)^2} \sqrt{\frac{(d_1 \vee d_2)N_2}{g(\tau)\kappa_\star^5 \log \frac{d_1 \vee d_2}{\delta}}} \right\},
\tag{47}
$$

the following holds with probability at least $1 - \delta$:

$$
\|\boldsymbol{\Theta}_1 - \boldsymbol{\Theta}_\star\|_{\mathrm{op}} \leq \sigma_{\mathrm{thres}} \triangleq 2\sqrt{\frac{2(1 + R_s)g(\tau)\mathrm{GL}(\pi)}{N_2} \log \frac{4(d_1 + d_2)}{\delta}}.
\tag{48}
$$

As the last step of the proof, we recall the Weyl's inequality for singular values:

**Lemma D.4** (Problem 7.3.P16 of Horn & Johnson (2012)). *For any $\boldsymbol{A}, \Delta \in \mathbb{R}^{d_1 \times d_2}$, we have*

$$
|\sigma_k(\boldsymbol{A} + \Delta) - \sigma_k(\boldsymbol{A})| \leq \sigma_1(\Delta), \quad \forall k \in [\min\{d_1, d_2\}].
$$

As $\sigma_k(\boldsymbol{\Theta}_\star) = 0$ for $k \geq r + 1$, we have that $\sigma_k(\boldsymbol{\Theta}_1) \leq \sigma_{\mathrm{thres}}$ for the same $k$'s as well. This proves that the thresholding part of our algorithm (line 11) indeed yields $\mathrm{rank}(\widehat{\boldsymbol{\Theta}}) \leq r$. The final error bound follows from triangle inequality.

$\square$

*Proof of Lemma D.2.* We will bound $\left\|\mathbb{E}[\widetilde{\boldsymbol{\Theta}}_t \widetilde{\boldsymbol{\Theta}}_t^\top]\right\|_{\mathrm{op}}$ only, as the other one follows analogously.

We first establish the following: by the fundamental theorem of calculus,

$$|\mu(\langle \boldsymbol{X}, \boldsymbol{\Theta}_\star\rangle) - \mu(\langle \boldsymbol{X}, \boldsymbol{\Theta}_0\rangle)| = |\langle \boldsymbol{X}, \boldsymbol{\Theta}_\star - \boldsymbol{\Theta}_0\rangle| \int_0^1 \dot\mu(\langle \boldsymbol{X}, (1-z)\boldsymbol{\Theta}_\star + z\boldsymbol{\Theta}_0\rangle)dz \le ER_{\max},$$

and thus, for $y \sim p(\cdot|\boldsymbol{X}; \boldsymbol{\Theta}_\star)$ and $\boldsymbol{\Theta} \in \Omega$,

$$\begin{aligned}
\mathbb{E}[(y - \mu(\langle \boldsymbol{X}, \boldsymbol{\Theta}\rangle))^2] &\le 2\mathbb{E}[(y - \mu(\langle \boldsymbol{X}, \boldsymbol{\Theta}_\star\rangle))^2] + 2\mathbb{E}[(\mu(\langle \boldsymbol{X}, \boldsymbol{\Theta}_\star\rangle) - \mu(\langle \boldsymbol{X}, \boldsymbol{\Theta}\rangle))^2] \\
&\le 2g(\tau)\dot\mu(\langle \boldsymbol{X}, \boldsymbol{\Theta}_\star\rangle) + 2E^2 R_{\max}^2.
\end{aligned}$$

For notational simplicity, we introduce $\boldsymbol{A_X} := \mathrm{vec}^{-1}\left(\boldsymbol{H}(\pi; \boldsymbol{\Theta}_0)^{-1}\mathrm{vec}(\boldsymbol{X})\right)$. Then, we have

$$\begin{aligned}
\mathbb{E}[\widetilde{\boldsymbol{\Theta}}_t \widetilde{\boldsymbol{\Theta}}_t^\top] &= \mathbb{E}\left[(y_t - \mu(\langle \boldsymbol{X}_t, \boldsymbol{\Theta}_0\rangle))^2 \boldsymbol{A}_{\boldsymbol{X}_t} \boldsymbol{A}_{\boldsymbol{X}_t}^\top\right] \\
&= \mathbb{E}_{\boldsymbol{X} \sim \pi}\left[\mathbb{E}_{y \sim p(\cdot|\boldsymbol{X}; \boldsymbol{\Theta}_\star)}[(y - \mu(\langle \boldsymbol{X}, \boldsymbol{\Theta}_0\rangle))^2|\boldsymbol{X}]\boldsymbol{A_X}\boldsymbol{A_X}^\top\right] \\
&\preceq 2g(\tau)\mathbb{E}_{\boldsymbol{X} \sim \pi}\left[\dot\mu(\langle \boldsymbol{X}, \boldsymbol{\Theta}_\star\rangle)\boldsymbol{A_X}\boldsymbol{A_X}^\top\right] + 2E^2 R_{\max}^2 \mathbb{E}_{\boldsymbol{X} \sim \pi}\left[\boldsymbol{A_X}\boldsymbol{A_X}^\top\right] \\
&\preceq 2\left(g(\tau) + \frac{E^2 R_{\max}^2}{\kappa_\star}\right)\mathbb{E}_{\boldsymbol{X} \sim \pi}\left[\dot\mu(\langle \boldsymbol{X}, \boldsymbol{\Theta}_\star\rangle)\boldsymbol{A_X}\boldsymbol{A_X}^\top\right]. \qquad \text{(Recall } \kappa_\star = \min_{\boldsymbol{X} \in \mathcal{A}} \dot\mu(\langle \boldsymbol{X}, \boldsymbol{\Theta}_\star\rangle))
\end{aligned}$$

The proof then concludes by following the proof Lemma B.2 of Jang et al. (2024), which we provide here for completeness:

$$\begin{aligned}
&\left\|\mathbb{E}_{\boldsymbol{X} \sim \pi}\left[\dot\mu(\langle \boldsymbol{X}, \boldsymbol{\Theta}_\star\rangle)\boldsymbol{A_X}\boldsymbol{A_X}^\top\right]\right\|_{\mathrm{op}} \\
&= \max_{\boldsymbol{u} \in \mathcal{S}^{d_1-1}} \boldsymbol{u}^\top \mathbb{E}_{\boldsymbol{X} \sim \pi}\left[\dot\mu(\langle \boldsymbol{X}, \boldsymbol{\Theta}_\star\rangle)\boldsymbol{A_X}\boldsymbol{A_X}^\top\right]\boldsymbol{u} \\
&= \max_{\boldsymbol{u} \in \mathcal{S}^{d_1-1}} \boldsymbol{u}^\top \mathbb{E}_{\boldsymbol{X} \sim \pi}\left[\dot\mu(\langle \boldsymbol{X}, \boldsymbol{\Theta}_\star\rangle)\boldsymbol{A_X}\left(\sum_{m=1}^d \boldsymbol{e}_m \boldsymbol{e}_m^\top\right)\boldsymbol{A_X}^\top\right]\boldsymbol{u} \quad \text{(let } \{\boldsymbol{e}_m\}_{m \in [d]} \text{ be the standard basis vectors of } \mathbb{R}^d) \\
&= \max_{\boldsymbol{u} \in \mathcal{S}^{d_1-1}} \mathbb{E}_{\boldsymbol{X} \sim \pi}\left[\dot\mu(\langle \boldsymbol{X}, \boldsymbol{\Theta}_\star\rangle)\sum_{m=1}^d \left(\boldsymbol{u}^\top \boldsymbol{A_X}\boldsymbol{e}_m\right)^2\right] \\
&= \max_{\boldsymbol{u} \in \mathcal{S}^{d_1-1}} \mathbb{E}_{\boldsymbol{X} \sim \pi}\left[\dot\mu(\langle \boldsymbol{X}, \boldsymbol{\Theta}_\star\rangle)\sum_{m=1}^d \langle \boldsymbol{e}_m \otimes \boldsymbol{u}, \mathrm{vec}(\boldsymbol{A_X})\rangle^2\right] \quad (\boldsymbol{x}^\top \boldsymbol{A}\boldsymbol{y} = \langle \boldsymbol{y} \otimes \boldsymbol{x}, \mathrm{vec}(\boldsymbol{A})\rangle; \text{ Eqn. (40) of Minka (1997)}) \\
&= \max_{\boldsymbol{u} \in \mathcal{S}^{d_1-1}} \mathbb{E}_{\boldsymbol{X} \sim \pi}\left[\dot\mu(\langle \boldsymbol{X}, \boldsymbol{\Theta}_\star\rangle)\sum_{m=1}^d \langle \boldsymbol{e}_m \otimes \boldsymbol{u}, \boldsymbol{H}(\pi; \boldsymbol{\Theta}_0)^{-1}\mathrm{vec}(\boldsymbol{X})\rangle^2\right] \quad \text{(Definition of } \boldsymbol{A_X}) \\
&= \max_{\boldsymbol{u} \in \mathcal{S}^{d_1-1}} \sum_{m=1}^d (\boldsymbol{e}_m \otimes \boldsymbol{u})^\top \boldsymbol{H}(\pi; \boldsymbol{\Theta}_0)^{-1}\boldsymbol{H}(\pi; \boldsymbol{\Theta}_\star)\boldsymbol{H}(\pi; \boldsymbol{\Theta}_0)^{-1}(\boldsymbol{e}_m \otimes \boldsymbol{u}) \\
&\le (1 + 2R_s E)\max_{\boldsymbol{u} \in \mathcal{S}^{d_1-1}} \sum_{m=1}^d (\boldsymbol{e}_m \otimes \boldsymbol{u})^\top \boldsymbol{H}(\pi; \boldsymbol{\Theta}_0)^{-1}(\boldsymbol{e}_m \otimes \boldsymbol{u}) \quad \text{(Lemma D.3)} \\
&= (1 + 2R_s E)\max_{\boldsymbol{u} \in \mathcal{S}^{d_1-1}} \sum_{m=1}^d \boldsymbol{u}^\top\left([(\boldsymbol{H}(\pi; \boldsymbol{\Theta}_0)^{-1})_{jk}]_{j,k \in \{m+d_1(\ell-1):\ell \in [d_2]\}}\right)\boldsymbol{u} \\
&= (1 + 2R_s E)\underbrace{\lambda_{\max}\left([(\boldsymbol{H}(\pi; \boldsymbol{\Theta}_0)^{-1})_{jk}]_{j,k \in \{m+d_1(\ell-1):\ell \in [d_2]\}}\right)}_{=H^{(\mathrm{col})}(\pi; \boldsymbol{\Theta}_0)}.
\end{aligned}$$

All in all, we have that

$$\left\|\mathbb{E}[\widetilde{\boldsymbol{\Theta}}_t \widetilde{\boldsymbol{\Theta}}_t^\top]\right\|_{\mathrm{op}} \le \frac{1}{2}(1 + 2R_s E)\left(g(\tau) + \frac{E^2 R_{\max}^2}{\kappa_\star}\right)H^{(\mathrm{col})}(\pi; \boldsymbol{\Theta}_0). \tag{49}$$

Similarly, one can obtain

$$\left\| \mathbb{E}[\widetilde{\boldsymbol{\Theta}}_t^\top \widetilde{\boldsymbol{\Theta}}_t] \right\|_{\mathrm{op}} \leq \frac{1}{2}(1 + 2R_s E) \left( g(\tau) + \frac{E^2 R_{\max}^2}{\kappa_\star} \right) H^{(\mathrm{row})}(\pi; \boldsymbol{\Theta}_0), \tag{50}$$

and we are done. □

# E. Proof of Proposition 3.2 – **GL-LowPopArt** is Tighter than Nuclear Norm-Regularized Estimator

Here, we largely follow the proof strategies of Appendix C.2 and D.2 of Jang et al. (2024), but with some differences due to the heterogeneity caused by $\dot{\mu}$'s.

### E.1. Upper Bound of $\mathrm{GL}(\pi)$

We have that

$$
\begin{aligned}
H^{(\mathrm{col})}(\pi) &= \lambda_{\max}\left(\sum_{m=1}^{d_2} \boldsymbol{D}_m{}^{(\mathrm{col})}(\pi)\right) \\
&\leq \sum_{m=1}^{d_2} \lambda_{\max}\left(\boldsymbol{D}_m{}^{(col)}(\pi)\right) &&(\lambda_{\max} \text{ is convex and 1-homogenous}) \\
&= \sum_{m=1}^{d_2} \max_{\boldsymbol{u}\in\mathcal{S}^{d_1-1}} \boldsymbol{u}^\top \boldsymbol{D}_m{}^{(\mathrm{col})}(\pi)\boldsymbol{u} \\
&= \sum_{m=1}^{d_2} \max_{\boldsymbol{u}\in\mathcal{S}^{d_1-1}} (\boldsymbol{e}_m\otimes\boldsymbol{u})\top \boldsymbol{H}(\pi;\boldsymbol{\Theta}_0)^{-1}(\boldsymbol{e}_m\otimes\boldsymbol{u}) &&(\text{see proof of Lemma D.2}) \\
&\leq \sum_{m=1}^{d_2} \max_{\boldsymbol{u}\in\mathcal{S}^{d_1 d_2-1}} \boldsymbol{u}^\top \boldsymbol{H}(\pi;\boldsymbol{\Theta}_0)^{-1}\boldsymbol{u} \\
&= d_2\lambda_{\max}(\boldsymbol{H}(\pi;\boldsymbol{\Theta}_0)^{-1}) \\
&= \frac{d_2}{\lambda_{\min}(\boldsymbol{H}(\pi;\boldsymbol{\Theta}_0))} \\
&\leq \frac{d_2(1+R_s)}{\lambda_{\min}(\boldsymbol{H}(\pi;\boldsymbol{\Theta}_\star))}.
\end{aligned}
$$

One can similarly prove that $H^{(\mathrm{row})}(\pi) \leq \frac{d_1(1+R_s)}{\lambda_{\min}(\boldsymbol{H}(\pi;\boldsymbol{\Theta}_\star))}$, and the desired conclusion follows.

### E.2. Lower Bound of $\mathrm{GL}(\pi)$

We first consider the case of $\boldsymbol{X}\in\mathcal{B}_{\mathrm{op}}^{d_1\times d_2}(1)$.

Again, by definition,

$$
\begin{aligned}
\mathrm{GL}(\pi) &\geq \lambda_{\max}\left(\sum_{m=1}^{d_2}[(\boldsymbol{H}(\pi;\boldsymbol{\Theta}_0)^{-1})_{jk}]_{j,k\in\{\ell+d_1(m-1):\ell\in[d_1]\}}\right) \\
&\geq \frac{1}{d_1}\mathrm{tr}\left(\sum_{m=1}^{d_2}[(\boldsymbol{H}(\pi;\boldsymbol{\Theta}_0)^{-1})_{jk}]_{j,k\in\{\ell+d_1(m-1):\ell\in[d_1]\}}\right) &&(\lambda_{\max}(\boldsymbol{A})\geq\tfrac{1}{d}\mathrm{tr}(\boldsymbol{A}) \text{ for any symmetric } \boldsymbol{A}\in\mathbb{R}^{d\times d}) \\
&= \frac{1}{d_1}\mathrm{tr}\left(\boldsymbol{H}(\pi;\boldsymbol{\Theta}_0)^{-1}\right) \\
&\geq \frac{1}{d_1}\frac{(d_1 d_2)^2}{\mathrm{tr}\left(\boldsymbol{H}(\pi;\boldsymbol{\Theta}_0)\right)}, &&(\text{AM-HM inequality on the eigenvalues of } \boldsymbol{H}(\pi;\boldsymbol{\Theta}_0))
\end{aligned}
$$

and similarly,

$$
\mathrm{GL}(\pi) \geq \frac{1}{d_2}\frac{(d_1 d_2)^2}{\mathrm{tr}\left(\boldsymbol{H}(\pi;\boldsymbol{\Theta}_0)\right)},
$$

i.e., $\mathrm{GL}(\pi) \geq \frac{(d_1 d_2)^2}{(d_1\wedge d_2)\mathrm{tr}(\boldsymbol{H}(\pi;\boldsymbol{\Theta}_0))}$.

Now note that

$$
\mathrm{tr}\left(\boldsymbol{H}(\pi;\boldsymbol{\Theta}_0)\right) \leq (1+R_s)\mathrm{tr}\left(\boldsymbol{H}(\pi;\boldsymbol{\Theta}_\star)\right) \qquad\qquad (\text{Lemma D.3})
$$

$$= (1 + R_s)\mathbb{E}_{\boldsymbol{X} \sim \pi} \left[ \dot{\mu}(\langle \boldsymbol{X}, \boldsymbol{\Theta}_\star \rangle) \mathrm{tr}(\mathrm{vec}(\boldsymbol{X})\mathrm{vec}(\boldsymbol{X})^\top) \right]$$

$$= (1 + R_s)\mathbb{E}_{\boldsymbol{X} \sim \pi} \left[ \dot{\mu}(\langle \boldsymbol{X}, \boldsymbol{\Theta}_\star \rangle) \|\boldsymbol{X}\|_F^2 \right]$$

$$\leq (1 + R_s)(d_1 \wedge d_2)\mathbb{E}_{\boldsymbol{X} \sim \pi} \left[ \dot{\mu}(\langle \boldsymbol{X}, \boldsymbol{\Theta}_\star \rangle) \right] \qquad (\boldsymbol{X} \in \mathcal{B}_{\mathrm{op}}^{d_1 \times d_2}(1) \Rightarrow \boldsymbol{X} \in \mathcal{B}_F^{d_1 \times d_2}(\sqrt{d_1 \wedge d_2}))$$

$$= (1 + R_s)(d_1 \wedge d_2)\overline{\kappa}(\pi).$$

Chaining the above two inequalities gives $\mathrm{GL}(\pi) \geq \frac{(d_1 d_2)^2}{(1+R_s)(d_1 \wedge d_2)^2 \overline{\kappa}(\pi)} = \frac{(d_1 \vee d_2)^2}{(1+R_s)\overline{\kappa}(\pi)}$.

From the above proof, it is clear that when $\boldsymbol{X} \in \mathcal{B}_F^{d_1 \times d_2}(1)$, we can shave off an extra $d_1 \wedge d_2$ from the denominator, leading to the desired conclusion.

$\square$

## F. Comparing with Kang et al. (2022)

### F.1. Overview

For the comparison, we assume that the underlying GLM is 1-subGaussian, which adds an extra factor of $d_1 \wedge d_2$ for our Stage I guarantee (see $f(\delta, d_1, d_2)$ in our Lemma C.4). In Table 2, we provide the complete comparison of $\left\| \widehat{\Theta}_0 - \Theta_\star \right\|_F^2$, for our results (Stage I and Stage I + II) vs. the results of Kang et al. (2022). We consider three arm-sets: unit Frobenius/operator norm balls, and $\mathcal{X} := \{ e_i (e_j')^\top : (i, j) \in [d_1] \times [d_2] \}$, the matrix completion basis.

| | $\mathcal{A} = \mathcal{B}_F^{d_1 \times d_2}(1)$ | $\mathcal{A} = \mathcal{B}_{\mathrm{op}}^{d_1 \times d_2}(1)$ | $\mathcal{A} = \mathcal{X}$ | Limitations |
|---|---|---|---|---|
| Theorem 4.1 Kang et al. (2022) | $\frac{(d_1 \vee d_2) d_1 d_2 r}{\overline{\kappa}(\pi)^2 N}$ | $\frac{(d_1 \vee d_2)^3 r}{\overline{\kappa}(\pi)^2 N}$ | N/A | $\pi \in \mathcal{P}(\mathcal{A})$ must have a continuously differentiable density with $\mathrm{supp}(\pi) = \mathbb{R}^{d_1 \times d_2}$. |
| Theorem J.4 Kang et al. (2022) | $\frac{(d_1 \vee d_2) d_1 d_2 r}{c_\mu^2 N}$ | $\frac{(d_1 \vee d_2)^2 r}{c_\mu^2 N}$ | $\frac{(d_1 \vee d_2)(d_1 d_2)^4 r}{c_\mu^2 N}$ | Requires subGaussianity of $\mathrm{vec}(X)$'s for $X \sim \pi$, $c_\mu \ll \kappa_\star$ |
| Stage I Our Theorem 3.4 | $\frac{(d_1 \wedge d_2)(d_1 d_2)^2 r}{\kappa_\star^2 N}$ | $\frac{(d_1 \vee d_2) d_1 d_2 r}{\kappa_\star^2 N}$ | $\frac{(d_1 \vee d_2) d_1 d_2 r}{\kappa_\star^2 N}$ | |
| Stage I + II Our Theorem 3.1 | $\frac{\mathrm{GL}_{\min} r}{N} \lesssim \frac{(d_1 \vee d_2) d_1 d_2 r}{\kappa_\star N}$ | $\frac{\mathrm{GL}_{\min} r}{N} \lesssim \frac{(d_1 \vee d_2)^2 r}{\kappa_\star N}$ | $\frac{\mathrm{GL}_{\min} r}{N} \lesssim \frac{(d_1 \vee d_2)^2 r}{\kappa_\star N}$ | |

*Table 2.* Here, we only consider the dependencies on the rank $r$, dimensions $d_1, d_2$, sample size $N$, and curvature-dependent quantities $c_\mu$ and $\kappa_\star$. All the other factors, including polylog factors, are ignored. (row 4) For a clear and fair comparison, we also write the upper bound on $\mathrm{GL}_{\min}(\mathcal{A})$ as proved in Proposition 3.2.

### F.2. Their Theorem 4.1 – Stein's Lemma-based Estimator (row 1)

Their first estimator achieves the following error bound (Kang et al., 2022, Theorem 4.1)

$$\left\| \widehat{\Theta}^{\mathrm{Kang},1} - \Theta_\star \right\|_F^2 \lesssim \frac{M(\pi)(d_1 \vee d_2) r}{\overline{\kappa}(\pi)^2 N}, \tag{51}$$

*given* that $\pi$ has a continuously differentiable density supported over $\mathbb{R}^d$. This is because they rely on the generalized Stein's lemma (Stein et al., 2004, Proposition 1.4) This limits their applicability to discrete arm-sets, while our framework is applicable for both continuous and discrete arm-sets. Also, from the perspective of optimal experimental design, it is not clear how to optimize their bound for $\pi$ while satisfying the conditions above. Even without those conditions, the function $\pi \mapsto \frac{M(\pi)}{\overline{\kappa}(\pi)}$ is likely to be nonconvex. On the other hand, we mention that their result is applicable to the general single index model of the form $y_t = \mu(\langle X_t, \Theta_\star \rangle) + \eta_t$ for some subGaussian noise $\eta_t$.

Here, $M(\pi)$ is a quantity related to the variance of the score function of $\pi$ that often scales with the dimension. For $\mathcal{A} = \mathcal{B}_F^{d_1 \times d_2}(1)$ and $\pi \sim \mathcal{N}(0, \frac{c}{d_1 d_2 \log T} I)$ for a constant $c > 0$, it can be computed that $M(\pi) \lesssim d_1 d_2$ (Jang et al., 2024, Appendix H.2), which is what we use in the Table. For the other arm-sets, we set $M \lesssim (d_1 \vee d_2)^2$ as suggested by Kang et al. (2022).

### F.3. Their Theorem J.4 – Nuclear Norm-regularized Estimator (row 2)

Their second estimator, which is exactly the nuclear norm-regularized estimator, achieves the following error bound (Kang et al., 2022, Theorem J.4):

$$\left\| \widehat{\Theta}^{\mathrm{Kang},2} - \Theta_\star \right\|_F^2 \lesssim \frac{(d_1 \vee d_2) r \sigma(\pi)^2}{c_\mu^2 \lambda_{\min}(V(\pi))^4 N}, \tag{52}$$

*given* that the following assumptions hold:

**Assumption J.1.** $\pi \in \mathcal{P}(\mathcal{A})$ is such that $\text{vec}(\boldsymbol{X})$ is $\sigma(\pi)$-subGaussian[9] for $\boldsymbol{X} \sim \pi$.

**Assumption J.2.** There is two (dimension-independent) constants $S_2 \leq S$ such that $\mathcal{A} \subseteq \mathcal{B}^{d_1 \times d_2} \triangleq \mathcal{B}_F^{d_1 \times d_2}(S) \cap \mathcal{B}_{\text{op}}^{d_1 \times d_2}(S_2)$ and likewise for $\boldsymbol{\Theta}_\star$.

**Assumption J.3.** There is a constant $c_2 > 0$ such that

$$c_\mu := \min \left( \inf_{\boldsymbol{X} \in \mathcal{A}, \boldsymbol{\Theta} \in \mathcal{B}^{d_1 \times d_2}} \dot{\mu}(\langle \boldsymbol{X}, \boldsymbol{\Theta} \rangle), \inf_{|z| \leq (S+2)\sigma c_2} \dot{\mu}(z) \right) > 0. \tag{53}$$

Kang et al. (2022) assumed that $\lambda_{\min}(\boldsymbol{V}(\pi)) \asymp \sigma(\pi)^2 \asymp \frac{1}{d_1 d_2}$, which was also the assumption made by Lu et al. (2021, Assumption 2). Indeed, as argued by the two works, one can easily find $\pi$ that satisfies the above conditions, e.g. $\text{Unif}(\mathcal{B}_F^{d_1 \times d_2}(1))$ or require for "the convex hull of a subset of arms to contain a ball with radius $R \leq 1$ that does not scale with $d_1$ or $d_2$." *But*, similar to the previous subsection, it is unclear how to optimize for $\pi$ in the optimal experimental design setup. Moreover, the above assumption may fail even for a simple arm-set. Consider $\mathcal{X} = \{\boldsymbol{e}_i (\boldsymbol{e}'_j)^\top : 1 \leq i \leq d_1, 1 \leq j \leq d_2\}$ and $\pi \sim \text{Unif}(\mathcal{X})$. Then, one can show that $\lambda_{\min}(V(\pi)) = \frac{1}{d_1 d_2}$ while $\sigma(\pi)^2 = 1$, leading to a suboptimal guarantee as shown in Table 2. Another point is that their curvature-dependent quantity is $c_\mu$, which, by definition, may be much smaller than our $\kappa$. Roughly speaking, $c_\mu$ is a globally worst-case curvature, while $\kappa$ is the worst-case curvature at the specific instance $\boldsymbol{\Theta}_\star$.

Still, note that for $\mathcal{B}_F^{d_1 \times d_2}(1)$ and $\mathcal{B}_{\text{op}}^{d_1 \times d_2}(1)$, even when utilizing uniform distribution, their result is better than our Stage I guarantees by a factor of $d_1 \wedge d_2$. This difference is mainly from utilizing a different proof technique, involving truncation and peeling technique (Raskutti et al., 2010) (Wainwright, 2018, Theorem 10.17), which is distinct from our proof of Stage I and of Fan et al. (2019).

Lastly, we mention that our GL-LowPopArt improves upon all the aforementioned guarantees, showing the effectiveness of the Catoni-style estimator (Catoni, 2012; Minsker, 2018) and the tightness of our theoretical analysis.

---

[9]This means that for any unit vector $\boldsymbol{u} \in \mathcal{S}^{d_1 d_2 - 1}$, $\boldsymbol{u}^\top \text{vec}(\boldsymbol{X})$ is $\sigma(\pi)$-subGaussian.

# G. Proof of Theorem 4.1 – Local Minimax Lower Bound

WLOG assume that $d_1 = \max(d_1, d_2)$. For given $\boldsymbol{\Theta}_\star$, let $\boldsymbol{UDV}^\top$ be its SVD.

Inspired by Rohde & Tsybakov (2011, Theorem 5) and Abeille et al. (2021, Theorem 2), we consider the following set of $d_1 \times d_2$ matrices:

$$\Theta_{r,\varepsilon,\beta} := \left\{ (1-\varepsilon)\boldsymbol{\Theta}_\star + \varepsilon \boldsymbol{U}'\boldsymbol{V}^\top \in \mathbb{R}^{d_1 \times d_2} : \boldsymbol{U}' \in \{0, \beta\}^{d_1 \times r} \right\}, \tag{54}$$

where $\varepsilon \in (0, 1)$ and $\beta > 0$ will be specified later. By construction, we have that for any $\boldsymbol{\Theta} \in \Theta_{r,\varepsilon,\beta}$, $\mathrm{rank}(\boldsymbol{\Theta}) \leq r$ and

$$\begin{aligned}
\|\boldsymbol{\Theta}\|_{\mathrm{nuc}} &\leq (1-\varepsilon)\|\boldsymbol{\Theta}_\star\|_{\mathrm{nuc}} + \varepsilon\|\boldsymbol{U}'\boldsymbol{V}^\top\|_{\mathrm{nuc}} \\
&= (1-\varepsilon)S_* + \varepsilon\|\boldsymbol{U}'\|_{\mathrm{nuc}} && \text{(unitary invariance of } \|\cdot\|_{\mathrm{nuc}}) \\
&\leq (1-\varepsilon)S_* + \varepsilon\sqrt{r}\|\boldsymbol{U}'\|_F && \text{(Cauchy-Schwartz inequality on the singular values of } \boldsymbol{U}') \\
&\leq (1-\varepsilon)S_* + \varepsilon\beta r\sqrt{d_1}. && \text{(by construction)}
\end{aligned}$$

Thus, it can be verified that $\beta \leq \frac{S_*}{r\sqrt{d_1}}$ implies $\|\boldsymbol{\Theta}\|_{\mathrm{nuc}} \leq S_*$, i.e., $\Theta_{r,\varepsilon,\beta} \subset \mathcal{N}(\boldsymbol{\Theta}_\star; \varepsilon, r, S_*)$.

By construction, $\|\boldsymbol{\Theta}_1 - \boldsymbol{\Theta}_2\|_F^2$ is closely related to the Hamming distance of the $\mathrm{vec}(\boldsymbol{U}')$'s, which are basically binary sequences. With this, we recall the Gilbert-Varshamov bound:

**Lemma G.1** (Gilbert–Varshamov bound; Lemma 2.9 of Tsybakov (2009); Theorem 1 of Gilbert (1952); Varshamov (1964)). *Let $m \geq 8$ and $\Omega := \{0, 1\}^m$. Then there exists $\{\omega^{(0)}, \omega^{(1)}, \cdots, \omega^{(M)}\} \subset \Omega$ with $M \geq 2^{m/8}$ such that $\omega^{(0)} = (0, \cdots, 0)$ and*

$$d_H(\omega^{(j)}, \omega^{(k)}) := \sum_{\ell=1}^m \mathbb{1}[(\omega^{(j)})_\ell \neq (\omega^{(k)})_\ell] \geq \frac{m}{8}, \quad \forall 0 \leq j < k \leq M. \tag{55}$$

Thus, we can find a $\Theta_{r,\varepsilon,\beta}^0 \subset \Theta_{r,\varepsilon,\beta}$ such that $|\Theta_{r,\varepsilon,\beta}^0| \geq 2^{\frac{rd_1}{8}}$, and for any $\boldsymbol{\Theta}_i = (1-\varepsilon)\boldsymbol{\Theta}_\star + \varepsilon\boldsymbol{U}_i'\boldsymbol{DV}^\top \in \Theta_{r,\varepsilon,\beta}^0$ with $i \in \{1, 2\}$ and $\boldsymbol{U}_1 \neq \boldsymbol{U}_2$,

$$\|\boldsymbol{\Theta}_1 - \boldsymbol{\Theta}_2\|_F^2 = \varepsilon^2 \|(\boldsymbol{U}_1' - \boldsymbol{U}_2')\boldsymbol{V}^\top\|_F^2 = \varepsilon^2 \|(\boldsymbol{U}_1' - \boldsymbol{U}_2')\|_F^2 \geq \varepsilon^2 \frac{\beta^2 rd_1}{8}, \tag{56}$$

where we denote $\sigma_{\min} = \sigma_{\min}(\boldsymbol{\Theta}_\star)$ to be the minimum non-zero singular value of $\boldsymbol{\Theta}_\star$.

Furthermore, we have that for any $\boldsymbol{\Theta} = (1-\varepsilon)\boldsymbol{\Theta}_\star + \varepsilon\boldsymbol{U}'\boldsymbol{V}^\top \in \Theta_{r,\varepsilon,\beta}^0$,

$$\begin{aligned}
\left\|\boldsymbol{\Theta}_\star - \left((1-\varepsilon)\boldsymbol{\Theta}_\star + \varepsilon\boldsymbol{U}'\boldsymbol{V}^\top\right)\right\|_F^2 &= \varepsilon^2 \left\|\boldsymbol{\Theta}_\star - \boldsymbol{U}'\boldsymbol{V}^\top\right\|_F^2 \\
&\geq \varepsilon^2 \left(\|\boldsymbol{\Theta}_\star\|_F^2 - \|\boldsymbol{U}'\|_F^2\right) && \text{(triangle inequality and unitary invariance of } \|\cdot\|_F) \\
&\geq \varepsilon^2 \left(\|\boldsymbol{\Theta}_\star\|_F^2 - \beta^2 rd_1\right) && \text{(by construction)} \\
&\geq \varepsilon^2 \frac{\beta^2 rd_1}{8},
\end{aligned}$$

which in turn holds when $\|\boldsymbol{\Theta}_\star\|_F^2 \geq \frac{9\beta^2 rd_1}{8}$. We will see that this indeed holds with our $\beta$ specified later.

For $\boldsymbol{\Theta} \in \mathbb{R}^{d_1 \times d_2}$, let $\mathbb{P}_{\boldsymbol{\Theta}}$ be the probability distribution of the observations $\{(\boldsymbol{X}_t, y_t)\}_{t \in [N]}$, with $y_t \sim p(\cdot|\boldsymbol{X}_t; \boldsymbol{\Theta})$.

We now compute the KL between $\mathbb{P}_{(1-\varepsilon)\boldsymbol{\Theta}_\star + \varepsilon\boldsymbol{\Theta}'}$ and $\mathbb{P}_{\boldsymbol{\Theta}_\star}$ for any $\boldsymbol{\Theta}' = \boldsymbol{U}'\boldsymbol{V}^\top \in \Theta_{r,\varepsilon,\beta}$ by connecting it with the Bregman divergence:

**Definition G.2.** For a $m : \mathbb{R} \to \mathbb{R}$, the **Bregman divergence** $D_m(\cdot, \cdot)$ is defined as follows:

$$D_m(z_1, z_2) := m(z_1) - m(z_2) - m'(z_2)(z_1 - z_2).$$

We recall the following well-known lemma from information geometry, which simplifies the computation of KL between two GLMs by implicitly making use of their dually flat structure (Amari, 2016; Nielsen, 2020; Brekelmans et al., 2020):

**Lemma G.3.** *Consider two GLMs $p_1 \triangleq p(\cdot|\boldsymbol{X};\boldsymbol{\Theta}_1)$ and $p_2 \triangleq p(\cdot|\boldsymbol{X};\boldsymbol{\Theta}_2)$ with the same log-partition function $m$. Then, we have that $D_{\mathrm{KL}}(p_2, p_1|\boldsymbol{X}) = D_m(\langle\boldsymbol{X},\boldsymbol{\Theta}_1\rangle, \langle\boldsymbol{X},\boldsymbol{\Theta}_2\rangle)$.*

We then have that

$$D_{\mathrm{KL}}(\mathbb{P}_{(1-\varepsilon)\boldsymbol{\Theta}_\star+\varepsilon\boldsymbol{\Theta}'}, \mathbb{P}_{\boldsymbol{\Theta}_\star}|\boldsymbol{X}) = \frac{1}{g(\tau)} D_m(\langle\boldsymbol{X},\boldsymbol{\Theta}_\star\rangle, (1-\varepsilon)\langle\boldsymbol{X},\boldsymbol{\Theta}_\star\rangle + \varepsilon\langle\boldsymbol{X},\boldsymbol{\Theta}'\rangle)$$

$$= \frac{1}{g(\tau)}\varepsilon^2\langle\boldsymbol{X},\boldsymbol{\Theta}_\star-\boldsymbol{\Theta}'\rangle^2\int_0^1 v\dot{\mu}(\langle\boldsymbol{X},\boldsymbol{\Theta}_\star\rangle + \varepsilon\langle\boldsymbol{X},\boldsymbol{\Theta}'-\boldsymbol{\Theta}_\star\rangle v)dv.$$

(Taylor expansion with integral remainder)

We recall a useful self-concordance control lemma from Abeille et al. (2021); Faury et al. (2020):

**Lemma G.4** (A Modification of Lemma 9 of Abeille et al. (2021)). *Let $\mu : \mathbb{R} \to \mathbb{R}$ be a strictly increasing function satisfying $|\ddot{\mu}| \leq R_s\dot{\mu}$ for some $R_s \geq 0$. Then, for any $z_1, z_2 \in \mathbb{R}$ and $\varepsilon > 0$, $\dot{\mu}(z_1 + \varepsilon z_2) \leq \dot{\mu}(z_1)\exp(R_s\varepsilon|z_2|)$.*

With this, we have that

$$D_{\mathrm{KL}}(\mathbb{P}_{(1-\varepsilon)\boldsymbol{\Theta}_\star+\varepsilon\boldsymbol{\Theta}'}, \mathbb{P}_{\boldsymbol{\Theta}_\star}|\boldsymbol{X}) \leq \frac{1}{g(\tau)}\varepsilon^2\dot{\mu}(\langle\boldsymbol{X},\boldsymbol{\Theta}_\star\rangle)\langle\boldsymbol{X},\boldsymbol{\Theta}_\star-\boldsymbol{\Theta}'\rangle^2\int_0^1 v\exp(R_s\varepsilon|\langle\boldsymbol{X},\boldsymbol{\Theta}'-\boldsymbol{\Theta}_\star\rangle|v)dv$$

$$\leq \frac{1}{2g(\tau)}\varepsilon^2\dot{\mu}(\langle\boldsymbol{X},\boldsymbol{\Theta}_\star\rangle)\langle\boldsymbol{X},\boldsymbol{\Theta}_\star-\boldsymbol{\Theta}'\rangle^2\exp(R_s\varepsilon|\langle\boldsymbol{X},\boldsymbol{\Theta}'-\boldsymbol{\Theta}_\star\rangle|)$$

$$\overset{(*)}{\leq} \frac{1}{2g(\tau)}\varepsilon^2\dot{\mu}(\langle\boldsymbol{X},\boldsymbol{\Theta}_\star\rangle)\langle\boldsymbol{X},\boldsymbol{\Theta}_\star-\boldsymbol{\Theta}'\rangle^2\exp\left(R_s\varepsilon(1+\beta\sqrt{d_1 r})S_*\right)$$

$$\leq \frac{e}{2g(\tau)}\varepsilon^2\dot{\mu}(\langle\boldsymbol{X},\boldsymbol{\Theta}_\star\rangle)\langle\boldsymbol{X},\boldsymbol{\Theta}_\star-\boldsymbol{\Theta}'\rangle^2,$$

*given* that $R_s\varepsilon(1+\beta\sqrt{d_1 r})S_* \leq 1$. Note that $(*)$ holds regardless of whether we assume $\mathcal{A} \subseteq \mathcal{B}_F^{d_1\times d_2}(1)$ (which is what we assume in the statement) or $\mathcal{A} \subseteq \mathcal{B}_{\mathrm{op}}^{d_1\times d_2}(1)$ (which is implied from the first case). To see this, if the first case holds, then

$$\langle\boldsymbol{X},\boldsymbol{\Theta}_\star-\boldsymbol{\Theta}'\rangle \leq \|\boldsymbol{X}\|_F\|\boldsymbol{\Theta}-\boldsymbol{\Theta}_\star\|_F \leq \|\boldsymbol{\Theta}-\boldsymbol{\Theta}_\star\|_{\mathrm{nuc}} \leq (1+\beta\sqrt{d_1 r})S_*,$$

and if the second case holds,

$$\langle\boldsymbol{X},\boldsymbol{\Theta}_\star-\boldsymbol{\Theta}'\rangle \leq \|\boldsymbol{X}\|_{\mathrm{op}}\|\boldsymbol{\Theta}-\boldsymbol{\Theta}_\star\|_{\mathrm{nuc}} \leq (1+\beta\sqrt{d_1 r})S_*.$$

**Remark 11.** *Lee et al. (2024b, Lemma 4) has utilized a similar argument (Taylor integral remainder with self-concordance) to provide a lower bound on the KL divergence during the online learning regret analysis. However, they restricted their attention to the Bernoulli distribution.*

Thus,

$$D_{\mathrm{KL}}(\mathbb{P}_{(1-\varepsilon)\boldsymbol{\Theta}_\star+\varepsilon\boldsymbol{\Theta}'}, \mathbb{P}_{\boldsymbol{\Theta}_\star}) = N\mathbb{E}_{\boldsymbol{X}\sim\pi}[D_{\mathrm{KL}}(\mathbb{P}_{(1-\varepsilon)\boldsymbol{\Theta}_\star+\varepsilon\boldsymbol{\Theta}'}, \mathbb{P}_{\boldsymbol{\Theta}_\star}|\boldsymbol{X})]$$

$$\leq \frac{eN}{2g(\tau)}\varepsilon^2\mathrm{vec}(\boldsymbol{\Theta}_\star-\boldsymbol{\Theta}')^\top\boldsymbol{H}(\pi;\boldsymbol{\Theta}_\star)\mathrm{vec}(\boldsymbol{\Theta}_\star-\boldsymbol{\Theta}')$$

$$\leq \frac{eN}{2g(\tau)}\varepsilon^2\lambda_{\max}(\boldsymbol{H}(\pi;\boldsymbol{\Theta}_\star))\|\boldsymbol{\Theta}_\star-\boldsymbol{\Theta}'\|_F^2$$

$$\leq \frac{eN}{2g(\tau)}\varepsilon^2\lambda_{\max}(\boldsymbol{H}(\pi;\boldsymbol{\Theta}_\star))(1+\beta\sqrt{d_1 r})^2 S_*^2.$$

Then we have that

$$\frac{1}{|\Theta_{r,\varepsilon}^0|}\sum_{\boldsymbol{\Theta}'\in\Theta_{r,\varepsilon}^0} D_{\mathrm{KL}}(\mathbb{P}_{\boldsymbol{\Theta}'}, \mathbb{P}_{\boldsymbol{\Theta}_\star}) \leq \frac{e\varepsilon^2 N\lambda_{\max}(\boldsymbol{H}(\pi;\boldsymbol{\Theta}_\star))(1+\beta\sqrt{d_1 r})^2 S_*^2}{2g(\tau)}$$

$$= \frac{4eN\varepsilon^2\lambda_{\max}(\boldsymbol{H}(\pi;\boldsymbol{\Theta}_\star))(1+\beta\sqrt{d_1 r})^2 S_*^2}{g(\tau)rd_1}\frac{rd_1}{8}.$$

As $\log |\Theta^0_{r,\varepsilon,\beta}| \geq \log(2^{\frac{rd_1}{8}}) = \frac{rd_1}{8} \log 2$,

$$\frac{1}{|\Theta^0_{r,\varepsilon,\beta}|} \sum_{\Theta' \in \Theta^0_{r,\varepsilon,\beta}} D_{\mathrm{KL}}(\mathbb{P}_{\Theta'}, \mathbb{P}_{\Theta_\star}) \leq \frac{1}{16} \log |\Theta^0_{r,\varepsilon}|$$

holds with $\varepsilon^2 \leq \frac{rd_1 g(\tau) \alpha \log 2}{2^6 e N \lambda_{\max}(\boldsymbol{H}(\pi;\Theta_\star))(1+\beta\sqrt{d_1}r)^2 S_*^2}$ where $\alpha = \frac{1}{16}$.

We choose

$$\beta^2 = \frac{\gamma}{rd_1} \Rightarrow \varepsilon^2 = \frac{\alpha \log 2}{2^6 e (1+\sqrt{\gamma})^2} \frac{rd_1 g(\tau)}{N \lambda_{\max}(\boldsymbol{H}(\pi;\Theta_\star))S_*^2}. \tag{57}$$

We now check the requirements:

$$\beta \leq \frac{S_*}{r\sqrt{d_1}} \iff \gamma \leq \frac{S_*^2}{r} \tag{58}$$

$$\|\Theta_\star\|_F^2 \geq \frac{9\beta^2 rd_1}{8} \iff \gamma \leq \frac{8}{9} \|\Theta_\star\|_F^2 \tag{59}$$

$$R_s \varepsilon (1 + \beta\sqrt{d_1}r)S_* \leq 1 \iff N \geq \frac{R_s^2}{2^{10}} \frac{\log 2}{e} \frac{rd_1 g(\tau)}{\lambda_{\max}(\boldsymbol{H}(\pi;\Theta_\star))}. \tag{60}$$

The proof concludes by invoking Tsybakov (2009, Theorem 2.5) with $\alpha = \frac{1}{16}$,[10] which we recall here for completeness:

**Lemma G.5** (Theorem 2.5 of Tsybakov (2009)). *Let $\Theta$ be a subset of a metric space with metric $d(\cdot,\cdot)$, and let $\theta \mapsto \mathbb{P}_\theta$ be the probability measure parametrized by $\theta$. Suppose that there exists $\{\theta_0, \theta_1, \cdots, \theta_M\} \subset \Theta$ for some $M \geq 2$ such that*

*(i) $d(\theta_j, \theta_k) \geq 2b > 0, \quad \forall 0 \leq j < k \leq M$,*

*(ii) $\mathbb{P}_{\theta_j} \ll \mathbb{P}_{\theta_0}, \quad \forall j = 1, 2, \cdots, M$, and*

*(iii) there exists a $\alpha \in (0, 1/8)$ such that $\frac{1}{M} \sum_{j=1}^M D_{\mathrm{KL}}(\mathbb{P}_{\theta_j}, \mathbb{P}_{\theta_0}) \leq \alpha \log M$.*

*Then, we have the following high-probability minimax lower bound:*

$$\inf_{\widehat{\theta}} \sup_{\theta_\star \in \Theta} \mathbb{P}_{\theta_\star}(d(\widehat{\theta}, \theta_\star) \geq b) \geq \frac{\sqrt{M}}{1 + \sqrt{M}} \left(1 - 2\alpha - \sqrt{\frac{2\alpha}{\log M}}\right) > 0. \tag{61}$$

$\square$

We now provide the proofs of the missing lemmas:

*Proof of Lemma G.3.* This follows from brute-force computation:

$$D_{\mathrm{KL}}(p_2, p_1) = \mathbb{E}_{y \sim p_2} \left[\log \frac{p_2(y)}{p_1(y)}\right]$$

$$= \frac{1}{g(\tau)} \mathbb{E}_{y \sim p_2} [y\langle \boldsymbol{X}, \Theta_2 - \Theta_1 \rangle + m(\langle \boldsymbol{X}, \Theta_1 \rangle) - m(\langle \boldsymbol{X}, \Theta_2 \rangle)] \quad \text{(recall the probability density of GLMs)}$$

$$= \frac{m(\langle \boldsymbol{X}, \Theta_1 \rangle) - m(\langle \boldsymbol{X}, \Theta_2 \rangle) - m'(\langle \boldsymbol{X}, \Theta_2 \rangle)\langle \boldsymbol{X}, \Theta_1 - \Theta_2 \rangle}{g(\tau)} \quad (\mathbb{E}[y] = m'(\langle \boldsymbol{X}, \Theta_2 \rangle))$$

$$= \frac{1}{g(\tau)} D_m(\langle \boldsymbol{X}, \Theta_1 \rangle, \langle \boldsymbol{X}, \Theta_2 \rangle).$$

$\square$

---

[10] No efforts were made to optimize the constants.

*Proof of Lemma G.4.* We provide the slightly modified proof of Abeille et al. (2021, Lemma 9) for completeness.

Starting from the self-concordance, we have that for any $z_1, z_2 \in \mathbb{R}$

$$-R_s \leq \frac{\ddot{\mu}(z)}{\dot{\mu}(z)} \leq R_s, \quad \forall z \in \mathbb{R} \implies -R_s \varepsilon |z_2| \leq \underbrace{\int_{(z_1 + \varepsilon z_2) \wedge z_1}^{\dot{\mu}(z_1 + \varepsilon z_2) \vee z_1} \frac{\ddot{\mu}(z)}{\dot{\mu}(z)} dz}_{= \log \frac{\dot{\mu}((z_1 + \varepsilon z_2) \vee z_1)}{\dot{\mu}((z_1 + \varepsilon z_2) \wedge z_1)}} \leq R_s \varepsilon |z_2|.$$

If $z_2 \geq 0$, then we have that from the upper bound,

$$\dot{\mu}(z_1 + \varepsilon z_2) \leq \dot{\mu}(z_1) \exp(R_s \varepsilon z_2) = \dot{\mu}(z_1) \exp(R_s \varepsilon |z_2|).$$

If $z_2 < 0$, then we have that from the lower bound,

$$\dot{\mu}(z_1 + \varepsilon z_2) \exp(R_s \varepsilon z_2) \leq \dot{\mu}(z_1) \implies \dot{\mu}(z_1 + \varepsilon z_2) \leq \dot{\mu}(z_1) \exp(-R_s \varepsilon z_2) = \dot{\mu}(z_1) \exp(R_s \varepsilon |z_2|).$$

$\square$

# H. Missing Discussions from Section 5.2 – Bilinear Dueling Bandits Part I (Setting)

## H.1. Motivation

Transitivity — the property that if $i \succ j$ and $j \succ k$, then $i \succ k$ — is one of the key assumptions that distinguish the dueling bandit setting (Yue & Joachims, 2009; Yue et al., 2012; Sui et al., 2018; Bengs et al., 2021). Within this stochastic transitivity framework, the most commonly considered model is the Bradley-Terry-Luce (BTL) model (Bradley & Terry, 1952): each arm $k$ has an unknown utility(reward) $r_k \in \mathbb{R}$ such that for each $(i,j) \in [K] \times [K]$, $p_{i,j} := \mathbb{P}(i \succ j) = \mu(r_i - r_j)$ with $\mu(z) := (1 + e^{-z})^{-1}$. When $K$ is large, without any additional structural assumption, the statistical guarantees (e.g., regret in dueling bandits) often increase polynomially in $K$. One very natural way of bypassing this issue is to impose a linear structure on the utility, resulting in the so-called linear BTL model: each arm $k$ is endowed with a known feature vector $\phi_k \in \mathbb{R}^d$ and $r_k = \langle \phi_k, \theta_\star \rangle$ for some unknown $\theta_\star \in \mathbb{R}^d$. This model has been successfully applied in various domains, with reinforcement learning with human feedback (Rafailov et al., 2023) being one of the most prominent applications. Coming back to dueling bandits, with such linear structure, the regret of dueling bandits has been improved from $\text{poly}(K)$ to $d$ or $\sqrt{d \log K}$ by exploiting the linear BTL model (Saha, 2021; Bengs et al., 2022).

However, the literature has two main gaps, both of which we intend to fill with our newly proposed setting and new analyses.

**Linear-like Structure in Dueling Bandits with General Preferences.** The (linear) BTL model cannot model nontransitive preferences, which hinders its applicability in various scenarios, from simple nontransitive games such as rock-paper-scissors, Blotto-style games (Balduzzi et al., 2018; 2019; Bertrand et al., 2023), and even human preferences (May, 1954; Tversky, 1969; Munos et al., 2024; Azar et al., 2024; Swamy et al., 2024; Zhang et al., 2024b).

In most of the prior literature on dueling bandits and general preference learning (i.e., not assuming linear BTL model), the learner must either learn or adapt to the entire unstructured preference matrix $P \in [0,1]^{K \times K}$. This means that, again, the statistical guarantees are expected to depend polynomially in $K$. Given that the linear structure has enabled the development of efficient algorithms for linear and dueling bandits with large action spaces and contextual information, the question of how to impose linear-like structure to arbitrary preference matrix $P$ has been a significant and longstanding open question.

There have been two notable advancements in this direction, one theoretical and one practical. The first advancement is by Wu et al. (2024), whose setting we briefly describe here. The learner has access to a feature map $(i,j) \in [K] \times [K] \mapsto \phi_{i,j} \in \mathbb{R}^d$ satisfying $\phi_{i,j} = -\phi_{j,i}$. The preference probability is defined as $p_{i,j} = \mu(\langle \phi_{i,j}, \theta_\star \rangle)$, where $\theta_\star \in \mathbb{R}^d$ is unknown. With this model, the authors have improved the Borda regret's dependency on $K$ from polynomial to logarithmic. However, it is unrealistic to know all item *pair-wise* features that linearly encode the underlying preferences. Arguably, a more realistic scenario is knowing only item-wise features, namely, $\phi_k \in \mathbb{R}^d$ for $k \in [K]$.

One may wonder if there is a contextual preference model that incorporates *item-wise* features while being potentially nontransitive. The second advancement, due to Zhang et al. (2024b), tackles this by proposing the contextual bilinear preference model: for each item pair $(i,j) \in [K] \times [K]$, the preference model is defined as

$$p_{i,j} = \mu\left(\phi_i^\top \Theta_\star \phi_j\right), \tag{62}$$

where $\Theta_\star$ is a $d \times d$ skew-symmetric matrix of low rank. However, their paper does not provide any statistical guarantees when this is used in dueling bandits, or even regarding the estimation error of the preference model; rather, their main focus is experimentally validating this model in modeling human preferences and its implications for the downstream RLHF task. Note that we adopt the same preference model, exept we allow for the underlying arm-set $\mathcal{A}$ to be continuous.

Although not discussed further in Zhang et al. (2024b), we believe this is a very natural way of incorporating some sort of linearity into general preferences, and that it deserves more attention from the dueling bandits community as well. Indeed, such bilinear model has been used in modeling interaction of two items, with applications to drug discovery (Luo et al., 2017), server scheduling (Kim & Vojnović, 2021), personalized recommendation (Chu & Park, 2009), link prediction (Menon & Elkan, 2011), relational learning (Nickel et al., 2011), and more. The bandit community was introduced to this model by bilinear bandits (Jang et al., 2021; Jun et al., 2019), later extended to low-rank matrix-armed bandits (Lu et al., 2021; Kang et al., 2022; Jang et al., 2024); refer to Appendix A for further related works on low-rank bandits. Roughly speaking, the learner now only needs to learn $\Theta(d^2)$ parameters of $\Theta_\star$ instead of $\Theta(K^2)$ parameters of $P$. Furthermore, using the low-rank structure of $\Theta_\star$, the learner can further improve the regret's dependency in $d$. Although not discussed in Zhang et al. (2024b), we also note that this is the rank-$d$ version of the low-rank preference model of Rajkumar & Agarwal (2016), as one can write $\mu^{-1}(P) = \Phi^\top \Theta_\star \Phi$ where $\Phi = [\phi_1 \cdots \phi_K] \in \mathbb{R}^{d \times K}$ and $\mu^{-1}$ is applied entry-wise.

**Variance-Aware Borda Regret Bound.** The Borda regret resembles the strong regret (Yue et al., 2012), and it "respects" the inherent problem of the difficulty of dueling bandits where two arms are chosen rather than a single arm (Saha et al., 2021; Wu et al., 2024). Its original motivation is from search engine, in which the regret corresponds to "the fraction of users who would prefer the best retrieval function over the selected ones." (Yue & Joachims, 2009).

All the existing guarantees for the Borda regret either assume a fixed gap (Saha et al., 2021) or incur a $1/c_\mu$ dependency (Wu et al., 2024), where $c_\mu$ can be thought of as the worst-case badness of linear approximation of the true preference signal. In other words, the current Borda regret bound seems to suggest that the lower the variance (which roughly corresponds to the derivative of the inverse link function in the context of GLMs), the higher the regret. However, the vast literature on logistic and generalized linear bandits (Abeille et al., 2021; Lee et al., 2024a;b) suggest otherwise. Abeille et al. (2021) first proved a $\widetilde{\mathcal{O}}(d\sqrt{T\kappa_\star})$ regret bound for logistic bandits as well as a matching (local minimax) lower bound, the correct dependency on the variance-dependent quantity. Thus, it should be expected that a similar variance-dependent quantity should pop up in the optimal Borda regret bounds.

### H.2. A Sufficient Condition for the Bilinear Preference to be Stochastic Transitive

A preference model is **stochastic transitive w.r.t.** $\mu$ (Bengs et al., 2022) if there exists a $f : [K] \to \mathbb{R}$ such that $(\boldsymbol{P})_{ij} = \mu(f(i) - f(j))$. Here, we prove that certain collinearity between the features $\phi_i$'s in the bilinear preference model (Eqn. (62)) implies stochastic transitivity:

**Theorem H.1.** *If there exists an orthonormal $\boldsymbol{Q} \in \mathbb{R}^{d \times d}$ such that $\{((\boldsymbol{Q}^\top \phi_k)_{2m-1}, (\boldsymbol{Q}^\top \phi_k)_{2m})\}_{k \in [K]}$ is collinear in $\mathbb{R}^2$ for each $m \in [r]$, then the bilinear preference model is stochastic transitive w.r.t. $\mu$. When $r = 1$ (i.e., $\mathrm{rank}(\boldsymbol{\Theta}_\star) = 2$), this is also a necessary condition.*

*Proof.* The proof is heavily inspired by Jiang et al. (2011), where the authors provide a decomposition of the space of preferences via combinatorial Hodge theory; this has been also utilized in later machine learning literature on ranking with potentially nontransitive components (Bertrand et al., 2023; Balduzzi et al., 2018; 2019).

From the combinatorial Hodge decomposition (Jiang et al., 2011, Theorem 2), a $f$ that satisfies the stochastic transitivity exists if and only if for any $(i, j, k) \in [K]^3$,

$$\phi_i^\top \boldsymbol{\Theta}_\star \phi_j + \phi_j^\top \boldsymbol{\Theta}_\star \phi_k + \phi_k^\top \boldsymbol{\Theta}_\star \phi_i = 0.$$

The quantity on the LHS is known as the *combinatorial curl* (Jiang et al., 2011).

Let $\boldsymbol{\Theta}_\star = \boldsymbol{Q}\boldsymbol{\Lambda}\boldsymbol{Q}^\top$ be its canonical form (Lemma H.2), and let $\varphi_i := \boldsymbol{Q}^\top \phi_i$. Let $\{\lambda_m\}_{m \in [r]} \subset \mathbb{R}_{>0}$ be the nonzero components of $\boldsymbol{\Lambda}$. Then, the above curl-free requirement boils down to

$$\sum_{m=1}^{r} \lambda_m \underbrace{\begin{vmatrix} 1 & 1 & 1 \\ (\varphi_i)_{2m-1} & (\varphi_j)_{2m-1} & (\varphi_k)_{2m-1} \\ (\varphi_i)_{2m} & (\varphi_j)_{2m} & (\varphi_k)_{2m} \end{vmatrix}}_{\triangleq V_m} = 0. \tag{63}$$

One sufficient condition for above to hold (necessary as well if $r = 1$) is if $V_m = 0$ for all $m \in [r]$. Geometrically, $V_m$ is the signed volume of the parallelopipe in $\mathbb{R}^3$, spanned by the three column vectors. For the volume to be zero, it must be that $\{((\varphi_i)_{2m-1}, (\varphi_i)_{2m}), ((\varphi_j)_{2m-1}, (\varphi_j)_{2m}), ((\varphi_k)_{2m-1}, (\varphi_k)_{2m})\}$ is collinear in $\mathbb{R}^2$. As this must hold for any $i, j, k \in [K]^3$, it must be that $\{((\varphi_k)_{2m-1}, (\varphi_k)_{2m})\}_{k \in [K]}$ is collinear as well, for each $m \in [r]$. $\square$

**Remark 12.** *We believe that the above result is extendable to the general case via decomposing the general preference into its transitive and cyclic components (Jiang et al., 2011). But then, geometrically, it is unclear how to choose the right features such that the non-transitive and transitive components are compatible with each other, which corresponds to the "harmonic" component from the combinatorial Hodge decomposition (Jiang et al., 2011).*

### H.3. Miscellaneous Mathematical Preliminaries

Here, for completeness and to foster future directions, we provide a bit orthogonal, yet interesting (and hopefully useful) mathematical preliminaries regarding skew-symmetric matrices and anti-symmetric tensor product space.

### H.3.1. SKEW-SYMMETRIC MATRIX

A matrix $\boldsymbol{A} \in \mathbb{R}^{d \times d}$ is **skew-symmetric** (or anti-symmetric) if $\boldsymbol{A}^\top = -\boldsymbol{A}$. It is known that the rank of a skew-symmetric matrix must be even (Hoffman & Kunze, 1971, Section 10.3), and it admits the following decomposition, which is its canonical form:

**Lemma H.2** (Corollary 2.5.11 of Horn & Johnson (2012)[11]). *$\boldsymbol{A}$ is a skew-symmetric of rank $2r \leq d$ if and only if there exists a (unique) orthogonal $\boldsymbol{Q}$ (i.e., $\boldsymbol{Q}^\top \boldsymbol{Q} = \boldsymbol{Q}\boldsymbol{Q}^\top = \boldsymbol{I}_d$) and $\{\lambda_\ell\}_{\ell \in [r]} \subset \mathbb{R}_{>0}$ such that $\boldsymbol{A} = \boldsymbol{Q}\boldsymbol{\Lambda}\boldsymbol{Q}^\top$, where*

$$\boldsymbol{\Lambda} = \left( \bigoplus_{\ell \in [r]} \lambda_\ell \boldsymbol{S} \right) \oplus \boldsymbol{0}_{d-2r}, \tag{64}$$

*where $\oplus$ is the matrix direct sum and $\boldsymbol{S} := \begin{bmatrix} 0 & 1 \\ -1 & 0 \end{bmatrix}$. Moreover, $\{\pm \lambda_\ell i\}_{\ell \in [r]}$ are the eigenvalues of $\boldsymbol{A}$.*

We also remark that the above form can be quite efficiently computed (Ward & Gray, 1978; Penke et al., 2020).

Let $\mathrm{Skew}(d) := \{\boldsymbol{\Theta} \in \mathbb{R}^{d \times d} : \boldsymbol{\Theta}^\top = -\boldsymbol{\Theta}\}$. It is a well-known that $\mathrm{Skew}(d)$ is a linear subspace of $\mathbb{R}^{d \times d}$, and that the mapping $\boldsymbol{A} \mapsto \frac{1}{2}(\boldsymbol{A} - \boldsymbol{A}^\top)$ is an orthogonal projection onto $\mathrm{Skew}(d)$ (Hoffman & Kunze, 1971, Chapter 6.6). We will also consider rank-constrained $\mathrm{Skew}(d)$, defined as $\mathrm{Skew}(d; 2r) := \{\boldsymbol{\Theta} \in \mathbb{R}^{d \times d} : \boldsymbol{\Theta}^\top = -\boldsymbol{\Theta}, \mathrm{rank}(\boldsymbol{\Theta}) = 2r\}$. This is a matrix manifold whose dimension is given as follows (see Appendix H.4 for the proof):

**Proposition H.3.** $\dim(\mathrm{Skew}(d; 2r)) = 2dr - (2r^2 + r)$.

### H.3.2. 2ND-ORDER TENSOR PRODUCT SPACE

Here, we largely follow the exposition of Section 2 of Garcia et al. (2023) and Section I.5 of Bhatia (1997), to which we refer interested readers for an overview of general tensor algebra over Hilbert space.

We define the **2nd-order tensor power** of $\mathbb{R}^d$ as $(\mathbb{R}^d)^{\otimes 2} := \{\boldsymbol{x} \otimes \boldsymbol{y} : \boldsymbol{x}, \boldsymbol{y} \in \mathbb{R}^d\}$, where the inner product[12] is such that $\langle \boldsymbol{x}_1 \otimes \boldsymbol{x}_2, \boldsymbol{y}_1 \otimes \boldsymbol{y}_2 \rangle = \langle \boldsymbol{x}_1, \boldsymbol{y}_1 \rangle \langle \boldsymbol{x}_2, \boldsymbol{y}_2 \rangle$. Then, its orthonormal basis is given as $\{\boldsymbol{e}_i \otimes \boldsymbol{e}_j\}_{(i,j) \in [d]^2}$.

Consider the symmetrization and antisymmetrization operators, defined as $\mathcal{P}_S(\boldsymbol{x} \otimes \boldsymbol{y}) := \boldsymbol{x} \odot \boldsymbol{y} := \frac{1}{2}(\boldsymbol{x} \otimes \boldsymbol{y} + \boldsymbol{y} \otimes \boldsymbol{x})$ and $\mathcal{P}_A(\boldsymbol{x} \otimes \boldsymbol{y}) := \boldsymbol{x} \wedge \boldsymbol{y} := \frac{1}{2}(\boldsymbol{x} \otimes \boldsymbol{y} - \boldsymbol{y} \otimes \boldsymbol{x})$. Then, one can orthogonally decompose $(\mathbb{R}^d)^{\otimes 2} = (\mathbb{R}^d)^{\odot 2} \oplus (\mathbb{R}^d)^{\wedge 2}$, where the two spaces are spanned by their respective *orthonormal* basis: $(\mathbb{R}^d)^{\odot 2} = \mathrm{span}\left(\{\boldsymbol{e}_i \odot \boldsymbol{e}_i\}_{i \in [d]} \cup \{\sqrt{2}(\boldsymbol{e}_i \odot \boldsymbol{e}_j)\}_{1 \leq i < j \leq d}\right)$, and $(\mathbb{R}^d)^{\wedge 2} = \mathrm{span}\left(\{\sqrt{2}(\boldsymbol{e}_i \wedge \boldsymbol{e}_j)\}_{1 \leq i < j \leq d}\right)$.

Let us focus on the antisymmetric part. It is known that $\mathcal{P}_A$ is an orthogonal projection onto $\mathbb{R}^{\wedge 2}$ with the following idempotent, full row-rank matrix representation of $\mathcal{P}_A$:

$$\boldsymbol{P}_A := \sqrt{2}\begin{bmatrix} \boldsymbol{e}_1 \wedge \boldsymbol{e}_2 & \boldsymbol{e}_1 \wedge \boldsymbol{e}_3 & \cdots & \boldsymbol{e}_{d-1} \wedge \boldsymbol{e}_d \end{bmatrix} \in \mathbb{R}^{d^2 \times \binom{d}{2}}. \tag{65}$$

It satisfies $\boldsymbol{P}_A^\top \boldsymbol{P}_A = \boldsymbol{I}_{\binom{d}{2}}$ and $\boldsymbol{P}_A \boldsymbol{P}_A^\top(\boldsymbol{x} \otimes \boldsymbol{y}) = \boldsymbol{x} \wedge \boldsymbol{y}$.

### H.4. Proof of Proposition H.3

The proof utilizes some tools from topology, Lie group theory and matrix theory. Our main references are Munkres (2018), Chapter 21 of Lee (2012) and Horn & Johnson (2012).

Consider the generalized linear group $\mathrm{GL}_d(\mathbb{R}) := \{\boldsymbol{X} \in \mathbb{R}^{d \times d} : \det(\boldsymbol{X}) \neq 0\}$, which is a Lie group of dimension $d^2$. We then define the group action of $\mathrm{GL}_d(\mathbb{R})$ on $\mathrm{Skew}(d; 2r)$ as the following:

$$(\boldsymbol{X}, \boldsymbol{A}) \mapsto \boldsymbol{X}\boldsymbol{A}\boldsymbol{X}^\top, \quad \boldsymbol{X} \in \mathrm{GL}_d(\mathbb{R}), \boldsymbol{A} \in \mathrm{Skew}(d; 2r). \tag{66}$$

We now utilize the following lemma:

---

[11]A fun(?) historical note: this decomposition has been repeatedly rediscovered and renamed: Murnaghan-Wintner decomposition (Murnaghan & Wintner, 1931), Youla decomposition (Youla, 1961), or the Schur decomposition (Balduzzi et al., 2018), although the latter is a bit inaccurate as Schur decomposition should result in an upper triangular matrix in the middle.

[12]Such inner product is unique (Bhatia, 1997, Proposition 3.8.2).

**Lemma H.4** (Theorem 21.20 of Lee (2012)). *Let $X$ be a set and $G$ be a Lie group that acts on $X$ transitively, i.e., for any $x, y \in X$ there exists a $g \in G$ such that $(g, x) = y$. Suppose that there exists a point $p \in X$ such that the stabilizer group $G_p$ is closed in $G$. Then, $X$ has a unique smooth manifold structure w.r.t. which the given action is smooth. With this structure, $\dim X = \dim G - \dim G_p$.*

We first show that our group action indeed satisfies the assumptions of the above lemma. For simplicity, let us denote

$$\boldsymbol{S}_{d,2r} := \underbrace{\bigoplus_{\ell \in [r]} \begin{bmatrix} 0 & 1 \\ -1 & 0 \end{bmatrix}}_{=: \boldsymbol{S}_{2r}} \oplus \boldsymbol{0}_{d-2r}. \tag{67}$$

**Claim H.1.** *The action as defined in Eqn. (66) is transitive.*

*Proof.* To see this, consider two $\boldsymbol{A}, \boldsymbol{B} \in \text{Skew}(d; 2r)$. Then by Lemma H.2, there exists $\boldsymbol{U_A}, \boldsymbol{U_B} \in O(d)$ and $\{\lambda_{\ell, \boldsymbol{A}}^2, \lambda_{\ell, \boldsymbol{B}}^2\}_{\ell \in [r]}$ such that $\boldsymbol{A} = \boldsymbol{U_A} \boldsymbol{\Lambda_A} \boldsymbol{S}_{d,2r} \boldsymbol{\Lambda_A}^\top \boldsymbol{U_A}^\top$ and $\boldsymbol{B} = \boldsymbol{U_B} \boldsymbol{\Lambda_B} \boldsymbol{S}_{d,2r} \boldsymbol{\Lambda_B}^\top \boldsymbol{U_B}^\top$, where

$$\boldsymbol{\Lambda_A} = \text{diag}(\underbrace{\lambda_{1,\boldsymbol{A}}, \lambda_{1,\boldsymbol{A}}}_{\text{twice}}, \cdots, \underbrace{\lambda_{r,\boldsymbol{A}}, \lambda_{r,\boldsymbol{A}}}_{\text{twice}}, \underbrace{0, 0, \ldots, 0}_{\text{remaining entries}})$$

and similarly for $\boldsymbol{\Lambda_B}$. Then, defining $\boldsymbol{X} = (\boldsymbol{U_B} \boldsymbol{\Lambda_B})(\boldsymbol{U_A} \boldsymbol{\Lambda_A})^{-1} \in \text{GL}_d(\mathbb{R})$, it can be seen that $(\boldsymbol{X}, \boldsymbol{A}) = \boldsymbol{B}$. □

For the point $p$ in the above lemma, we choose $\boldsymbol{S}_{d,2r} \in \text{Skew}(d; 2r)$. Let us denote its stabilizer group as $S_{d,2r} := \{\boldsymbol{X} \in \text{GL}_{d-2r}(\mathbb{R}) : \boldsymbol{X} \boldsymbol{S}_{d,2r} \boldsymbol{X}^\top = \boldsymbol{S}_{d,2r}\}$.

**Claim H.2.** $S_{d,2r}$ *is closed in* $\text{GL}_d(\mathbb{R})$.

*Proof.* Consider a mapping $\rho : \boldsymbol{X} \mapsto \boldsymbol{X} \boldsymbol{S}_{d,2r} \boldsymbol{X}^\top$, which is continuous. Noting that $S_{d,2r} = \rho^{-1}(\{\boldsymbol{S}_{d,2r}\})$ and that $\{\boldsymbol{S}_{d,2r}\}$ is closed (in Hausdorff space, which $\text{GL}_d(\mathbb{R})$ is), $S_{d,2r}$ is also closed by continuity. □

We now characterize $S_{d,2r}$.

Using block matrix notation, we need to characterize $\boldsymbol{X} = \begin{bmatrix} \boldsymbol{X}_{11} & \boldsymbol{X}_{12} \\ \boldsymbol{X}_{21} & \boldsymbol{X}_{22} \end{bmatrix}$ such that $\boldsymbol{X}$ is invertible and $\boldsymbol{X} \boldsymbol{S}_{2r} \boldsymbol{X}^\top = \boldsymbol{S}_{2r}$.
After some tedious computations, we have that

$$\begin{bmatrix} \boldsymbol{X}_{11} \boldsymbol{S}_{2r} \boldsymbol{X}_{11}^\top & \boldsymbol{X}_{11} \boldsymbol{S}_{2r} \boldsymbol{X}_{21}^\top \\ \boldsymbol{X}_{21} \boldsymbol{S}_{2r} \boldsymbol{X}_{11}^\top & \boldsymbol{X}_{21} \boldsymbol{S}_{2r} \boldsymbol{X}_{21}^\top \end{bmatrix} = \begin{bmatrix} \boldsymbol{S}_{2r} & \boldsymbol{0}_{2r \times (d-2r)} \\ \boldsymbol{0}_{(d-2r) \times 2r} & \boldsymbol{0}_{2r \times 2r} \end{bmatrix}.$$

Consider the first block. Taking the determinant, we can deduce that $\det(\boldsymbol{X}_{11})^2 = 1 \neq 0$, i.e., $\boldsymbol{X}_{11}$ should be invertible. As $\boldsymbol{S}_{2r}$ is also invertible, the antidiagonal blocks implies that $\boldsymbol{X}_{21} = \boldsymbol{0}_{(d-2r) \times 2r}$.

So far, we have that $\boldsymbol{X}$ should be of the form

$$\boldsymbol{X} = \begin{bmatrix} \boldsymbol{X}_{11} & \boldsymbol{X}_{12} \\ \boldsymbol{0}_{(d-2r) \times 2r} & \boldsymbol{X}_{22,} \end{bmatrix}$$

where $\boldsymbol{X}_{11} \in \text{Sym}(2p) := \{\boldsymbol{X} \in \text{GL}_n(\mathbb{R}) : \boldsymbol{X} \boldsymbol{S}_{2r} \boldsymbol{X}^\top = \boldsymbol{X}\}$. By Schur's determinant formula, as $\boldsymbol{X}$ must be invertible, we must have that

$$\det(\boldsymbol{X}) = \det(\boldsymbol{X}_{11}) \det(\boldsymbol{X}_{22}) \neq 0,$$

i.e., $\boldsymbol{X}_{22}$ should also be invertible.

We now derive the dimension of $\text{GL}_{d-2r}(\mathbb{R}) \text{Sym}(2r)$.

**Claim H.3.** $\dim(\text{GL}_{d-2r}(\mathbb{R})) = (d-2r)^2$.

*Proof.* Let $n = d - 2r$. Then, note that $\text{GL}_n(\mathbb{R}) = \det^{-1}(\mathbb{R} \setminus \{0\})$. As $\det$ is continuous and $\mathbb{R} \setminus \{0\}$ is open, $\text{GL}_n(\mathbb{R}) \subset \mathbb{R}^{n \times n}$ is open, and we are done. □

**Claim H.4.** $\dim(\mathrm{Sym}(2r)) = 2r^2 + r$.

*Proof.* We do this by counting the number of independent constraints, then subtracting it from $\dim(\mathrm{GL}_{2r}(\mathbb{R})) = 4r^2$. Let us denote $\boldsymbol{S} := \begin{bmatrix} 0 & 1 \\ -1 & 0 \end{bmatrix}$ for simplicity. First, for a $\boldsymbol{A} \in \mathbb{R}^{2 \times 2}$, note that

$$\boldsymbol{A}\boldsymbol{S}\boldsymbol{A}^\top = \det(\boldsymbol{A})\boldsymbol{S}.$$

Now consider a $\boldsymbol{X} \in \mathrm{GL}_{2r}(\mathbb{R})$, consisting of $r$ number of $2 \times 2$ blocks:

$$\boldsymbol{X} = \begin{bmatrix} \boldsymbol{X}_{11} & \boldsymbol{X}_{12} & \cdots & \boldsymbol{X}_{1r} \\ \boldsymbol{X}_{21} & \boldsymbol{X}_{22} & \cdots & \boldsymbol{X}_{2r} \\ \vdots & \vdots & \ddots & \vdots \\ \boldsymbol{X}_{r1} & \boldsymbol{X}_{r2} & \cdots & \boldsymbol{X}_{rr} \end{bmatrix}.$$

Then, by the block matrix multiplication and the above result, we have that

$$\left(\boldsymbol{X}\boldsymbol{S}_{2r}\boldsymbol{X}^\top\right)_{i,j} = \begin{cases} \left(\sum_{k=1}^r \det(\boldsymbol{X}_{ik})\right)\boldsymbol{S}, & i = j, \\ \sum_{k=1}^r \boldsymbol{X}_{ik}\boldsymbol{J}\boldsymbol{X}_{kj}^\top, & i \neq j \end{cases} = \begin{cases} \boldsymbol{S}, & i = j, \\ \boldsymbol{0}_{2 \times 2}, & i \neq j \end{cases}.$$

where here, $(\cdot)_{i,j}$ refers to the $2 \times 2$ block at the $(i, j)$ location.

There are $r$ constraints for $i = j$ and $4\binom{r}{2} = 2r(r-1)$ constraints for $i \neq j$, which amounts to $2r^2 - r$ constraints in total. Thus, the dimension of $\mathrm{Sym}(2r)$ becomes $4r^2 - (2r^2 - r) = 2r^2 + r$. $\qquad\square$

All in all, we have that

$$\dim(S_{d,2r}) = \underbrace{\dim(\mathrm{Sym}(2r))}_{\text{degrees of freedom for } \boldsymbol{X}_{11}} + \underbrace{\dim(\mathbb{R}^{2r \times (d-2r)})}_{\text{degrees of freedom for } \boldsymbol{X}_{12}} + \underbrace{\dim(\mathrm{GL}_{d-2r}(\mathbb{R}))}_{\text{degrees of freedom for } \boldsymbol{X}_{22}}$$

$$= (2r^2 + r) + 2r(d - 2r) + (d - 2r)^2$$

$$= d^2 + 2r^2 + r - 2dr.$$

Applying Lemma H.4, we have that

$$\dim(\mathrm{Skew}(d; 2r)) = \dim(\mathrm{GL}_d(\mathbb{R})) - \dim(S_{d,2r}) = 2dr - (2r^2 + r).$$

$\qquad\square$

# I. Missing Discussions from Section 5.2 – Bilinear Dueling Bandits Part II (Regret Analysis)

## I.1. Proof of Theorem 5.1 – Borda Regret Upper Bound for Bilinear Dueling Bandits

We state the full version of the Borda regret bound and give its proof:

**Theorem I.1** (Full Statement of Theorem 5.1). *Let us denote* $\mathrm{GL}_{\min} := \mathrm{GL}_{\min}(\mathcal{A})$. *Choose* $N_1$ *and* $N_2$ *as*

$$N_1 \asymp \frac{r^2 R_{\max}^2}{\kappa_\star^2 C_{\min}^2} \max \left\{ d^4 + \log \frac{1}{\delta} + \frac{R_s^2 r^2 R_{\max} \log \frac{d}{\delta}}{\kappa_\star^2 C_{\min}^2}, \ R_s d \left( \log \frac{d}{\delta} \right)^{2/3} \left( \frac{\mathrm{GL}_{\min}}{\kappa_\star^3} \right)^{1/6} (\kappa_\star^B T)^{1/3} \right\}, \tag{68}$$

$$N_2 = \left( \mathrm{GL}_{\min} \log \frac{d}{\delta} \right)^{1/3} (\kappa_\star^B T)^{2/3}, \tag{69}$$

*and let us assume that* $T \geq N_1 + N_2$. *Then, the following Borda regret bound of* `BETC-GLM-LR`[13] *holds with probability at least* $1 - \delta$:

$$\mathrm{Reg}^B(T) \lesssim \left( \mathrm{GL}_{\min} \log \frac{d}{\delta} \right)^{1/3} (\kappa_\star^B T)^{2/3} + R_s R_{\max} \left( \frac{\mathrm{GL}_{\min}}{\kappa_\star^B} \log \frac{d}{\delta} \right)^{2/3} T^{1/3} + N_1. \tag{70}$$

*Here, it is clear that the first term dominates when* $T$ *is sufficiently large.*

*Proof.* We naïvely bound the instantaneous regret from the exploration phase with 1, and thus, the cumulative regret up to the forced exploration is $N_1 + N_2$.

After the exploration phase, the instantaneous regret is the same as $B(\phi_\star) - B(\widehat{\phi})$. This is bounded as follows:

$$\begin{aligned} B(\phi_\star) - B(\widehat{\phi}) &= \mathbb{E}_{\phi' \sim \mathrm{Unif}(\mathcal{A})} \left[ \mu \left( \phi_\star^\top \Theta_\star \phi' \right) - \mu(\widehat{\phi}^\top \Theta_\star \phi') \right] \\ &\leq \mathbb{E}_{\phi' \sim \mathrm{Unif}(\mathcal{A})} \left[ \mu \left( \phi_\star^\top \Theta_\star \phi' \right) - \mu(\phi_\star^\top \widehat{\Theta} \phi') \right] \qquad \text{(Definition of } \widehat{\phi}) \\ &\overset{(*)}{=} \underbrace{\mathbb{E}_{\phi' \sim \mathrm{Unif}(\mathcal{A})} \left[ \dot{\mu} \left( \phi_\star^\top \Theta_\star \phi' \right) \phi_\star^\top (\Theta_\star - \widehat{\Theta}) \phi' \right]}_{\triangleq Q_1} + \underbrace{\mathbb{E}_{\phi' \sim \mathrm{Unif}(\mathcal{A})} \left[ - \left( \phi_\star^\top (\Theta_\star - \widehat{\Theta}) \phi' \right)^2 \tilde{\theta}(\phi') \right]}_{\triangleq Q_2} \end{aligned}$$

(First-order Taylor expansion with integral remainder)

where at $(*)$, we define

$$\tilde{\theta}(\phi') := \int_0^1 (1 - z) \ddot{\mu} \left( \phi_\star^\top \left( (1 - z)\Theta_\star + z\widehat{\Theta} \right) \phi' \right) dz.$$

$Q_1$ can be bounded as

$$\begin{aligned} Q_1 &= \mathbb{E}_{\phi' \sim \mathrm{Unif}(\mathcal{A})} \left[ \dot{\mu} \left( \phi_\star^\top \Theta_\star \phi' \right) \phi_\star^\top (\Theta_\star - \widehat{\Theta}) \phi' \right] \\ &\leq \left( \max_{\phi' \in \mathcal{A}} \left| \phi_\star^\top (\Theta_\star - \widehat{\Theta}) \phi' \right| \right) \mathbb{E}_{\phi' \sim \mathrm{Unif}(\mathcal{A})} \left[ \dot{\mu} \left( \phi_\star^\top \Theta_\star \phi' \right) \right] \\ &\leq \kappa_\star^B \left\| \widehat{\Theta} - \Theta_\star \right\|_{\mathrm{op}} \qquad \text{(rectangular quotient relation for } \|\cdot\|_{\mathrm{op}} \text{ \& } \phi_\star, \phi' \in \mathcal{B}^d(1) \text{ \& definition of } \kappa_\star^B) \\ &\lesssim \kappa_\star^B \sqrt{\frac{\mathrm{GL}_{\min}}{N_2} \log \frac{d}{\delta}}. \qquad \text{(Theorem 3.1)} \end{aligned}$$

By self-concordance,

$$|\tilde{\theta}(\phi')| \leq \int_0^1 (1 - z) \left| \ddot{\mu} \left( \phi_\star^\top \left( (1 - z)\Theta_\star + z\widehat{\Theta} \right) \phi' \right) \right| dz$$

---

[13]This is an acronym for Borda Explore-Then-Commit for Generalized Linear Models with Low-Rank structure.

$$\leq R_s \int_0^1 (1-z)\dot{\mu}\left(\phi_\star^\top \left((1-z)\boldsymbol{\Theta}_\star + z\widehat{\boldsymbol{\Theta}}\right)\phi'\right)dz \qquad \text{(self-concordance)}$$

$$\leq R_s R_{\max} \int_0^1 (1-z)dz$$

$$= \frac{1}{2} R_s R_{\max},$$

and thus $Q_2$ can be bounded as

$$Q_2 = \mathbb{E}_{\phi'\sim\text{Unif}(\mathcal{A})}\left[-\left(\phi_\star^\top(\boldsymbol{\Theta}_\star - \widehat{\boldsymbol{\Theta}})\phi'\right)^2 \tilde{\theta}(\phi')\right] \leq \frac{1}{2}R_s R_{\max}\mathbb{E}_{\phi'\sim\text{Unif}(\mathcal{A})}\left[\left(\phi_\star^\top(\boldsymbol{\Theta}_\star - \widehat{\boldsymbol{\Theta}})\phi'\right)^2\right] \lesssim \frac{R_s R_{\max}\text{GL}_{\min}}{N_2}\log\frac{d}{\delta}.$$

Combining everything, we have that

$$B(\phi_\star) - B(\widehat{\phi}) \lesssim \kappa_\star^B\sqrt{\frac{\text{GL}_{\min}}{N_2}\log\frac{d}{\delta}} + \frac{R_s R_{\max}\text{GL}_{\min}}{N_2}\log\frac{d}{\delta}.$$

All in all, we have

$$\text{Reg}^B(T) \lesssim N_1 + N_2 + (T - N_1 - N_2)\left(\kappa_\star^B\sqrt{\frac{\text{GL}_{\min}}{N_2}\log\frac{d}{\delta}} + \frac{R_s R_{\max}\text{GL}_{\min}}{N_2}\log\frac{d}{\delta}\cdot\right)$$

$$\leq N_1 + N_2 + T\sqrt{\frac{\text{GL}_{\min}}{N_2}\log\frac{d}{\delta}}\left(\kappa_\star^B + R_s R_{\max}\sqrt{\frac{\text{GL}_{\min}}{N_2}\log\frac{d}{\delta}}\right). \tag{71}$$

Let us optimize for $N_2$ using the last expression.

If we choose $N_2 = \left(\text{GL}_{\min}\log\frac{d}{\delta}\right)^{1/3}(\kappa_\star^B T)^{2/3}$, we have

$$\text{Reg}^B(T) \lesssim N_1 + \left(\text{GL}_{\min}\log\frac{d}{\delta}\right)^{1/3}(\kappa_\star^B T)^{2/3} + R_s R_{\max}\left(\frac{\text{GL}_{\min}}{\kappa_\star^B}\log\frac{d}{\delta}\right)^{2/3}T^{1/3}. \tag{72}$$

With this choice of $N_2$, one can simplify the requirement on $N_1$ (as stated in Theorem 3.1) as follows: denoting $C_{\min} := \max_{\pi_1\in\mathcal{P}(\mathcal{A})}\lambda_{\min}(\boldsymbol{V}(\pi_1))$,

$$N_1 \asymp \frac{r^2 R_{\max}^2}{\kappa_\star^2 C_{\min}^2}\max\left\{d^4 + \log\frac{1}{\delta} + \frac{R_s^2 r^2 R_{\max}\log\frac{d}{\delta}}{\kappa_\star^2 C_{\min}^2}, \ R_s d\sqrt{\frac{N_2\log\frac{d}{\delta}}{\kappa_\star}}\right\}$$

$$= \frac{r^2 R_{\max}^2}{\kappa_\star^2 C_{\min}^2}\max\left\{d^4 + \log\frac{1}{\delta} + \frac{R_s^2 r^2 R_{\max}\log\frac{d}{\delta}}{\kappa_\star^2 C_{\min}^2}, \ R_s d\left(\log\frac{d}{\delta}\right)^{2/3}\left(\frac{\text{GL}_{\min}}{\kappa_\star^3}\right)^{1/6}(\kappa_\star^B T)^{1/3}\right\}. \qquad \text{(Plug in } N_2\text{)}$$

The proof then concludes by rearranging and going through some tedious computations. $\qquad\square$

### I.2. Relations to Wu et al. (2024)

**Reduction to Wu et al. (2024).** To our knowledge, Wu et al. (2024) is the only comparable competitor in our setting of Borda regret minimization. To do that, we first describe how to reduce our bilinear dueling bandits to their setting. Recall that Wu et al. (2024) require vector-valued features for each pair of items, $\phi_{i,j} = -\phi_{j,i}$. As $\boldsymbol{\Theta}_\star = \widetilde{\boldsymbol{\Theta}}_\star - \widetilde{\boldsymbol{\Theta}}_\star^\top$ for some $\widetilde{\boldsymbol{\Theta}}_\star \in \mathbb{R}^{d\times d}$, one can rewrite the bilinear preference as

$$\mu\left(\phi_i^\top(\widetilde{\boldsymbol{\Theta}}_\star - \widetilde{\boldsymbol{\Theta}}_\star^\top)\phi_j\right) = \mu\left(\left\langle\widetilde{\boldsymbol{\Theta}}_\star, \phi_i\phi_j^\top - \phi_j\phi_i^\top\right\rangle\right).$$

One may be tempted to set $\phi_{i,j} = \text{vec}(\phi_i\phi_j^\top - \phi_j\phi_i^\top)$. However, recalling the discussions from Appendix H.3.2, one must set $\phi_{i,j} = \boldsymbol{P}_A^\top\text{vec}(\phi_i\phi_j^\top - \phi_j\phi_i^\top)$ for $\phi_{i,j}$'s to be able to fully span $\mathbb{R}^{\wedge 2}$. Setting $\theta_\star = \boldsymbol{P}_A^\top\text{vec}(\widetilde{\boldsymbol{\Theta}}_\star)$ and the reduction is complete.

**Regret Upper Bound.** A naïve application of the algorithm of Wu et al. (2024) using the above reduction attains a Borda regret bound of $\widetilde{\mathcal{O}}(c_\mu^{-1} d^{4/3} T^{2/3})$ up to some epsilon-net error (see their Remark 5.3), where

$$c_\mu := \min_{\|\boldsymbol{x}\|_2 \leq 1, \|\boldsymbol{\theta} - \boldsymbol{\theta}_\star\| \leq 1} \dot{\mu}(\langle \boldsymbol{x}, \boldsymbol{\theta} \rangle) > 0. \tag{73}$$

They have also assumed that $\lambda_{\min}(\boldsymbol{V}(\pi^U)) \geq \lambda_0$ for some constant $\lambda_0 > 0$, where $\pi^U \sim \mathrm{Unif}(\mathcal{A} \times \mathcal{A})$ (Wu et al., 2024, Assumption 3.1). We remark that in many cases, $\lambda_0$ is *not* constant and can be arbitrarily small dimension-wise. In particular, both Wu et al. (2024) and our work assumes $\|\phi_{i,j}\|_2 \leq 1$, one can prove that $\lambda_0 \leq \frac{1}{d^2}$ **for any** $\mathcal{A}$ under this assumption and it is *impossible* to make $\lambda_0$ as a constant, since

$$
\begin{aligned}
\mathrm{tr}\left(\boldsymbol{V}(\pi)\right) &= \mathrm{tr}\left(\sum_{i,j} \pi(\phi_{i,j}) \phi_{i,j} \phi_{i,j}^\top\right) \\
&= \sum_{i,j} \pi(\phi_{i,j}) \mathrm{tr}\left(\phi_{i,j} \phi_{i,j}^\top\right) && \text{(Linearity of } \mathrm{tr}) \\
&\leq \sum_{i,j} \left(\pi(\phi_{i,j})\right) = 1 && \text{(For a vector } v, \mathrm{tr}(vv^\top) = \|v\|_2^2 \text{ and } \phi_{i,j} \leq 1)
\end{aligned}
$$

and $\mathrm{tr}(\boldsymbol{V}(\pi)) = \sum_{i=1}^{d^2} \lambda_i(\boldsymbol{V}(\pi))$.

Still, for a fair comparison, let us first compare with our bound under the same assumption. By Theorem 5.1 and Proposition 3.2, our BETC-GLM-LR achieves a Borda regret bound of $\widetilde{\mathcal{O}}\left(\left(\frac{(\kappa_\star^B)^2}{\lambda_0 \kappa_\star}\right)^{1/3} d^{1/3} T^{2/3}\right)$. While the regret depends on the geometry of $\mathcal{A}$, making a direct comparison challenging in cases where $\mathcal{A}$ is ill-distributed, our algorithm demonstrates a superior regret bound in terms of $d$ in many practical scenarios. Notably, when $\lambda_0 \geq \frac{1}{d^3}$, which holds in a wide range of common settings, our method outperforms Wu et al. (2024). For example, in the case of the entrywise dueling bandit, $\mathcal{A} = \{e_i : i \in [d]\}$, owing to Corollary 3.3, our regret bound becomes $\widetilde{\mathcal{O}}\left(\left(\frac{(\kappa_\star^B)^2}{\kappa_\star}\right)^{1/3} dT^{2/3}\right)$, which is strictly better than the $d^{4/3}$-dependency of Wu et al. (2024).

Curvature-wise, it is easy to see that $c_\mu \leq \kappa_\star^B$, and the gap may be large (Faury et al., 2020, Section 2). Indeed, our Borda regret bound analysis provides an interesting quantity that determines the problem difficulty, $\frac{(\kappa_\star^B)^2}{\kappa_\star}$, which has not been reported before. Let us first recall their definitions:

$$\kappa_\star := \min_{\boldsymbol{\phi}, \boldsymbol{\phi}' \in \mathcal{A}} \dot{\mu}\left(\boldsymbol{\phi}^\top \boldsymbol{\Theta}_\star \boldsymbol{\phi}'\right), \quad \kappa_\star^B := \mathbb{E}_{\boldsymbol{\phi}' \sim \mathrm{Unif}(\mathcal{A})}[\dot{\mu}(\boldsymbol{\phi}^\top \boldsymbol{\Theta} \boldsymbol{\phi}')]. \tag{74}$$

$\kappa_\star$ is the worst-case flatness across all pairs of arms while $\kappa_\star^B$ is the worst-case flatness for the *Borda winner* vs. other arms. This then gives the interpretation that if the hardness of identifying the true winner for all possible pairs is of same order (i.e., $\kappa_\star^B \asymp \kappa_\star$), then our regret bound scales as $\widetilde{\mathcal{O}}(\kappa_\star^{1/3}(dT)^{2/3})$, i.e., flatter problem indicates lower permanent regret. Here, permanent means the regime after identifying $\boldsymbol{\Theta}_\star$ (Abeille et al., 2021). On the other hand, if there exists an item pair such that identifying the true winner is much harder than that when one of the items is the Borda winner (e.g., $(\kappa_\star^B)^2 \asymp \kappa_\star$), then our permanent regret does not benefit from the flatness. This is because our GL-LowPopArt exploits the low-rankness of $\mathcal{A}$ (which is of rank 1) and the parameter space $\mathrm{Skew}(d; 2r)$, analogous to bilinear bandits (Jun et al., 2019; Jang et al., 2021) and low-rank bandits (Jang et al., 2024; Lu et al., 2021; Kang et al., 2022).

**Remark 13.** *Surprisingly, our regret bound is independent of the rank $r$ of the matrix $\boldsymbol{\Theta}_\star$, which is also the case for bilinear bandits (Jang et al., 2021, Theorem 4.6) albeit for a different reason. We believe that this showcases how GL-LowPopArt is adaptive to the arm-set geometry of $\mathcal{A} \subseteq \mathcal{B}_{\mathrm{op}}^{d \times d}(1)$, quantified by $\mathrm{GL}_{\min}(\mathcal{A}) \leq \frac{d}{\kappa_\star \lambda_0}$.*

**Regret Lower Bound.** Wu et al. (2024, Theorem 4.1) obtain a regret lower bound of $\Omega(d^{2/3} T^{2/3})$ for $\phi_{i,j}, \boldsymbol{\theta}_\star \in \mathbb{R}^d$, and a similar lower bound for unstructured dueling bandits has been obtained by Saha et al. (2021, Theorem 16); $T^{2/3}$ stems from the fact that the exploration and exploitation cannot be mixed. This suggests that at least in terms of $T$, our BETC-GLM-LR is also optimal.

However, their lower bound cannot be directly applied to our setting, as our bilinear dueling bandits, in essence, constrain the matrix arm to be of rank-1. It is clear that their hard instance, based on the lower bound for stochastic linear bandits (Dani et al., 2008), cannot be instantiated as our setting. We leave obtaining a tight lower bound to future work, considering how even in stochastic bilinear bandits (non-dueling), the lower bound remains open (Kotłowski & Neu, 2019; Jang et al., 2021; Jun et al., 2019). A potential starting point may be from the regret lower bound of Jang et al. (2024, Theorem 6.1), although they do not consider the Borda regret nor nonlinear link function.

# J. Preliminary Experiments: 1-Bit Matrix Completion/Recovery

In this Appendix, we present numerical results on 1-bit matrix completion/recovery (Davenport et al., 2014) to demonstrate the effectiveness of `GL-LowPopArt`; for results in the Gaussian (i.e., linear) setting, we refer readers to the experiments in Jang et al. (2024). The code is publicly available on our GitHub repository.[14]

## J.1. Experimental Setting

**Dataset.**  We largely follow the setup in Jang et al. (2024). We set $d = d_1 = d_2 = 3$ and rank $r = 1$. To observe average performance, we repeat each experiment 60 times for each sample size $N \in \{10^4, 2 \cdot 10^4, 3 \cdot 10^4, 4 \cdot 10^4, 5 \cdot 10^4\}$. Each repetition samples a random instance as $\mathbf{\Theta}_\star = 2\mathbf{U}\mathbf{U}^\top$, where $\mathbf{U} = \mathrm{QR}(\mathbf{U}')$ with $\mathbf{U}' \sim \mathcal{N}(0,1)^{d \times r}$.

We evaluate two arm sets $\mathcal{A}$: (i) the matrix completion basis $\mathcal{X} = \{\mathbf{e}_i\mathbf{e}_j^\top : 1 \le i, j \le 3\}$ (and $\{\mathbf{e}_i\}_i$ is the standard basis of $\mathbb{R}^{d_1}$) and (ii) random measurements sampled uniformly from $\partial\mathcal{B}_F^{d_1 \times d_2}(1)$. For matrix recovery, the arm set is sampled once at the beginning and fixed with $|\mathcal{A}| = K = 150$. In the other two settings, the arm set satisfies $|\mathcal{A}| = d_1 d_2 = 9$.

**Algorithms.**  To provide a complete picture of each of the components, we consider a total of 7 different algorithms, summarized in the table below:

|  | Acronym | Algorithm | E-opt | GL-opt |
|---|---|---|---|---|
| Nuclear norm regularized MLE | E | Stage I (E-opt) | ✓ | – |
|  | U | Stage I (Uniform) | ✗ | – |
| `GL-LowPopArt` | E + GL | Stage I (E-opt) + II (GL-opt) | ✓ | ✓ |
|  | E + U | Stage I (E-opt) + II (Uniform) | ✓ | ✗ |
|  | U + GL | Stage I (Uniform) + II (GL-opt) | ✗ | ✓ |
|  | U + U | Stage I (Uniform) + II (Uniform) | ✗ | ✗ |
| Burer-Monteiro Factorization (BMF) | BMF-GD | Gradient Descent | – | – |

*Table 3.* "E-opt" and "GL-opt" indicate whether E-optimal and GL-optimal designs are used in Stage I and II, respectively. GL-optimal design refers to $\min_{\pi_2} \mathrm{GL}(\pi_2)$; see Section 3.2. When the experimental design is not utilized, we default to uniform distribution over $\mathcal{A}$.

For both Stage I and II, we use the theoretically prescribed hyperparameters without tuning. Specifically, we set $\lambda_N = \sqrt{\frac{2}{N}\log\frac{6}{\delta}}$ for Stage I only, and $\lambda_N = \sqrt{\frac{2}{N_1}\log\frac{6}{\delta}}$ and $\sigma_{\mathrm{thres}} = \sqrt{\frac{16\mathrm{GL}(\pi_2;\mathbf{\Theta}_0)}{N_2}\log\frac{24}{\delta}}$ when both stages are used. To ensure fairness, we fix the total sample size $N$ across all methods and enforce $N_1 + N_2 = N$, where $N_i$ is the number of samples used in Stage $i$. Specifically, for this experiment, we set $N_1 = \lfloor N/2 \rfloor$ and $N_2 = N - N_1$.[15]

For the BMF approach, we utilize a small random initialization (Stöger & Soltanolkotabi, 2021; Kim & Chung, 2023) of $\mathbf{U}_0 \sim 10^{-4} \cdot \mathcal{N}(0,1)^{d_1 \times r}$, and factorize $\mathbf{\Theta} = \mathbf{U}\mathbf{U}^\top$. We optimize the (negative) log-likelihood over samples collected via the uniform policy, using gradient descent with a learning rate of $0.01$. We impose a stopping criterion of either when the gradient norm drops below $10^{-6}$ or after a maximum of $10^4$ iterations.

## J.2. Results & Discussion

We report $95\%$ studentized bootstrapped confidence intervals with bias correction (DiCiccio & Efron, 1996; Hall, 1992) for each (algorithm, $N$) pair, using 1000 outer bootstrap samples and 500 inner samples. When the empirical variation is too small for reliable studentization, we fall back to the percentile bootstrap interval.

Figure 1 summarizes the main results. First, note that BMF-GD fails for all considered settings, showing that the non-convex

---

[14] https://github.com/nick-jhlee/GL-LowPopArt

[15] In the main text, we mentioned how $N_1 \asymp \sqrt{N}$ suffices. However, that is the case in the asymptotic scenario; to account for finite size effect, we divide the samples equally to two parts. We leave further ablation studies on the effect of $N_1$-$N_2$ splits to future work.

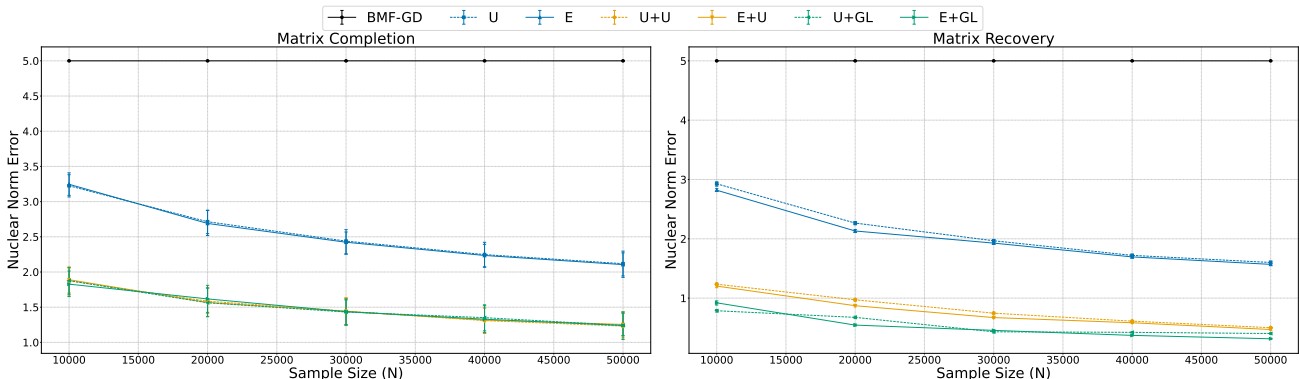

*Figure 1.* Plots of the nuclear norm errors for $N \in \{10^4, 2 \cdot 10^4, 3 \cdot 10^4, 4 \cdot 10^4, 5 \cdot 10^4\}$.

loss landscape is not-so-benign in the noisy setting, as suggested by Ma & Fattahi (2023). For matrix completion, we observe no difference in performance with or without the Stage II design. This is consistent with expectations: since $\mathcal{X}$ consists of independent, orthogonal basis matrices, the optimal design reduces to the uniform distribution $\mathrm{Unif}([K])$.

In contrast, for matrix recovery, we find that incorporating the Stage II design consistently improves performance across all tested sample sizes. This is due to the heterogeneous structure of the randomly sampled $\mathcal{A}$, for which an adaptive design more effectively prioritizes informative measurements.

### J.3. Ablation: Necessity of Stage I

A natural question is whether Stage I is truly necessary in practice. Theoretically, Stage I provides an asymptotically consistent initial estimator that linearizes the problem, which is essential for the self-concordance analysis underlying the Stage II Catoni estimator.

We empirically investigate this by comparing Stage II performance under four different initializations: U+GL, E+GL, 0-GL (a zero initialization: $\boldsymbol{\Theta}_0 = \mathbf{0}$), and Rand-GL (a random Gaussian initialization: $\boldsymbol{\Theta}_0 \sim \mathcal{N}(0, 1)^{d_1 \times d_2}$). For the latter two initializations (which we refer to as "naïve", we allocate the entire sample budget $N$ to Stage II. For (i) and (ii), we follow the same protocol as done previously, splitting the budget into $N_1 = \lfloor N/2 \rfloor$ for Stage I and $N_2 = N - N_1$ for Stage II.

As illustrated in Figure 2, the MLE-based initializations from Stage I (both with and without the E-optimal design) significantly outperform the naïve alternatives; notably, those alternatives' errors do not decay with the number of samples. This highlights the practical importance of Stage I in reducing bias and enabling effective downstream estimation in Stage II.

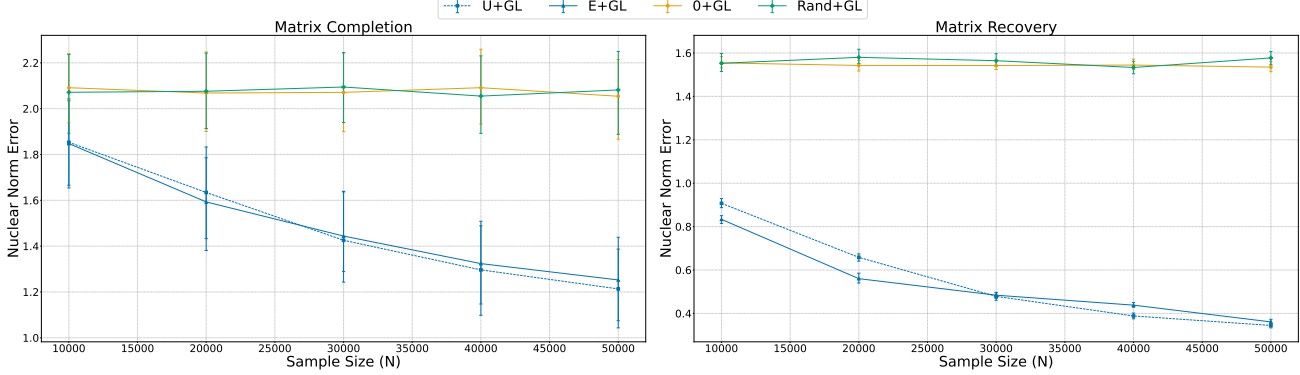

*Figure 2.* Ablation plots of the nuclear norm errors for $N \in \{10^4, 2 \cdot 10^4, 3 \cdot 10^4, 4 \cdot 10^4, 5 \cdot 10^4\}$.

