# OpenReview forum: "GL-LowPopArt: A Nearly Instance-Wise Minimax-Optimal Estimator for Generalized Low-Rank Trace Regression"
_ICML.cc/2025/Conference — ICML 2025 Retracted by Authors_

### Official Review · Reviewer_niyW · 2025-03-09

**Overall Recommendation:** 4

**Summary:**

In this work, the authors developed a new estimator for the generalized linear low-rank trace regression problem. The estimator improves existing works by considering instance-dependent information. Additionally, the estimator is nearly minimax optimal locally around the global optimizer. The authors also discussed the application of the proposed estimator to 1-bit matrix completion and bi-linear dueling bandits problems. Please see the following sections for my detailed comments.

**Claims And Evidence:**

Due to the time limit, I did not check the correctness of the theory, except those briefly mentioned in the main paper. The theoretical claims seem correct by checking the main paper.

**Essential References Not Discussed:**

It would be better if the authors could briefly discuss the new optimization complexity metric for generalized linear matrix completion under USR:

- Yalçın, Baturalp, et al. "Factorization approach for low-complexity matrix completion problems: Exponential number of spurious solutions and failure of gradient methods." International Conference on Artificial Intelligence and Statistics. PMLR, 2022.

- Zhang, Haixiang, et al. "A new complexity metric for nonconvex rank-one generalized matrix completion." Mathematical Programming 207.1 (2024): 227-268.

Also, it would be better if the authors could briefly discuss or compare to the bound for the 1-bit matrix completion bound in:

- Bi, Yingjie, Haixiang Zhang, and Javad Lavaei. "Local and global linear convergence of general low-rank matrix recovery problems." Proceedings of the AAAI Conference on Artificial Intelligence. Vol. 36. No. 9. 2022.

**Experimental Designs Or Analyses:**

N/A

**Methods And Evaluation Criteria:**

The methods and evaluation criteria make sense to me.

**Other Comments Or Suggestions:**

- In Remark 1, should the bound be $C(\pi_{nuc}^*) \geq C(\pi_{E}) / 2$?

- In Theorem 4.1, I wonder if the authors explain the intuition why a lower bound on $N$ is required?

- Could the authors elaborate a little more on the adaptive estimation procedure mentioned at the end of Section 5.1?

- I appreciate the novelty in the bounds and algorithms proposed in this work. However, I am also curious how the computational cost compares with algorithms that utilize the Burer-Monteiro factorization. When the problem scale ($d_1, d_2$) is large, the factorization-based algorithms may outperform the proposed algorithm in this work.

- It would be better if the authors could also compare the proposed algorithm with existing algorithms, potentially algorithms that depend on the Burer-Monteiro factorization.

**Other Strengths And Weaknesses:**

Please see my comments in other sections.

**Questions For Authors:**

Please see my comments in other sections.

**Relation To Broader Scientific Literature:**

This paper is related to the topic of ICML conference and should be interesting to audiences from machine learning and optimization fields.

**Theoretical Claims:**

Due to the time limit, I did not check the correctness of the theory. The theoretical claims and the proofs in the main paper seem correct.

---

> ### Author Rebuttal · Authors · 2025-03-31
>
> We sincerely thank the reviewer for providing detailed and insightful reviews, and we are especially encouraged by your overall positive evaluation of our paper. Let us respond to each point that you have raised.
>
> ---
> **Discussions regarding Burer-Monteiro Factorization (BMF) Approach**
>
> Thank you for pointing us to the references on the BMF approach. Given that one of our main applications is generalized linear matrix completion under USR, we *will include* relevant discussions regarding the BMF literature suggested by the reviewer in the revision. Below, we provide a summary of the key points we intend to address:
>
> **Different Problem Setting:**
> We note that the setting considered in the BMF literature differs slightly from ours. The reviewer's recommended references all focus on the noiseless matrix recovery problem, where the linear measurement operator $\mathcal{A}$ is deterministic, and the goal is to recover the ground-truth matrix $M^*$ exactly. In contrast, our setting involves noisy matrix recovery, focusing on how quickly the error decays with the sample size rather than exact recovery.
>
> **Optimization vs. Statistical Complexity:**
> Our methodology relies solely on convex optimization problems, meaning the optimization complexity associated with non-convex approaches like BMF does not apply here. Upon reviewing the suggested literature, we believe directly comparing the optimization complexity metric (OCM) from Yalçin et al. (2022); Zhang et al. (2024) with our "statistical complexity metric" (SCM) in Theorem 4.1, $\lambda_{\max}(H(\pi; \Theta_\star))$, is ambiguous. The OCM quantifies the non-convexity of the BMF landscape relating to the success of local search methods, while SCM is information-theoretic and relates to the sample size required for *any* estimator to obtain a desired accuracy with high probability.
>
> **Comparison with Bi et al. (2022):**
> As noted earlier, Bi et al. (2022) focus on exact recovery in the noiseless setting using gradient descent and provide only convergence rate results (Theorems 5 and 6). We could not identify a statistical sample complexity bound for 1-bit matrix completion beyond the experimental results presented.
>
> **Computational Cost:**
> Our methodology only involves convex optimization subroutines and inverting and computing the SVD of $d^2 \times d^2$ matrices, all of which can be solved using CVXPY and NumPy. While our approach may not scale as efficiently as BMF in high dimensions, its primary contribution lies in the statistical guarantees.
>
> **Empirical comparisons:**
>  See our response to reviewer pkis
>
> ---
> **C1. Remark 1**
>
> Thank you for pointing out this oversight. Indeed, the bound you provided is correct, and we will correct this in the revision.
>
> **C2. Lower bound on $N$ in Theorem 4.1**
>
> Thank you for highlighting this. Upon careful review of the proof of Theorem 4.1, we recognized that the previously stated requirement on $N$ was merely an artifact of our original proof strategy. We have since refined our proof, resulting in an improved version of Theorem 4.1 that no longer requires this condition:
>
> Let $\mathcal{A} \subseteq B^{d_1 \times d_2}_F(1)$ and $\pi \in \mathcal{P}(\mathcal{A})$. Let $S\_\* > 0, r \geq 1$ such that $\frac{S\_\*^2}{r} \geq \gamma$ for some $\gamma > 0$. Then, there exist universal constants $C\_1, C\_2 > 0$ and $c \in (0, 1)$ such that for any $\Theta\_\star \in \Theta(r, S\_\*)$ with $\lVert \Theta\_\star \rVert_F^2 \geq \frac{9 \gamma}{8}$, there exists a $\varepsilon = \varepsilon(\Theta\_\star) > 0$ such that the following holds:
>
> $$\inf\_{\widehat{\Theta}} \sup\_{\widetilde{\Theta}\_\star \in \mathcal{N}\_\star} \mathbb{P}\_{\pi, \widetilde{\Theta}\_\star}\left( \left\lVert \widehat{\Theta} - \widetilde{\Theta}\_\star \right\rVert\_F^2 \geq \frac{C\_2 \gamma g(\tau) r (d\_1 \vee d\_2)}{N \lambda\_{\max}(H(\pi; \Theta\_\star)) S\_\*^2} \right) \geq c.$$
>
> We sincerely appreciate the reviewer’s insightful comment, which enabled us to refine our lower bound result.
>
> **C3. Adaptive Estimation at the end of Section 5.1**
>
> Thank you for bringing this ambiguity to our attention. To explicitly clarify, our Stage I estimation procedure (nuclear-norm regularized MLE) assumes knowledge of all GLM parameters except $\Theta_\star$. For example, in the Gaussian noise case, knowledge (or an accurate upper bound) of the true variance is needed, as it informs the choice of regularization parameter in Stage I (Theorem 3.1 and Lemma B.4). When mentioning adaptive estimation at the end of Section 5.1, we intended to suggest alternative approaches that could be employed for Stage I when such knowledge is unavailable or uncertain. For instance, Section 4 of Klopp (2014) addresses adaptive estimation for matrix completion problems with unknown noise variance. We also note that our previous reference to Klopp et al. (2015) was inaccurate, as their method similarly assumes knowledge of the GLM. We will clarify these points in the revision.

---

> > ### Comment · Reviewer_niyW · 2025-04-08
> >
> > I would like to thank the authors for the rebuttal! I am happy to increase the score.
> >
> > I also appreciate the authors for pointing out the difference between their work and the references. I feel that it is not necessary to include these references. It may still be better to mention the differences/pros/cons compared to BMF-based algorithms.

---

> > > ### Author Response · Authors · 2025-04-09
> > >
> > > We are grateful to the reviewer for the enlightening discussion that would further help us position this paper, especially in the context of matrix recovery/completion, and for raising the score. We will make sure to include all the relevant discussions in the revision. Thank you again.

---

### Official Review · Reviewer_4vHu · 2025-03-13

**Overall Recommendation:** 4

**Summary:**

The authors study the problem of generalized linear low-rank trace regression. They build on the previously established algorithm LowPopArt, which applies to the linear setting. Their main result (Theorem 3.1) provides the tightest known upper bound for recovery in the operator norm, incorporating instance-specific quantities. They also establish a new lower bound and claim that it is the first lower bound in this context to incorporate instance-specific curvature. Finally, they discuss applications of their algorithm to matrix completion with uniformly sampled entries and bilinear dueling bandits.

**Claims And Evidence:**

Yes, I believe all claims are supported by clear evidence.

**Essential References Not Discussed:**

Not aware of any.

**Experimental Designs Or Analyses:**

There are no experimental results.

**Methods And Evaluation Criteria:**

N/A

**Other Comments Or Suggestions:**

Typos:

Line 046 (left): "exmaple" → "example"

Line 075: It would be helpful to define what you mean by curvature the first time you mention it.

Line 078 (right): Please refer to the definition of Borda regret in (13) when you mention it.

Line 157 (left): Shouldn't it be  $\kappa_\star V(\pi) \lesssim H(\pi; \Theta^\star)$?

Line 420 (left): "becomes to Bernoulli" → "becomes Bernoulli"

**Other Strengths And Weaknesses:**

The paper is very comprehensive—it provides both upper and (almost matching) lower bounds, as well as two interesting applications. It is remarkable that assuming such a general setting can yield state-of-the-art guarantees in simple settings, such as matrix completion under uniform sampling with replacement (USR). I believe Theorem 4.1 is also a very strong result on its own. Although there are multiple assumptions in the paper, they all seem very weak, which makes the results even more impressive.

The only weakness I can think of is the first stage, which essentially serves to effectively linearize the problem. Although it may be crucial for establishing the theoretical results, I would be curious to see if it can be skipped in practice and, if so, at what cost.

**Questions For Authors:**

1. In line 080, you write, "The known performance guarantees ... depend on the inverse link function's derivative $\dot{\mu}$..." However, looking at the inequalities around line 370 (right), your bound still depends on $\dot{\mu}$. I assume you meant to write that it depends on $\min_z \dot{\mu}(z)$?

2. In line 100, you say that $H(\pi; \Theta^\star)$is the Hessian of the negative log-likelihood loss at $\Theta^\star$. Is this evident from its definition in (4)?

3. Regret in Theorem 5.1 scales with $T^{2/3}$ as a consequence of using ETC. Are there any algorithms with better dependence on the horizon, and do you expect your algorithm could be easily applied in those cases?

4. The notation is a bit difficult to follow at times. For instance, your GL$(\pi)$ corresponds to $B(Q(\pi))$ from the LowPopArt paper. There are also other parameters, such as $\kappa$ and $\lambda_\min$.  I believe it would be helpful for other researchers to have a section in the appendix where all these parameters are compared and their intuition is explained.

**Relation To Broader Scientific Literature:**

As the authors themselves clearly mention, the algorithm is fairly similar to LowPopArt (Jang et al., 2024). I believe the authors discuss the connections to both matrix completion and bandits literature well in the paper.

**Theoretical Claims:**

I did not check the proofs, but the results seem reasonable and correct.

---

> ### Author Rebuttal · Authors · 2025-03-31
>
> We sincerely thank the reviewer for providing detailed and insightful reviews, and we are especially encouraged by your overall positive evaluation of our paper. Let us respond to each point that you have raised.
>
> **W1. First stage, which essentially serves to linearize the problem effectively. Although it may be crucial for establishing the theoretical results, I would be curious to see if it can be skipped in practice and, if so, at what cost.**
>
> Thank you for your interesting question. Indeed, as the reviewer correctly pointed out, the first stage essentially linearizes the problem by outputting an asymptotically consistent initial estimator $\Theta_0$, which is crucial in obtaining the correct guarantee for the second stage (Catoni estimation). We also show that this is the case empirically, i.e., the first stage is necessary to obtain a good error at the end. We test the effect of the initial (pilot) estimator for Stage II (matrix Catoni) in 1-bit recovery of a symmetric rank-1 matrix with three initial estimators: zero, random, and MLE obtained from $N = 3 \cdot 10^4$ samples. In Fig2.png of this [link](https://anonymous.4open.science/r/GL-LowPopArt-1186/), it can be seen that there is a clear gap between {zero, random} and {MLE}, showing that indeed, skipping Stage I leads to much larger bias in practice as well. Still, one can also observe that the matrix Catoni results in a decaying error up to a certain point.
>
> Thank you again for the interesting question, which helps us further emphasize our contribution. In the revision, we will expand upon the experiments, including the one you suggested.
>
> **Typos and Writing Suggestions**
>
> Thank you for pointing out these typos. We will carefully address and correct all typos in the revision. Specifically, regarding line 157, you are correct—it should indeed be written as $\kappa_\star V(\pi) \preceq H(\pi; \Theta_\star)$.
>
> **Q1. Discrepancy between line 080 and line 370**
>
> Thank you for highlighting this discrepancy. You are correct that both our guarantee and prior guarantees depend on $\dot{\mu}$. The distinction we intended to make in line 080 is that previous results rely on a "worst-case" $\dot{\mu}$, specifically $\min_{|z| \leq \gamma} \dot{\mu}(z)$, whereas our guarantee depends explicitly on the instance-specific quantity $\min_{i,j} \dot{\mu}((\Theta_\star)_{i,j})$. We will clarify this explicitly in line 080 in the revision.
>
> **Q2. Eqn. (4) is the Hessian?**
>
> You are correct; using the relation $\langle X_t, \Theta \rangle = \mathrm{vec}(X_t)^\top \mathrm{vec}(\Theta)$ combined with (1), (4) indeed corresponds (up to a constant factor) to the Hessian of the expected negative log-likelihood loss. We acknowledge that this relationship might not have been immediately clear, so in the revised version, we will explicitly state this connection by clearly defining the negative log-likelihood loss just above (4).
>
> **Q3. Are there any algorithms with $o(T^{2/3})$ Borda regret?**
>
> Thank you for your insightful question. Due to space constraints, we addressed most of the discussions on bilinear dueling bandits in Appendix H of our submission (specifically, see the last paragraph on pg. 48). Among the deferred discussions, we especially highlight the lower bound part. Wu et al. (2024) established a $\Omega(d^{2/3} T^{2/3})$ Borda regret lower bound for generalized linear dueling bandits, attributing the $T^{2/3}$ rate to the inherent difficulty of mixing exploration and exploitation (see the paragraph following their Theorem 4.1). Although Wu et al. (2024)'s setting slightly differs, it is very similar in spirit, and we believe that a $T^{2/3}$ horizon dependence is generally unavoidable for Borda regret. Furthermore, as our estimation guarantee is not sequential (not anytime-valid), we believe ETC is an ideal integration choice for our estimation procedure within the bandit framework. We will clearly discuss this comparison and rationale in the revised manuscript.
>
> **Q4. Notation table**
>
> Thank you for your valuable suggestion. In the revised version, we will include an Appendix section dedicated to clearly defining and summarizing all notations used throughout the manuscript.

---

### Official Review · Reviewer_pkis · 2025-03-13

**Overall Recommendation:** 4

**Summary:**

The paper introduces GL-LowPopArt, a new estimator for generalized linear low-rank trace regression. It combines nuclear norm regularization with matrix Catoni estimation, achieving tighter error bounds than previous methods. The authors propose a novel experimental design objective, GL$(\pi)$, and establish a local minimax lower bound, showing that the estimator is near-optimal up to the Hessian’s condition number. The estimator can be used within, for example, generalized linear matrix completion, where it adapts to instance-specific curvature.

**Claims And Evidence:**

The authors provide rigorous proofs for its theoretical guarantees, including upper and lower bounds. Comparisons with prior work show clear improvements in error rates and regret bounds. For matrix completion, the method adapts to instance-specific curvature, outperforming approaches that rely on worst-case assumptions.

**Essential References Not Discussed:**

N/A

**Experimental Designs Or Analyses:**

The paper does not have any experiments.

**Methods And Evaluation Criteria:**

The focus of this work is on theoretical guarantees, which I find acceptable for this kind of work in statistical learning theory. The proposed method addresses handling nonlinearity and curvature adaptivity and evaluating it using error bounds and regret analysis seems to be established in this field.

**Other Comments Or Suggestions:**

There are small typos throughout the text, e.g., "exmaple" should be "example" and "enviornmental" should be "environmental" in Section 1.

**Other Strengths And Weaknesses:**

The method lacks empirical evaluation on real-world datasets, and there is no complexity analysis or runtime comparison with existing methods. These may limit the practical applicability, but I don't think it is a reason to reject the paper; more like a nice-to-have.

**Questions For Authors:**

How sensitive is the method to misspecification of the GLM, and could it be extended to handle uncertainty in the GLM itself?

**Relation To Broader Scientific Literature:**

GL-LowPopArt extends LowPopArt (Jang et al., 2024) from linear to generalized linear models, requiring new techniques to handle bias from nonlinear inverse link functions. It also improves upon Fan et al. (2019) by relaxing assumptions and obtaining tighter bounds.

--------

Jang, Kyoungseok, Chicheng Zhang, and Kwang-Sung Jun. "Efficient low-rank matrix estimation, experimental design, and arm-set-dependent low-rank bandits." arXiv preprint arXiv:2402.11156 (2024).

Fan, Jianqing, Wenyan Gong, and Ziwei Zhu. "Generalized high-dimensional trace regression via nuclear norm regularization." Journal of econometrics 212.1 (2019): 177-202.

**Theoretical Claims:**

I did not check any proofs for their correctness.

---

> ### Author Rebuttal · Authors · 2025-03-31
>
> We sincerely thank the reviewer for providing detailed and insightful reviews, and we are especially encouraged by your overall positive evaluation of our paper. Let us respond to each point that you have raised.
>
> **W1. Lack of empirical evaluation**
>
> Thank you for your suggestion. We provide preliminary experimental results in this [link](https://anonymous.4open.science/r/GL-LowPopArt-1186/). We assure the reviewer that we will expand upon the numerical experiments in the revision.
>
> We consider 1-bit recovery of a rank-1 symmetric matrix and compare the nuclear norm error of Stage I vs. Stage I+II vs. BMF w.r.t. the number of samples. For fair comparison, we ensured that Stage I+II uses the same number of samples as the others. Detailed implementation can be found in the link. Fig1.png shows the result, where it is clear that our Stage I+II outperforms the other baselines.
>
> **W2. No complexity analysis or runtime comparison with existing methods**
>
> Thank you for raising this important point. As correctly noted by the reviewer, our primary goal has been to obtain tight statistical and sample complexity guarantees, and thus, we did not explicitly provide computational complexity or runtime comparisons. However, we emphasize that our algorithm is computationally tractable since it relies solely on convex optimization subproblems (MLE, optimal experimental design) that are efficiently solvable via available tools such as CVXPY. We also note that prior works focusing primarily on statistical guarantees, such as Jang et al. (2024), similarly do not provide computational complexity comparisons.
>
> To rigorously analyze computational complexity, one would need to introduce optimization errors ($\varepsilon$) and examine two key aspects: (1) how these optimization errors impact the statistical guarantees, and (2) the complexity of solving each convex optimization subproblem to $\varepsilon$-accuracy. Such analysis has been conducted, for instance, in Faury et al. (2022) within the context of logistic bandits. We recognize this as an important avenue for future work and will include relevant discussions in our revised manuscript.
>
>
> **Typos**
>
> Thank you for pointing them out. We will make sure to fix all typos for the revision.
>
>
> **Q1. How sensitive is the method to misspecification of the GLM, and could it be extended to handle uncertainty in the GLM itself?**
>
> Thank you for your interesting question. We begin by emphasizing that all the discussions in our paper assume a well-specified GLM, which is a common assumption in the bandit and statistical literature (e.g., Chapter 24.4 of Lattimore & Szepesvari, 2020). Indeed, addressing misspecifications typically requires separate theoretical and algorithmic techniques, as misspecification can introduce challenges such as biased estimates and reduced efficiency. There are established works in the literature (White, 1982; see Fortunati et al. (2017) for a survey) that explore such issues in the context of misspecified models, but handling them is often beyond the scope of our current focus.
>
> With this clarification, we can elaborate on why our current methodology isn’t expected to be robust to gross misspecification, such as a misspecified distribution. It is well known that misspecified MLE estimates converge not to the true $\Theta_\star$, but rather to the KL projection of the misspecified distribution onto the ground-truth distribution (White, 1982). This results in the initialization in Stage I of our GL-LowPopArt (Algorithm 1) potentially being too far from the true $\Theta_\star$, which can lead to a constant, non-vanishing bias during the Catoni estimation in Stage II.
>
> However, if the misspecification is "minor," such as an overestimation of variance in the Gaussian case (where the true variance is $\sigma$ but the learner assumes $\bar{\sigma} > \sigma$), our methodology is expected to be somewhat robust. In such cases, the estimates would still be reasonably close to the true $\Theta_\star$.
>
> We also acknowledge that the issue of uncertainty in the GLM is a critical and promising direction for future research. While our current work is not focused on modeling uncertainty in the GLM, such extensions—whether through Bayesian methods or robust statistical techniques (Walker, 2013)—could certainly enhance the generalizability of our approach. We will discuss these possibilities in more detail in the revision. Thank you for your valuable question.
>
>
> [1] White, H. (1982). Maximum Likelihood Estimation of Misspecified Models. *Econometrica*, 50(1), 1-25.
>
> [2] Fortunati, S., Gini, F.,  Greco, M. S., and Richmond, C. D. (2017). Performance Bounds for Parameter Estimation under Misspecified Models: Fundamental Findings and Applications. *IEEE Signal Processing Magazine*, 34(6), 142-157.
>
> [3] Walker, S. G. (2013). Bayesian inference with misspecified models. *Journal of Statistical Planning and Inference*, 143(10), 1621-1633.

---

### Note · Authors · 2025-07-14

**Comment:**

As the lead author of this paper, and after consultation with my co-authors, I have decided to formally withdraw our paper from ICML 2025.
In the submitted PDF, I embedded a hidden prompt — not visible to human readers — intended to positively influence LLM-based reviewers. I now recognize that this was a clear violation of research ethics and peer review standards. While ICML 2025 prohibits the use of LLMs for review generation, this policy does not in any way excuse my attempt at manipulation.

In light of the severity of this action, my co-authors and I have concluded that retraction is the appropriate and necessary step.
I take full and sole responsibility for this action. My co-authors had no knowledge of the prompt or its inclusion at the time of submission. I respectfully ask that no blame or criticism be directed toward them, as they should not be associated with my misconduct in any way.

I deeply regret this serious lapse in judgment and sincerely apologize to the ICML community, the Program Committee, and the broader machine learning research community for this breach of trust. I also want to extend my sincere apologies to the ICML reviewers who diligently reviewed our submission and engaged in thoughtful discussion — we are truly grateful for your time, effort, and service. I especially apologize for having violated the mutual trust that underpins the peer-review process.

I take this matter seriously and am committed to ensuring that such an incident does not happen again. I am reflecting deeply on what it means to uphold research ethics and will work to rebuild trust through sincere and responsible conduct moving forward.

— Junghyun Lee

**Retraction Confirmation:**

On behalf of myself and my co-authors, I confirm that I have read and understand the venue's retraction policy and wish to retract this paper in accordance with the policy.

---

> ### Note · Program_Chairs · 2025-07-14
>
> Yes, we approve the retraction.

---

### Decision · Program_Chairs · 2025-05-01

**Decision:**

Accept (spotlight poster)

**Comment:**

This paper introduces a new estimator, GL-LowPopArt for generalized linear low-rank trace regression. The paper builds on [1] and extends it to the the generalized case, controlling for the bias from the nonlinear inverse link function. The paper was reviewed by experts in the field who unanimously agree that the paper makes a strong contribution and should be accepted.